# Catch-22: On the Fundamental Tradeoff Between Detectability and Robustness in LLM Watermarking

Kuheli Pratihar [1]    Debdeep Mukhopadhyay [1]

## Abstract

Large language models generate text by sampling tokens at random, a process now widely used for inference-time watermarking that verifies AI-generated content. We present an information-theoretic framework that captures the trade-off between robustness to text edits and detectability by observers who lack the watermark key or a keyless detector. The bounds we derive hold regardless of computational power, and what a keyless detector can actually achieve depends on what it can observe about the model and its outputs. At the heart of the analysis is an additive, Kullback-Leibler (KL) information measure that quantifies how well a hypothesis test can distinguish watermarked from unwatermarked text while the watermark remains stealthy. The measure remains zero for distribution-preserving schemes and increases with text length for token-level and sentence-level probability-modifying schemes. When edits are modeled as noise, the KL measure shrinks quadratically with the edit rate for token-level schemes and with an induced semantic flip rate for sentence-level schemes. This shrinkage exposes an unavoidable trilemma among robustness, stealth, and reliable verification. Guided by these limits, we propose a hybrid watermarking strategy that selects the Pareto-optimal scheme among distribution-preserving, semantic-level, and token-level methods based on the expected editing regime at deployment. Experiments on Llama-2-7B and Mistral-7B under paraphrasing attacks corroborate these theoretical predictions and confirm that the hybrid strategy lies near the Pareto frontier across the edit regimes we evaluate.

[1]Department of Computer Science and Engineering, Indian Institute of Technology Kharagpur, Kharagpur, India. Correspondence to: Kuheli Pratihar <its.kuheli96@gmail.com>.

*Proceedings of the 43rd International Conference on Machine Learning*, Seoul, South Korea. PMLR 306, 2026. Copyright 2026 by the author(s).

## 1. Introduction

The recent emergence of LLM watermarking has brought a persistent phenomenon into sharp focus: a trade-off between detectability and robustness is consistently observed across watermarking families (Fig. 1). Here, detectability captures how readily an outside observer can tell that a piece of LLM-generated text carries a watermark, while robustness captures how well that watermark stays verifiable after the text has been edited. Early token-level watermarking schemes that bias next-token probabilities, including the "green-list" approach of KGW (Kirchenbauer et al., 2023) and the unigram-style variants built for robustness (Zhao et al., 2024), hold up well under downstream edits, yet they leave statistical traces that even *keyless observers*, that is, third parties who do not hold the watermark key, can spot. The push for greater stealth has led to semantic and sentence-level schemes such as SemStamp, PMark, and SimMark (Hou et al., 2024; Huo et al., 2025; Dabiriaghdam & Wang, 2025), which place evidence at a coarser granularity so that the watermark survives paraphrasing and is far harder to catch with token-level tests.

This trade-off has also pushed researchers toward schemes that design the generator and the detector together to approach the best possible operating point. The distribution-adaptive watermarking algorithm (DAWA) couples the generator-side rule with the detection statistic (He et al., 2025), while HeavyWater and SimplexWater aim for the best worst-case detection in settings where the model is highly confident about its next token (Tsur et al., 2025). Outside of inference-time sampling, structural or training-time watermarks aim to stay robust and stealthy while preserving utility by embedding signals into model parameters or training dynamics (Gu et al., 2024; Block et al., 2025). At the far end of the spectrum, cryptographic distribution-preserving watermarks (Christ et al., 2024) are by construction perfectly undetectable to keyless observers, yet they break under even mild editing because verification depends on keeping fine-grained token alignment intact.

A central limitation of prior work is how the detectability-robustness trade-off is measured. Existing approaches rely almost entirely on detector-specific empirical statistics, in particular black-box tests and calibrated scores, such as z-

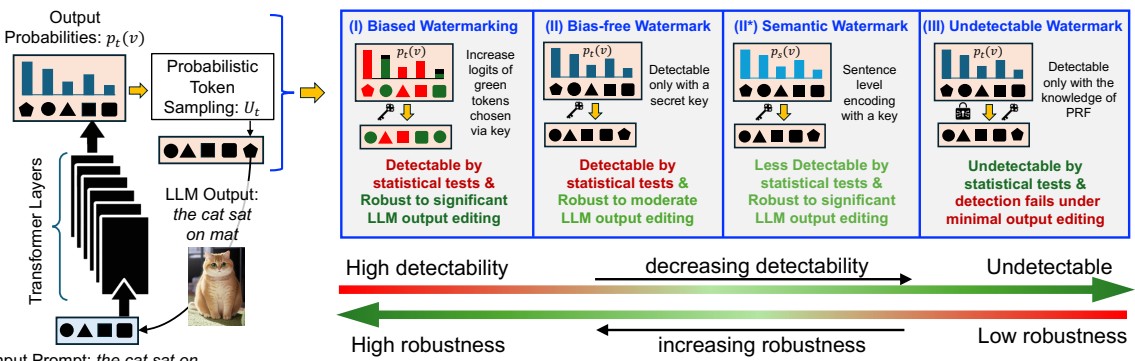

*Figure 1.* Watermarking schemes in modern LLMs exhibit a trade-off between detectability via statistical tests and robustness against LLM output editing.

scores, computed before and after paraphrasing or other editing attacks (Gloaguen et al., 2025; Liu et al., 2025; Li et al., 2025). While these tests do capture observable properties of watermarked text, *they fall short of a unified theoretical standpoint that can (i) compare different watermark families in a common currency and (ii) tie a scheme's robustness and keyless detectability directly to weaknesses built into its construction principles.*

In this work, we develop an information-theoretic framework for inference-time watermarking that maps the trade-off onto four watermark families: biased token-level, unbiased token-level, semantic-level, and distribution-preserving schemes. For token-level schemes, likelihood-ratio detectors are efficient when they have oracle or surrogate access to the model probabilities and the watermark rule, whereas for outsiders who only see the generated text and do not hold the watermark key, these bounds need not be tight. This gives a common language for analyzing existing proposals and for steering a hybrid strategy that sits empirically near the Pareto boundary. Our central quantity is the accumulated Kullback-Leibler (KL) divergence, which we treat as the *usable information budget*, that is, the amount of statistical signal a watermark can spend before its presence becomes visible. This budget upper-bounds stealth through Pinsker's inequality and lower-bounds verification power through Neyman-Pearson bounds. When text editing is modeled as a substitution noise process, the budget shrinks as more text is changed, yielding a trilemma for fixed-vocabulary LLMs.

Our framework proceeds in two steps. First, we measure detectability through total variation distance and establish a hierarchy across token-level and semantic-level schemes (Theorem 3.2). Second, we show how the KL budget shrinks under editing (Theorem 3.4). For token-level schemes, the information-theoretic limit can be reached efficiently once the detector has access to the model probabilities and the watermark rule, and, for keyed schemes, also to the key. For semantic schemes, optimal detection further requires the acceptance probabilities $a_t$. Building on these results,

we derive a *minimal-information hybrid selection rule* that picks among watermark families based on the anticipated edits and the desired stealth level (Theorem 4.2). Experiments on Llama and Mistral confirm that the hybrid strategy is near Pareto in post-edit verification across the regimes.

Our principal contributions are as follows:

1. **Detectability and post-edit verification characterization:** We show that a single KL quantity governs both stealth (upper-bounded through Pinsker) and robustness after edits (lower-bounded through Neyman-Pearson). This budget shrinks with the edit rate, yielding explicit feasibility regions (Theorem 3.4).

2. **Minimal-information hybrid selection rule:** We derive a rule that selects among watermarking families based on anticipated edits, choosing the method that requires the smallest KL budget to achieve a target verification power under a stealth cap (Theorem 4.2).

3. **Experimental validation:** We confirm the theoretical predictions through paraphrasing attacks on Llama and Mistral, showing near-Pareto post-edit verification at 15-30% edit rates while keeping a Pinsker TV bound below 0.1.

The remainder of the paper is organized as follows. Section 2 reviews existing watermarking approaches. Section 3 develops our information-theoretic framework. Section 4 derives the minimal-information hybrid selection rule. Section 5 experimentally validates the theoretical predictions, and Section 6 concludes the paper.

## 2. Related Works on LLM Watermarking and Research Gap

We organize inference-time watermarking schemes for LLMs along two axes: (i) *granularity*, which ranges from token-level to sentence-level, and (ii) whether the watermark *modifies* the output distribution under a fixed key or leaves it unchanged. This organization frames prior work and sets up the unified analysis in Section 3, with full technical de-

tail deferred to Appendix A. Existing token-level schemes modify generation in three distinct ways:

1. **Biased sampling** (Kirchenbauer et al., 2023; Zhao et al., 2024) marks certain tokens as "green" at each generation step and applies an exponential tilt to their sampling probabilities. These schemes are empirically robust (Kirchenbauer et al., 2024) but easy to detect using statistical tests (Gloaguen et al., 2025; Liu et al., 2025) and truncated goodness-of-fit methods (Li et al., 2025).

2. **Bias-free sampling** (Hu et al., 2024; Wu et al., 2024; Kuditipudi et al., 2024) uses reweighting functions $R_E$ that preserve the expected distribution, so that $\mathbb{E}_E[R_E(p_t)] = p_t$. Even with the mean preserved, the variance still gives the watermark away, and these schemes remain detectable through variance analysis (Gloaguen et al., 2025). Recent variants such as HeavyWater and SimplexWater (Tsur et al., 2025) cast watermark design as a minimax problem and improve detection in low-entropy regimes.

3. **Distribution-preserving sampling** (Christ et al., 2024; Zamir, 2024) leaves token probabilities exactly unchanged by using pseudorandom functions. These schemes are provably undetectable, but they break down completely once the LLM output is perturbed.

A recent extension by Golowich & Moitra (2024) adds substring robustness while keeping undetectability. The cost is a vocabulary size that grows polynomially in the security parameter, with degree $\Theta\!\left(\frac{1}{\alpha}\log\frac{1}{\alpha}\right)$ in an entropy-rate parameter $\alpha$. In plain terms, robustness to a constant fraction of edits at realistic entropy levels would demand vocabularies far larger than those used by current LLMs (see Appendix A.6).

Beyond token-level methods, several lines of work move to larger units of text to better resist meaning-preserving attacks. **Semantic and sentence-level watermarking** methods such as SemStamp (Hou et al., 2024), PMark (Huo et al., 2025), SIR (Liu et al., 2024), and SimMark (Dabiriaghdam & Wang, 2025) hide the watermark through keyed selection rules that act in embedding or proxy spaces. Working at the sentence level rather than the token level makes them less sensitive to surface-level rewrites and synonym swaps. All of these still fall under the probability-modifying category, because for any fixed key the selection rule yields a conditional distribution that differs from the base model. As a result, they improve robustness but do not escape the detectability-robustness trade-off (Theorem 3.4).

At an even larger granularity, **generator-side selection** methods such as WaterMax (Giboulot & Furon, 2024) sample several candidate continuations and pick the one that best matches a keyed watermark criterion. These methods trade logit manipulation for extra compute, yet the selection

*Table 1.* Notation guide for the main quantities used in Section 3.

| Symbol / Term | Meaning |
| --- | --- |
| $T$ | Number of tokens in the generated text. |
| $T_s$ | Number of sentences in the generated text. |
| $p_t(\cdot)$ | Baseline conditional distribution at step $t$. |
| $q_t(\cdot)$ | Watermarked conditional distribution at step $t$. |
| $P^s$ | Sequence distribution induced by the baseline sampler. |
| $Q$ | Sequence distribution induced by the watermarked sampler. |
| $\delta$ | Bias strength in biased token-level sampling. |
| $g_t$ | Baseline green-set mass at step $t$. |
| $\hat{\sigma}^2$ | Variance term for bias-free token-level watermarking. |
| $Z_t$ | Semantic evidence bit at sentence step $t$. |
| $\rho$ | Semantic bias parameter controlling watermark strength. |
| $D_0$ | Per-unit information budget before edits. |
| $\mathrm{TV}(P^s, Q)$ | Total variation between the baseline and watermarked distributions. |
| $\mathrm{KL}(Q\|P^s)$ | KL divergence from the watermarked distribution to the baseline distribution. |
| $\mathrm{Detect}_{\mathrm{IT}}$ | Information-theoretic detectability. |
| $\mathrm{Detect}_{\mathrm{comp}}$ | Computational detectability over efficient detectors. |
| keyless observer | Detector that only observes generated text, without access to the key, model probabilities, or watermark rule. |
| $D_\lambda$ | A randomized detector operating at security / problem size $\lambda$. |
| $\mathrm{Adv}_{D_\lambda}(\lambda)$ | Distinguishing advantage of detector $D_\lambda$, i.e., how well it separates watermarked from unwatermarked text. |
| $\mathrm{ED}(y, \tilde{y})$ | Edit distance between original text $y$ and edited text $\tilde{y}$. |
| $\varepsilon$ | Token edit rate after post-generation editing. |
| $\varepsilon_s$ | Probability that editing flips the semantic evidence bit. |
| $C_{\mathrm{tok}}(\varepsilon)$ | Usable post-edit information budget for token-level watermarking. |
| $C_{\mathrm{sem}}(\varepsilon)$ | Usable post-edit information budget for semantic watermarking. |
| $\alpha$ | False-alarm level of the detector. |
| $\beta$ | Target miss probability of the detector. |
| $D_{\mathrm{req}}^{\mathrm{tok}}$ | Required information budget for token-level schemes under the given edit regime. |
| $D_{\mathrm{req}}^{\mathrm{sem}}$ | Required information budget for semantic schemes under the given edit regime. |
| $\ell$ | Average number of tokens per sentence. |

step still shifts the distribution at the whole-text level, which keeps them in the probability-modifying family. WaterMax acts as a wrapper over existing watermarking schemes and inherits their detectability. Our taxonomy targets inference-time schemes, but **training-time methods** such as Gauss-Mark (Block et al., 2025) embed watermarks through weight perturbations, and they remain detectable to the extent that they shift the output distribution. We include GaussMark in our evaluation as a representative structural baseline.

**Research gap.** Despite rapid progress, existing analyses remain fragmented across scheme families and rarely deliver (i) a *cross-family detectability hierarchy* under one unified metric that places token-level, sentence-level, and distribution-preserving approaches on equal footing, (ii) a principled account of how *edits applied after generation erode the usable detection signal* beyond heuristic robustness measures, or (iii) an explicit *design rule* for choosing among scheme families given a target editing regime and stealth constraint. Our framework closes these gaps by linking detectability to a KL information budget and robustness to the way that budget shrinks under editing, which together justify a principled hybrid selection.

## 3. Robustness vs. Detectability Trade-off

The detectability and robustness of a watermark depend on how the model samples text during generation. At each step, the language model produces a probability distribution over the next token, and a sampling rule selects the token that

*Table 2.* Information-theoretic detectability bounds for fixed-key sampling methods.

| Sampling family | TV bound | Inline meaning |
|---|---|---|
| Greedy | $\mathrm{TV}(P^s, Q^{\mathrm{greedy}}) = 1 - P^s(\mathbf{y}^\star) = O(1)$ | $Q^{\mathrm{greedy}}$: point mass on $\mathbf{y}^\star$ |
| Biased ($\delta$-tilt) | $\mathrm{TV}(P^s, Q^{\mathrm{bias}_\delta}) \leq \|\delta\| \sqrt{\frac{1}{4} \sum_{t=1}^{T} g_t(1-g_t)} = O(\|\delta\|\sqrt{T})$ | $\delta$: bias strength; $g_t$: green-set mass |
| Bias-free (key $E$) | $\mathrm{TV}(P^s, Q_E^{\mathrm{bf}}) \leq \sqrt{\frac{1}{4} \sum_{t,v} \frac{\mathrm{Var}_E[R_E(p_t)(v)]}{p_t(v)}} = O(\sqrt{T})$ | $E$: key; $R_E$: reweighting operator |
| Semantic (key $k$) | $\mathrm{TV}(P^s, Q_k^{\mathrm{sem}}) \leq \sqrt{\frac{1}{2} \sum_{t=1}^{T_s} \mathbb{E}\left[\log \frac{1}{a_t}\right]} = O\left(\sqrt{T/\ell}\right)$ | $a_t$: acceptance mass; $T \approx \ell T_s$ |
| Distribution-preserving | $\mathrm{TV}(P^s, Q^{\mathrm{prf}}) = 0$ | $q_t \equiv p_t$ |

$P^s$ is the baseline text distribution from standard stochastic sampling and $Q$ is the corresponding watermarked distribution. $T$ counts token-generation steps, $T_s$ counts sentence-generation steps, and $\ell$ is the average tokens per sentence, so $T \approx \ell T_s$. Row-specific symbols are as follows. *Greedy*: $\mathbf{y}^\star$ is the deterministic greedy sequence and $Q^{\mathrm{greedy}}$ is the point mass on it. *Biased*: $\delta$ is the logit-tilt strength, $G_t$ the keyed green set at step $t$, and $g_t := p_t(G_t)$ its baseline mass. *Bias-free*: $E$ is the watermark key and $R_E$ the reweighting operator with $\mathbb{E}_E[R_E(p_t)] = p_t$. *Semantic*: $a_t$ is the acceptance probability at sentence step $t$. *Distribution-preserving*: $q_t \equiv p_t$ at every step.

actually appears in the output. The full distribution over generated text, therefore, depends on the model output probabilities, the sampling rule, and the secret key whenever one is used. In this section, we provide an information-theoretic framework that measures detectability and robustness for both token-level and semantic watermarking schemes, and use it to lay out the basic trade-offs that constrain every probability-modifying watermark. Our central quantity is the *usable KL budget*, which is the total KL divergence accumulated between the watermarked distribution and the baseline distribution. This budget, measured in bits, controls how cleanly a detector can separate the two distributions through hypothesis testing.

Randomness enters the generation process at every step $t$. In the token-level setting, the model produces a conditional distribution $p_t$ over the vocabulary, and a sampling rule uses a uniform random draw together with the secret key, when present, to pick the next token. In the semantic setting used by methods such as PMark and SemStamp, the sampler instead emits a full sentence at each step, and the watermark is applied through keyed sentence-level selection rules built from encoders, locality-sensitive hash functions, or proxy models. In either setting, a watermarked sampler can change the induced conditional distribution by making $q_t \neq p_t$ for a fixed key, or the source of randomness, or both. Any such change leaves a measurable statistical trace in the distribution over generated text, and we formalize that trace through the two notions of detectability defined below.

**Definition 3.1** (Detectability). Let $s$ denote a baseline sampling rule inducing distribution $P^s$ over texts $\Omega$, and let $\tilde{s}$ be a keyed watermarked rule inducing $Q^{\tilde{s}}$. The **information-theoretic detectability** is $\mathrm{Detect}_{\mathrm{IT}}(\tilde{s}) := \mathrm{TV}(P^s, Q^{\tilde{s}})$, with Pinsker's inequality providing the upper bound $\mathrm{Detect}_{\mathrm{IT}}(\tilde{s}) \leq \sqrt{\frac{1}{2}\mathrm{KL}(Q^{\tilde{s}}\|P^s)}$. The **computational detectability** is $\mathrm{Detect}_{\mathrm{comp}}(\tilde{s}_\lambda) := \sup_{D_\lambda \in \mathsf{PPT}} \mathrm{Adv}_{D_\lambda}(\lambda)$[1].

---

[1]$\mathrm{TV}(P, Q) := \sup_A |P(A) - Q(A)|$ is the total variation distance between $P$ and $Q$, and the *distinguishing advantage* of detector $D_\lambda$ is $\mathrm{Adv}_{D_\lambda}(\lambda) := \big| \Pr_{y \sim Q^{\tilde{s}}}[D_\lambda(y) = 1] - $

The tightness of each bound depends on what the detector is allowed to see, and we distinguish three access regimes that recur throughout the paper. An **oracle** detector has access to the true conditionals $p_t(\cdot \mid x, y_{<t})$, the watermark rule, and the key when acting as verifier. A **surrogate** detector instead uses an independently trained model $\hat{p}_t$ in place of the true conditionals. A **sample-only** detector observes only text samples, without access to any probabilities.

With these access regimes in mind, the two notions of detectability admit a clean comparison. Information-theoretic detectability is the best distinguishing advantage that any detector could achieve given unlimited computation, while computational detectability restricts the detector to efficient algorithms, so by construction $\mathrm{Detect}_{\mathrm{comp}} \leq \mathrm{Detect}_{\mathrm{IT}}$ (Lemma C.2). The two quantities coincide for oracle or surrogate detectors that can evaluate the likelihood ratio, and otherwise, only the upper bound is guaranteed. For sample-only detectors in particular, the Pinsker-based bound can be loose, and we therefore report $\sqrt{\frac{1}{2}\mathrm{KL}(Q\|P)}$ in our experiments as a Pinsker upper bound on the total variation rather than as a direct estimate of it.

### 3.1. Detectability Characterization

We now state bounds on information-theoretic detectability for each sampling family under fixed-key constructions.

**Theorem 3.2** (Information-theoretic detectability). *Fix a prompt $x$ and token length $T$, and let $P^s$ denote the baseline distribution induced by standard stochastic sampling. $\mathrm{TV}(P^s, Q)$ satisfies the bounds given in Table 2.*

**Informal summary.** If a watermark changes how the model samples text, then it leaves a detectable statistical trace, and that trace grows with the length of the generated text.

---

$\Pr_{y \sim P^s}[D_\lambda(y) = 1]\big|$, the absolute gap between $D_\lambda$'s acceptance rates on watermarked and unwatermarked text. The supremum is taken over PPT, the class of probabilistic polynomial-time detectors, that is, randomized algorithms whose running time is polynomial in the security parameter $\lambda$. All notation used in this section is summarized in Appendix B.

The watermarked distributions $Q^{\text{greedy}}$, $Q^{\text{bias}_\delta}$, $Q_E^{\text{bf}}$, $Q_k^{\text{sem}}$, and $Q^{\text{prf}}$ correspond to the five sampling families in the rows of Table 2, and the per-row symbols are defined in its footnote. The full proof is given in Appendix C.

> **Interpretation of Theorem 3.2**
>
> - **Evidence grows with length.** For probability-modifying schemes, $\text{TV}(P^s, Q)$ grows with text length as each step adds statistical evidence.
> - **Detectability hierarchy.** Greedy sampling leaves a constant-size trace of order $O(1)$, biased sampling grows as $O(|\delta|\sqrt{T})$, bias-free sampling as $O(\sqrt{T})$, and distribution-preserving sampling leaves no trace at all with $\text{TV} = 0$.
> - **Semantic schemes.** Semantic selection accumulates at $O(\sqrt{T_s}) = O\left(\sqrt{T/\ell}\right)$ for sentence length $\ell$.

For any probability-modifying watermark with a fixed key $k$, the watermarked distribution $Q_k$ differs from the baseline $P^s$, so $\text{TV}(P^s, Q_k) > 0$ and an information-theoretic distinguisher always exists. In the non-cryptographic setting, the resulting statistical drift admits test statistics that can be computed efficiently, consistent with recent statistical detectors that succeed in practice against both biased and bias-free watermarks (Gloaguen et al., 2025). Theorem 3.2 explains why these detectors grow more powerful as the text gets longer, and it provides the foundation for the robustness-detectability trade-off that we analyze next.

### 3.2. Robustness Analysis Under Text Perturbations

A core trade-off in watermarking is the balance between *stealth*, meaning low detectability to keyless outsiders, and *robustness*, meaning reliable verification by key holders even after the text has been edited. We measure stealth using KL divergence and total variation, and we measure robustness using detection power at miss probability $\beta$.

**Definition 3.3** (Robustness). Fix length $T$, edit tolerance $\varepsilon \in [0, 1]$, and false-alarm level $\alpha \in (0, 1)$. The family $\{\tilde{s}_\lambda\}$ is $(\varepsilon, \alpha, \beta)$-information-theoretically robust if there exists a level-$\alpha$ detector achieving power at least $1 - \beta$ on edited watermarked text satisfying $\text{ED}(y, \tilde{y}) \leq \varepsilon T$. Computational robustness is defined analogously with the supremum restricted to PPT detectors.

### 3.2.1. EDIT CHANNEL MODEL

We model edits using a mixture substitution channel. At each position, the original token is kept with probability $1 - \varepsilon$ and replaced with probability $\varepsilon$ by a draw from a replacement distribution. Our derivations use the i.i.d. simplification to obtain closed-form contractions, but the key $(1-\varepsilon)^2$ attenuation for token-level KL budgets arises for any mixture channel that linearly attenuates the perturbation (see Appendix D). This channel is a tractable first-order model, and its parameter $\varepsilon$ is set to match the actual fraction of text changed under real-world attacks. We empirically con-

firm that the predictions remain accurate under correlated paraphrasing edits in Section 5.

### 3.2.2. PER-UNIT KL BUDGETS AND USABLE INFORMATION AFTER EDITS

We now state how much watermark signal each step contributes and how much survives editing for each family. The quadratic approximations below assume small per-step perturbations, which is the regime watermarks are designed to operate in. Concretely, the biased tilt satisfies $|\delta| \ll 1$ with green-set mass $g_t := p_t(G_t)$ bounded away from $\{0, 1\}$, the bias-free perturbation satisfies $|\epsilon_{t,E}(v)| \leq \eta\, p_t(v)$ for some $\eta \ll 1$ so the watermarked distribution stays multiplicatively close to the baseline, and the semantic bias satisfies $|\rho| \ll 1$ so no single sentence is strongly tagged.

**Token-level signal.** For biased sampling, each token's watermark information budget, measured in bits, is $D_0^{(\text{biased})} \approx \delta^2 g_t(1 - g_t)/(2 \ln 2)$. The Bernoulli factor $g_t(1 - g_t)$ peaks at $g_t = 1/2$, that is, when the model places roughly half of its next-token probability on the green set, so a biased sampler invests most of its signal at steps where the model is genuinely uncertain about the next token. Under bias-free sampling, each token's watermark information budget is $D_0^{(\text{bias-free})} \approx \hat{\sigma}^2/(2 \ln 2)$, where $\hat{\sigma}^2 = \sum_v p_t(v) \text{Var}_E[R_E(v)]$ measures how much the reweighting varies across keys. The mean of the reweighting is fixed at $p_t$ by construction, so the watermark signal lives entirely in its variance, which is why distortion-free watermarks are hard to spot from text alone but still detectable to a verifier who holds the key. After editing at rate $\varepsilon$, the total usable watermark information budget shrinks to $C_{\text{tok}}(\varepsilon) \approx T(1 - \varepsilon)^2 D_0^{(\text{tok})}$. The $(1 - \varepsilon)^2$ factor combines two effects: each watermark shift survives editing only with probability $(1 - \varepsilon)$, and the KL contribution is quadratic in that surviving shift.

**Semantic signal.** A semantic watermark summarizes each sentence $S_t$ into a single keyed bit $Z_t := g_k(F(S_t)) \in \{0, 1\}$ obtained by passing the sentence embedding $F(S_t)$ through a locality-sensitive keyed hash $g_k$, so that sentences with similar meanings tend to produce the same bit. Without a watermark this bit is balanced at $\mathbb{E}[Z_t] = \frac{1}{2}$, and the watermark shifts it to $\frac{1}{2} + \rho$ by preferring candidate sentences whose bit lands on the desired side. Each sentence therefore carries a watermark information budget of about $D_0^{(\text{sem})} \approx 2\rho^2/\ln 2$ bits. Editing weakens this watermark signal only when it flips the sentence bit, an event captured by the induced semantic flip rate $\varepsilon_s(\varepsilon) := \Pr[g_k(F(\tilde{S}_t)) \neq g_k(F(S_t))]$, and the total usable watermark information budget after editing is $C_{\text{sem}}(\varepsilon) \approx T_s(1 - 2\varepsilon_s(\varepsilon))^2 D_0^{(\text{sem})}$. The factor of two in $2\varepsilon_s$ reflects that a flipped sentence bit not only erases its positive evidence for the watermark but contributes evidence against it of the same magnitude.

**Comparing the two.** Token-level schemes lose signal directly in proportion to $(1 - \varepsilon)^2$, while semantic schemes lose signal only through the flip rate $\varepsilon_s(\varepsilon)$, which can stay small even when $\varepsilon$ is large since many local edits leave the sentence-level meaning, and hence the sentence-level bit, unchanged. The ratio $\varepsilon_s(\varepsilon)/\varepsilon$ then summarizes how a given attack distributes its impact across the two scales. A ratio near 1 indicates a meaning-changing rewrite that hurts both families, while a ratio well below 1 indicates a clean paraphrase that hurts token-level schemes far more than semantic ones, and this ratio is what the selection rule in Section 4 uses to choose between the two families.

### 3.2.3. UNIFIED ROBUSTNESS-DETECTABILITY TRADE-OFF

**Theorem 3.4** (Robustness-Detectability Trade-off). *Fix $T$ tokens, $T_s$ sentences, edit rate $\varepsilon$, and false-alarm level $\alpha \in (0,1)$. For a level-$\alpha$ Neyman-Pearson test, achieving target miss probability $\beta$ requires $C(\varepsilon) \gtrsim \log_2(1/\beta)$. This yields the maximal tolerable edit rate for token-level schemes, $\varepsilon_\beta^{\mathrm{tok}} = 1 - \sqrt{\log_2(1/\beta)/(T D_0^{(\mathrm{tok})})}$, and the constraint on the semantic flip rate, $\varepsilon_s(\varepsilon) \leq \frac{1}{2}\left(1 - \sqrt{\log_2(1/\beta)/(T_s D_0^{(\mathrm{sem})})}\right)$.*

**Informal summary.** Editing erases part of the watermark signal, and reliable verification becomes impossible once the remaining signal falls below the $\log_2(1/\beta)$ bits a level-$\alpha$ test needs. For token-level schemes, the loss is set directly by the fraction of edited tokens, while for semantic schemes, it is set by how often editing flips the sentence-level bit. Token-level schemes, therefore, perform best under light editing, while semantic schemes take over when the wording changes substantially but the underlying meaning, and so the sentence-level bit, stays intact. The proof (Appendix D) follows from per-unit KL expansions, quadratic contraction under edits, and the chain rule. In practice, $\varepsilon$ is the normalized Levenshtein distance (Levenshtein, 1966) between the original and edited token sequences, and $\varepsilon_s(\varepsilon)$ is the fraction of aligned sentences whose evidence bit changes after editing (Appendix E.7).

> **Interpretation of Theorem 3.4**
>
> - **Quadratic contraction.** Robustness degrades as $(1 - \varepsilon)^2$ for token-level and $(1 - 2\varepsilon_s)^2$ for semantic schemes.
> - **Semantic resilience.** When attacks have high $\varepsilon$ but low $\varepsilon_s(\varepsilon)$, semantic schemes retain substantially more signal.
> - **Stealth-robustness tension.** Stealth requirements force small $D_0$, which reduces the tolerable corruption level.

### 3.3. Implications for Watermark Design

Theorem 3.4 gives a clean design rule. Robustness improves as we increase the redundancy ($T$ or $T_s$) and the per-unit

budget ($D_0$), and it degrades quadratically with the effective edit rate. Token-level schemes are the right choice when edits are uniformly distributed, and $\varepsilon$ is low. Semantic schemes are preferable when attacks heavily rewrite the surface form but preserve the underlying meaning. The ratio $\varepsilon_s(\varepsilon)/\varepsilon$ acts as a simple diagnostic for choosing.

**Corollary 3.5** (Impossibility Region). *Fix length $T$, watermark strength $D_0^{(\mathrm{tok})}$, and target power $1 - \beta$. Under the stochastic edit-channel model, if $\varepsilon > \varepsilon_\beta^{\mathrm{tok}}$, reliable detection is unattainable for a probability-modifying watermark with $(T, D_0^{(\mathrm{tok})})$, even for an information-theoretic detector.*

Corollary 3.5 formalizes the design dilemma, which is that one cannot at the same time achieve large edit tolerance and guaranteed verification. As a concrete example, with $T = 500$, $D_0^{(\mathrm{tok})} = 0.02$ bits per token, and $\beta = 0.01$, the maximal tolerable edit rate is $\varepsilon_\beta^{\mathrm{tok}} \approx 0.18$.

This trade-off raises a natural design question. *Given an anticipated edit regime, how should one select watermark parameters?* Section 4 answers this question through a composite loss that combines reliability, detectability, and parameter efficiency.

## 4. Optimal Watermark Selection Under Output Editing

Theorems 3.2 and 3.4 together suggest a clear design principle: to detect a watermark reliably under expected edits while still remaining hard to spot, one should use the smallest information budget that meets the robustness target. We turn this principle into a two-stage hybrid rule $\mathcal{H}$. Stage 1 picks a watermark family, distribution-preserving, token-level, or semantic, from the anticipated edit regime. Stage 2 picks a concrete scheme inside that family using a regime-conditioned score. The full optimization procedure and proofs appear in Appendix E.

### 4.1. Stage 1: Selecting the Watermark Family

Fix detector level $\alpha$ and miss probability $\beta$, so that the test has power $1 - \beta$. The Neyman-Pearson argument behind Theorem 3.4 tells us that reliable detection needs the available information to be at least $\log_2(1/\beta)$. Using the channel capacities from Section 3.2.2, the minimum per-unit information each family needs to reach the target power is

$$D_{\mathrm{req}}^{\mathrm{tok}}(\varepsilon, T, \beta) := \frac{\log_2(1/\beta)}{T(1 - \varepsilon)^2}, \qquad (1)$$

$$D_{\mathrm{req}}^{\mathrm{sem}}(\varepsilon, T_s, \beta) := \frac{\log_2(1/\beta)}{T_s(1 - 2\varepsilon_s(\varepsilon))^2}. \qquad (2)$$

Stealth places an upper limit on the same budgets. If an outside observer pools $M$ tokens and demands a total vari-

ation bound $\mathrm{TV} \leq \tau$, then Pinsker's inequality gives the KL-based cap

$$D_0^{(\mathrm{tok})} \;\leq\; D_{\mathrm{stealth}}^{\mathrm{tok}}(M, \tau) := \frac{2\tau^2}{M \ln 2}. \qquad (3)$$

Pooling $M_s$ sentences with budget $\tau_s$ gives the analogous semantic cap

$$D_0^{(\mathrm{sem})} \;\leq\; D_{\mathrm{stealth}}^{\mathrm{sem}}(M_s, \tau_s) := \frac{2\tau_s^2}{M_s \ln 2}. \qquad (4)$$

A family is feasible when its required budget stays within its stealth cap, that is, when $D_{\mathrm{req}} \leq D_{\mathrm{stealth}}$.

**Definition 4.1** (Hybrid Family Selection). Fix $(\varepsilon, T)$, $(\varepsilon_s(\varepsilon), T_s)$, target miss probability $\beta$, and stealth parameters $(M, \tau)$, $(M_s, \tau_s)$. Stage 1 of $\mathcal{H}$ picks a family as follows:

(i) If a distribution-preserving watermark with $K$ marked positions meets $\Pr[X < t] \leq \beta$ under $X \sim \mathrm{Binomial}(K, 1 - \varepsilon)$, pick it with $D_0 = 0$. The case $\varepsilon = 0$ is included here.

(ii) Otherwise, compute $D_{\mathrm{req}}^{\mathrm{tok}}$ and $D_{\mathrm{req}}^{\mathrm{sem}}$ from (1)–(2) and check each against the matching $D_{\mathrm{stealth}}$.

(iii) If both families are feasible, pick semantic watermarking when

$$D_{\mathrm{req}}^{\mathrm{sem}}(\varepsilon, T_s, \beta) < D_{\mathrm{req}}^{\mathrm{tok}}(\varepsilon, T, \beta), \qquad (5)$$

and pick token-level watermarking otherwise. If only one family is feasible, pick that one.

(iv) Set $D_0$ to the smallest feasible $D_{\mathrm{req}}$ for the chosen family.

The rule in (5) has a simple reading. Semantic watermarking is preferred when

$$\frac{(1 - 2\varepsilon_s(\varepsilon))^2}{(1 - \varepsilon)^2} > \frac{T}{T_s}, \qquad (6)$$

i.e., the semantic channel keeps a larger share of its original capacity than the token-level channel, once we adjust for the different unit counts. This holds when attacks change many tokens but rarely flip the underlying meaning, as with synonym substitution and meaning-preserving paraphrasing.

**Theorem 4.2** (Optimality of Stage 1). *Among schemes in this family class that meet the Pinsker-based stealth constraints (3)–(4) and reach power $1 - \beta$ at level $\alpha$ under edit rate $\varepsilon$, the hybrid rule $\mathcal{H}$ minimizes the KL-derived TV bound seen by keyless adversaries. The chosen family uses the smallest required information budget, and setting $D_0 = D_{\mathrm{req}}$ pays the smallest stealth cost while still meeting the robustness target. If neither family is feasible, no scheme in this class can reach the target power.*

In deployment, $\varepsilon$ and $\varepsilon_s(\varepsilon)$ are estimated offline by running the anticipated editing pipeline on representative outputs. The conservative version of $\mathcal{H}$ plugs upper-confidence

bounds $(\varepsilon^{\mathrm{U}}, \varepsilon_s^{\mathrm{U}})$ into $D_{\mathrm{req}}$ so that the chosen family meets the target power with high probability. The full estimation and robustification protocol is in Appendix E.7.

### 4.2. Stage 2: Selecting a Scheme Within the Family

Stage 1 fixes a family, but each family contains several published schemes, so a concrete deployment needs one more choice. Stage 2 scores every candidate $m$ at the estimated edit rate $\hat\varepsilon$ with

$$S(m, \hat\varepsilon) := \mathrm{AUC}(m, \hat\varepsilon) + \frac{1}{1 + \max(z(m, \hat\varepsilon), 0)}, \qquad (7)$$

where $\mathrm{AUC}(m, \hat\varepsilon)$ is the keyed verifier's AUC on watermarked text after attack and $z(m, \hat\varepsilon)$ is the keyless detectability $z$-score. The first term rewards schemes that the keyed verifier can still detect after editing, and the second penalizes schemes that an outside observer can spot. The candidate with the highest $S$ becomes the deployed scheme. For semantic candidates we use $\varepsilon_s(\hat\varepsilon)$ in place of $\hat\varepsilon$ in both terms. Table 4 in Appendix E.6 lists the scores at our operating points, where HCW wins the token-level family, PMark wins the semantic family, and CGW is the distribution-preserving representative.

## 5. Experimental Evaluation

This section tests our information-theoretic framework empirically by studying four families of *inference-time* watermarking schemes and measuring both their detectability and their robustness against paraphrasing attacks. The four families are *biased* token-level sampling, *bias-free* token-level sampling, *semantic* (sentence-level) rejection or selection methods, and *distribution-preserving* sampling. We also include several recent watermarking methods that extend or sit alongside these families: (i) **GaussMark** (Block et al., 2025), a *training-time* watermark baked into the model weights, (ii) **HeavyWater** and **SimplexWater** (Tsur et al., 2025), token-level designs that are bias-free in expectation for low-entropy next-token distributions, and (iii) **Sim-Mark** (Dabiriaghdam & Wang, 2025), a sentence-level similarity watermark. Practical deployments care about more than robustness and keyless detectability, since they also need to preserve output utility and keep runtime overhead under control. We therefore evaluate watermarking schemes on three fronts: (i) *robustness* of keyed verification after editing (AUROC, TPR at 1% FPR), (ii) *keyless detectability* by black-box statistical detectors, and (iii) *utility and cost* (MAUVE, BERTScore, LLM-as-judge, and compute overhead). Unless otherwise stated, all reported numbers are means over prompts.

All the relevant codes and a detailed user manual for replicating the experiments in this work are available at https://github.com/K1015/Catch-22-Pareto-Frontier-Watermark-in-LLMs.

*Table 3.* Robustness and detectability on Llama-2-7B across attack conditions. For each condition, we report AUROC (AUC), TPR at 1% FPR, and the keyless z-score ((Liu et al., 2025)). Higher AUC and TPR indicate greater robustness, whereas lower z-scores indicate lower detectability under this detector. Superscripts identify watermark families: [B] Biased, [F] Bias-free, [S] Semantic, [D] Dist-preserving, [W] Training-time, and [⋆] Hybrid. DAWA is shown separately as a co-designed baseline. The summarization column ([†]) reports the results of the summarization attack from *WaterJudge* with $\hat{\varepsilon} \approx 0.55$.

| Method | No attack | | | DIPPER ($\hat{\varepsilon}\approx 0.25$) | | | OPT-2.7B ($\hat{\varepsilon}\approx 0.15$) | | | WM-removal ($\hat{\varepsilon}\approx 0.15$) | | | Synonym ($\hat{\varepsilon}\approx 0.15$) | | | Back-trans. ($\hat{\varepsilon}\approx 0.42$) | | | Summ. [†] ($\hat{\varepsilon}\approx 0.55$) | | |
|---|---|---|---|---|---|---|---|---|---|---|---|---|---|---|---|---|---|---|---|---|---|
| | AUC | TPR | z | AUC | TPR | z | AUC | TPR | z | AUC | TPR | z | AUC | TPR | z | AUC | TPR | z | AUC | TPR | z |
| KGW[B] | .99 | 1.00 | 30.1 | .86 | .64 | 9.6 | .78 | .59 | 8.4 | .78 | .59 | 8.1 | .78 | .59 | 8.3 | .58 | .52 | −1.2 | .52 | .14 | −2.3 |
| Unigram[B] | .99 | 1.00 | 11.2 | .88 | .67 | 8.8 | .79 | .62 | 7.9 | .79 | .61 | 7.6 | .79 | .62 | 7.8 | .57 | .51 | −1.0 | .53 | .15 | −2.0 |
| DiPMark[F] | .99 | 1.00 | 43.2 | .90 | .80 | 3.9 | .91 | .86 | 3.6 | .90 | .85 | 3.4 | .91 | .86 | 3.5 | .60 | .54 | −0.8 | .55 | .18 | −1.1 |
| HCW[F] | .99 | 1.00 | 105 | .91 | .82 | 3.4 | .92 | .88 | 3.1 | .92 | .88 | 3.0 | .92 | .88 | 3.0 | .59 | .53 | −0.6 | .56 | .16 | −1.3 |
| HeavyWater[F] | .99 | 1.00 | 38.5 | .88 | .76 | 4.2 | .89 | .82 | 3.8 | .88 | .81 | 3.5 | .89 | .82 | 3.6 | .56 | .50 | −1.0 | .53 | .16 | −1.3 |
| SimplexWater[F] | .99 | 1.00 | 35.2 | .87 | .74 | 4.5 | .88 | .80 | 4.0 | .87 | .79 | 3.7 | .88 | .80 | 3.8 | .55 | .49 | −1.1 | .52 | .15 | −1.4 |
| Kuditipudi[F] | .99 | 1.00 | 27.5 | .93 | .85 | 3.4 | .94 | .89 | 3.7 | .93 | .88 | 3.5 | .94 | .89 | 3.6 | .65 | .43 | −1.2 | .62 | .28 | −1.6 |
| SemStamp[S] | .98 | .98 | 8.5 | .93 | .87 | 2.8 | .94 | .91 | 2.4 | .93 | .90 | 2.2 | .94 | .91 | 2.3 | .82 | .75 | 1.5 | .66 | .38 | 0.5 |
| PMark[S] | .99 | .99 | 7.2 | .94 | .89 | 2.5 | .95 | .93 | 2.1 | .94 | .92 | 1.9 | .95 | .93 | 2.0 | .85 | .79 | 1.2 | .64 | .43 | 0.3 |
| SimMark[S] | .98 | .97 | 9.1 | .92 | .85 | 3.0 | .93 | .89 | 2.6 | .92 | .88 | 2.4 | .93 | .89 | 2.5 | .80 | .72 | 1.7 | .64 | .35 | 0.7 |
| CGW[D] | .99 | 1.00 | −5.8 | .50 | .15 | −10 | .50 | .31 | −5.4 | .50 | .31 | −5.6 | .50 | .31 | −5.5 | .51 | .21 | −6.5 | .50 | .16 | −6.9 |
| GaussMark[W] | 1.00 | 1.00 | 12.4 | .82 | .71 | 8.6 | .85 | .76 | 7.8 | .83 | .74 | 7.3 | .84 | .75 | 7.5 | .62 | .55 | 2.1 | .56 | .21 | 0.4 |
| DAWA | 1.00 | 1.00 | 2.2 | .54 | .03 | 1.6 | .74 | .05 | 1.3 | .71 | .07 | 1.4 | .84 | .08 | 3.0 | .75 | .03 | 2.0 | .63 | .02 | 1.0 |
| **Hybrid**[⋆] | .99 | 1.00 | −7.8 | .94 | .89 | 4.2 | .96 | .93 | 3.8 | .95 | .92 | 4.3 | .95 | .93 | 4.4 | .86 | .79 | 1.8 | .61 | .47 | 0.6 |

Table 3 gives our main results on Llama-2-7B, reporting keyed robustness and keyless detectability across six conditions: no attack, two paraphrasers at moderate edit rates (DIPPER and OPT-2.7B), two lexical attacks (synonym substitution and watermark-removal paraphrasing), and back-translation at a high edit rate. Results for Mistral-7B, adaptive attack evaluation, utility metrics, and additional analyses appear in Appendix G.

> ## Experimental Setup
>
> ***Dataset and Models.*** We use 500 prompts from the LFQA dataset (Krishna et al., 2023), generating 100 to 1000 tokens per response with Llama-2-7B (Touvron et al., 2023) on a single NVIDIA H100 GPU. Mistral-7B (Jiang et al., 2023) results appear in Appendix G.
> ***Watermarking Schemes.*** We evaluate methods from each inference-time family: *biased* (KGW, Unigram (Kirchenbauer et al., 2023; Zhao et al., 2024)), *bias-free* (DiPMark, HCW (Wu et al., 2024; Hu et al., 2024)), *semantic* (SemStamp, PMark (Hou et al., 2024; Huo et al., 2025)), and *distribution-preserving* (CGW (Christ et al., 2024)). We additionally include HeavyWater and SimplexWater (Tsur et al., 2025), SimMark (Dabiriaghdam & Wang, 2025), and GaussMark (Block et al., 2025). Our **Hybrid** scheme (Theorem 4.2) dynamically selects among families based on edit characteristics.
> ***Attacks.*** Oblivious paraphrasers: DIPPER (Krishna et al., 2023) ($\hat{\varepsilon} \approx 0.25$), OPT-2.7B (Zhang et al., 2022) ($\hat{\varepsilon} \approx 0.15$), synonym substitution ($\hat{\varepsilon} \approx 0.15$), watermark-removal prompting ($\hat{\varepsilon} \approx 0.15$), and back-translation (Liu et al., 2025) ($\hat{\varepsilon} \approx 0.42$). Adaptive attacks are referred in Appendix G.
> ***Metrics.*** Keyed robustness: AUROC, TPR at 1% FPR. Keyless detectability: z-score detector (Liu et al., 2025), with the Tr-GoF detector (Li et al., 2025) discussed in Appendix G. Utility: MAUVE, BERTScore, LLM-as-judge (Appendix G).

### 5.1. Key Experimental Findings

**The robustness and detectability tradeoff sharpens with edit rate.** When no paraphrasing is applied, all schemes reach near-perfect robustness (AUROC ≥ 0.98), but de-tectability varies widely across families. Biased schemes have high z-scores (11.2 to 30.1), which makes them easy to spot with (Liu et al., 2025). CGW is not flagged by our keyless test and scores near the unwatermarked baseline, in line with its distribution-preserving design. As the edit rate grows, every family drifts away from the empirical Pareto frontier, and performance falls in step with the contraction predicted by Theorem 3.4, showing up as drops in AUROC and TPR. The effect is clearest under back-translation ($\hat{\varepsilon} \approx 0.42$), where even bias-free schemes fall to near-random performance (AUROC of 0.55 to 0.60).

**Semantic watermarks do better on verification metrics at moderate edit rates but lose ground under heavy paraphrasing.** Semantic schemes are more robust when token-level edits preserve the underlying meaning. Under the DIPPER attack ($\hat{\varepsilon} = 0.25$), PMark reaches AUROC of 0.94 against 0.91 for HCW, and under OPT-2.7B ($\hat{\varepsilon} = 0.15$) the gap widens to 0.95 against 0.92. These results match Theorem 3.4, which predicts that semantic schemes do well when the token edit rate $\varepsilon$ is high but the induced semantic flip rate $\varepsilon_s(\varepsilon)$ stays low. Under back-translation ($\hat{\varepsilon} \approx 0.42$), however, even semantic watermarks lose substantial ground. This supports the central message of the paper, which is that maintaining reliable verification at high edit rates usually requires stronger embeddings, and stronger embeddings increase detectability.

**The hybrid scheme matches or beats the best single-family method across conditions.** The hybrid scheme from Theorem 4.2 matches or beats the best evaluated baseline in most conditions while keeping detectability low. Under DIPPER, the hybrid matches PMark (AUROC of 0.94) by switching to semantic watermarking when paraphrasing preserves meaning but changes the surface tokens. Under back-translation, the hybrid achieves an AUROC of 0.86,

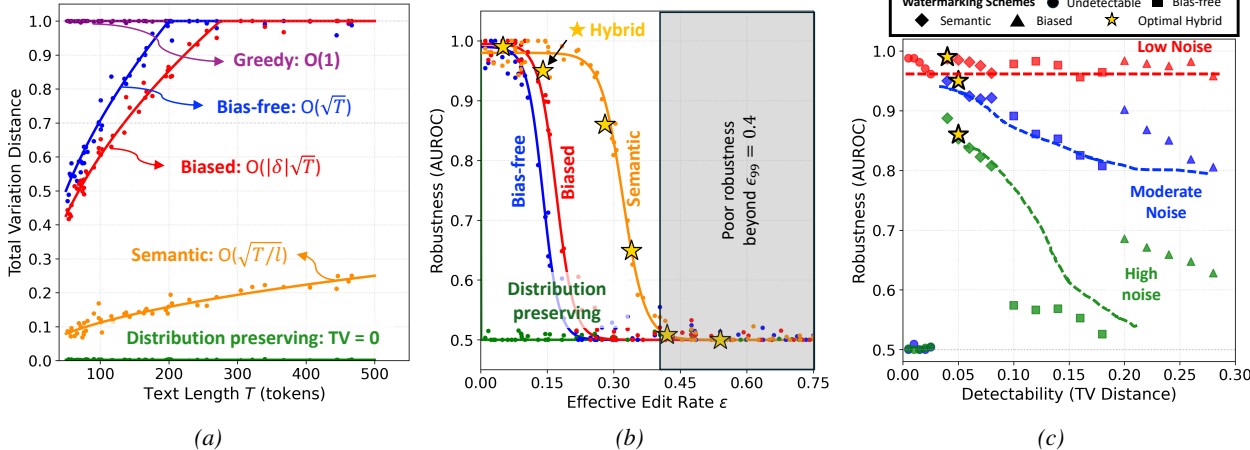

*Figure 2.* Empirical validation showing: (a) dependence of total variation (TV) on sampling rule and sequence length, (b) robustness AUROC versus edit noise in generated text, and (c) trade-off between attack resistance (indicated by high AUROC) and detectability (low TV) across low, moderate, and high noise regimes. The hybrid scheme aligns with the Pareto optimal boundary in every regime.

selecting the best feasible point when no evaluated method can simultaneously deliver high verification and low detectability. A sensitivity analysis shows that the hybrid selector relies on accurate estimates of $\varepsilon$ and $\varepsilon_s(\varepsilon)$, and a conservative selector that uses upper confidence bounds cuts worst-case regret with almost no loss in average performance (Appendix G).

**Recent baselines support rather than overturn the observed tradeoff.** GaussMark (Block et al., 2025) embeds a structural watermark at training time. Even though it sits outside our inference-time hierarchy, it still shifts the output distribution and remains detectable, reaching a z-score of 12.4 in the reference condition. HeavyWater and SimplexWater (Tsur et al., 2025) are designed to be bias-free in expectation for low-entropy regimes, but they prove less robust than standard bias-free methods under paraphrasing. SimMark (Dabiriaghdam & Wang, 2025) performs on par with SemStamp and PMark, which shows that sentence-level similarity watermarks still suffer degradation.

DAWA (He et al., 2025) jointly optimizes the embedder and detector and is strong in the clean regime, reaching a 100% detection rate at threshold 0.2 under its native matched detector. Under paraphrasing, however, DAWA's robustness drops sharply even with its own detector, with AUROC falling to 0.54 under DIPPER, 0.74 under OPT-2.7B, and 0.75 under back-translation, which is well below the Hybrid scheme in every edit regime in Table 3. This matches our framework's prediction that jointly optimizing the detector and embedder helps the clean operating point but does not protect against the edit-rate-driven contraction in Theorem 3.4.

**Empirical results line up with the theoretical predictions.** Figure 2(a) shows the scaling predicted by Theorem 3.2, where the Pinsker-based TV proxy stays at $O(1)$ for greedy decoding, grows as $\|\delta\|\sqrt{T}$ for biased sampling, grows as $\sqrt{T}$ for bias-free sampling, accumulates as $O(\sqrt{T_s}) \approx O(\sqrt{T/l})$ for semantic schemes, and remains near zero for distribution-preserving schemes. Figure 2(b) sweeps the i.i.d. edit rate up to $\hat{\varepsilon} \approx 0.75$ and reports detection performance across this range. The knee points where the Neyman-Pearson test still holds 99% power sit at $\hat{\varepsilon} \approx 0.15$ for biased schemes and $\hat{\varepsilon} \approx 0.32$ for semantic schemes, both inside the range predicted by Theorem 3.4. Robustness then drops smoothly with $\hat{\varepsilon}$ and reaches chance level once $\hat{\varepsilon}$ passes about $0.40$, which matches the degradation seen under back-translation at $\hat{\varepsilon} \approx 0.42$ and summarization at $\hat{\varepsilon} \approx 0.55$ in Table 3. The framework's predictions are therefore driven by the effective edit rate rather than by the structural details of any particular attack. Figure 2(c) plots the empirical tradeoff among the evaluated methods, where no single family delivers both high verification and low detectability across all regimes, and the hybrid consistently lies near the empirical Pareto frontier.

## 6. Conclusion

We present inference-time LLM watermarking as a hypothesis test under output editing and show that watermark information budget governs both stealth and post-edit verification across the biased, unbiased, semantic-level, and distribution-preserving families. This common currency frames the trilemma among robustness, stealth, and reliable verification as a budget-allocation problem, with computational attainability determined by what the detector can observe. The minimal-information hybrid selection rule that follows from this view spends the smallest budget needed to meet a target verification power under a stealth cap, and the resulting selector sits near the Pareto frontier across the edit regimes we evaluate. We believe this framework provides a starting point for analyzing the detectability, robustness, and stealth trilemma of emerging LLM watermarks.

## Impact Statement

**Practical Deployment.** Our framework casts watermark selection as a constrained information-allocation problem subject to a keyless-stealth budget. In access-controlled deployments, secure key management already provides much of the verification guarantee, so watermarks that introduce only small statistical drift, such as distribution-preserving or unbiased token-level schemes, are sufficient. In open deployments, the watermark itself must meet explicit detectability bounds and be recalibrated periodically as the editing distribution shifts. The minimal-information principle then yields a clear selection rule. Token-level schemes are preferable when the substitution rate is low, whereas semantic schemes are preferable when the substitution rate is large. The joint regime in which both rates exceed their respective critical thresholds constitutes an impossibility region where no probability-modifying construction can guarantee reliable verification. In practical settings, treating the edit regime as an estimable runtime parameter rather than a design-time prior yields a Pareto-optimal selection of watermarking schemes.

**Future Work.** Our edit model captures the dominant structure of common attacks but does not yet cover watermark-aware adaptive rewriting that deliberately decouples token and semantic changes. Regime-aware selection also presumes that edit levels can be estimated reliably, which calls for robust estimators paired with conservative fallbacks under active adversaries. Evaluation protocols should jointly report robustness, keyless detectability, and output quality, because reporting any one of these alone hides the trade-off practitioners actually face. Looking beyond inference-time watermarking, training-time schemes (Gu et al., 2024; Block et al., 2025) face a different threat surface centered on fine-tuning and distillation rather than text editing, and bringing them under the same minimal-information lens is a natural future research direction.

## Acknowledgements

Our profound appreciation goes to the anonymous reviewers for their constructive feedback, which greatly refined this paper. Additionally, the authors are thankful for the partial funding received from the Center for Hardware-Security Entrepreneurship Research & Development (C-HERD) and Information Security Education and Awareness (ISEA), both initiatives of MeitY, Govt. of India.

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

# A. Extended Review of LLM Watermarking Literature

This appendix extends Section 2 with a more detailed comparison of existing watermarking schemes, grouped by where the watermarking signal is added during generation, how a detector tests for it, and what quality guarantee the scheme offers.

## A.1. Token-Level Probability-Modifying Watermarks

These watermarks act on individual tokens, shifting the next-token distribution at each step by an amount that depends on a secret key.

**Biased Sampling.** The KGW scheme (Kirchenbauer et al., 2023) hashes previous tokens at step $t$ to split the vocabulary into a green list $G_t$ and a red list, then shifts the logits using $\hat{\ell}_t[v] = \ell_t[v] + \delta \cdot \mathbf{1}[v \in G_t]$. After the softmax, this multiplies green-token probabilities by about $e^\delta$, so the text contains more green tokens than a random guess. The Unigram-Watermark (Zhao et al., 2024) fixes the partition for all positions instead of rehashing, which improves robustness and gives tighter quality bounds through a Rényi divergence analysis. Both schemes leave a detectable signal that grows as $O(|\delta|\sqrt{T})$, easy to spot with a chi-square test and easy to amplify with adversarial prompts.

**Bias-Free Sampling.** These methods hide the watermark from tests that look only at the average token distribution. Let $p_t$ denote the base next-token distribution at step $t$ and let $E$ denote the secret key. These methods preserve the unwatermarked next-token distribution in expectation over the key, using a key-dependent reweighting function $R_E$ that satisfies $\mathbb{E}_E[R_E(p_t)] = p_t$ (Hu et al., 2024). The same goal is reached through key-dependent vocabulary permutations (Wu et al., 2024) or through inverse transform sampling (Kuditipudi et al., 2024). Higher moments are not preserved. The variance gains a key-dependent component of order $O(\sqrt{T})$, which a variance test can detect (Gloaguen et al., 2025). HeavyWater and SimplexWater (Tsur et al., 2025) pose watermark design as a worst-case optimization between encoder and detector, improving detection on low-entropy text through carefully chosen code structures and scoring rules while keeping the output distribution unchanged in expectation.

## A.2. Semantic-Level Watermarks

Token-level schemes break under paraphrase because their green and red labels are attached to specific words. Semantic schemes fix this by placing the signal at the level of meaning, which a faithful paraphrase preserves. The common recipe is rejection sampling at the sentence level. Given a context $\pi$ and the model's base distribution $P_M(\cdot \mid \pi)$, the scheme draws candidate sentences and keeps only those that pass a key-dependent test $A_k(s) = 1$, giving $P_M^w(s \mid \pi; k) = \frac{P_M(s|\pi)\mathbf{1}\{A_k(s)=1\}}{P_M(A_k=1|\pi)}$. Individual schemes differ in how they construct $A_k$.

SemStamp (Hou et al., 2024) builds $A_k$ from locality-sensitive hashes of sentence embeddings, with margin constraints that prevent small edits from flipping the bit. PMark (Huo et al., 2025) uses balanced partitions so that the selection is distortion-free when averaged over keys, although biased for any single key. SIR (Liu et al., 2024) encodes the watermark into features designed to survive rewording. SimMark (Dabiriaghdam & Wang, 2025) avoids the model logits altogether and accepts a sentence only when the cosine similarity between consecutive sentence embeddings falls into a keyed interval, with a soft counting rule for paraphrase resilience. In every case, the token-level artifacts of earlier schemes are traded for sentence-level selection artifacts, and the basic stealth-robustness tension remains.

## A.3. Alternative Embedding Strategies

Another group of schemes leaves the sampling distribution untouched at generation time and instead moves the watermarking step elsewhere in the pipeline.

**Generator-Side Selection.** WaterMax (Giboulot & Furon, 2024) keeps the base distribution unchanged, draws $K$ candidate generations, and returns the one with the highest keyed score, $y^* = \arg\max_{i \in [K]} \text{score}_k(y^{(i)})$. Each candidate is drawn from the natural model distribution, so its quality matches the unwatermarked model. The bias appears only in the selected text, and the generation cost grows linearly with $K$.

**Detector-in-the-Loop Co-Design.** A different approach designs the embedding rule together with the detector that will read it. DAWA (He et al., 2025) optimizes the two jointly under explicit false-positive and distortion limits, so the embedding adapts to the local token distribution rather than applying a uniform shift. The scheme is, however, tuned to one detector and may not generalize.

**Training-Time Methods.** Other schemes move the watermark out of the sampling step entirely and into the model itself. GaussMark (Block et al., 2025) perturbs the model weights with a small secret Gaussian key and verifies authorship through a gradient correlation test. The catch is that the weight perturbation also moves the output distribution, so the watermark remains detectable from outputs alone whenever an observer can estimate the unperturbed distribution well enough.

**Evaluation Standards.** A complementary thread focuses not on new embedding rules but on how schemes should be compared. WaterPark (Liang et al., 2025) and similar platforms argue for adaptive watermark-aware attacks, quality-matched comparisons, and careful hypothesis testing. We follow these principles in Section 5.

### A.4. Distribution-Preserving Watermarks

Distribution-preserving schemes keep the marked and unmarked token distributions exactly equal at every step, $q_t \equiv p_t$, so no statistical test on a single output can separate them on average. The verifier instead relies on a secret pseudorandom function that links specific output positions to a key (Christ et al., 2024). The cost is fragility under editing. A light paraphrase shifts token positions, breaks the link, and erases the signal. Adding redundancy to repair the alignment introduces the statistical drift that the scheme sought to avoid.

### A.5. Detection Methods

Better detectors expose weaknesses across every family above. For expectation-preserving schemes, the second-moment test $T = \sum_{t=1}^{n} \left( \|\hat{p}_t\|_2^2 - \mathbb{E}[\|p_t\|_2^2] \right)$ sums, over the $n$ generated tokens, the gap between the observed squared $\ell_2$ norm of the empirical token distribution $\hat{p}_t$ and its expected value under the unwatermarked model $p_t$. The squared $\ell_2$ norm measures how concentrated a distribution is, so a watermark that quietly piles probability mass onto key-favored tokens raises $\|\hat{p}_t\|_2^2$ even when the mean is preserved, and the test reads this variance gap directly (Gloaguen et al., 2025). Active attacks via adversarial prompting amplify the watermark bias on demand, cutting the sample size needed for detection by orders of magnitude (Liu et al., 2025). Biased schemes fall to frequency analysis, bias-free schemes to variance tests, and both to adversarial prompting.

A separate question is how well detection works after a human edits the watermarked text. The analysis of Li et al. (2025) models the edited text as a mixture in which only an $\varepsilon$ fraction of positions still carries watermark evidence and the rest look unwatermarked. As the generation length $n$ grows, the watermarked fraction shrinks as $\varepsilon_n \asymp n^{-p}$ while the per-token entropy decays as $n^{-q}$, so $p$ controls how fast edits erase the signal and $q$ controls how predictable the underlying text becomes. Detection is possible if and only if $q + 2p < 1$, which says the watermarked fraction must shrink slowly enough relative to the entropy for any signal to remain. Their Truncated Goodness-of-Fit (Tr-GoF) test matches this optimal boundary without requiring knowledge of $p$ or $q$ in advance. Simpler detectors that sum per-token scores stop working at the tighter boundary $q + p = 1/2$ and become much less reliable once the text has been edited.

### A.6. Coding-Theoretic Constructions

A different line of work builds watermarks from coding theory. The construction of Golowich & Moitra (2024) uses indexing pseudorandom codes, which embed a hidden index into each output symbol and survive a constant fraction of adversarial edits over high-entropy substrings. The scheme lifts a binary pseudorandom code to a larger-alphabet code whose extra symbols give the redundancy needed to absorb insertions and deletions. For an entropy-rate parameter $\alpha \in (0, 1)$ that captures how unpredictable the underlying text is, the scheme tolerates an edit fraction $p = \Theta(\alpha^2)$ but requires an alphabet of size at least $|\Sigma(\lambda)| \geq n(\lambda)^{C_2 \frac{1}{\alpha} \log \frac{1}{\alpha}}$. Since the exponent grows as $\Theta(\frac{1}{\alpha} \log \frac{1}{\alpha})$, this alphabet is far larger than the 30k to 100k tokens of a real LLM vocabulary at realistic entropy. Their result therefore shows that edit-tolerant watermarking is possible when the alphabet can be made arbitrarily large, while our Theorems 3.2 and 3.4 show that the same goal is information-theoretically impossible once the alphabet is fixed to a deployed model's vocabulary.

## B. Notation and Variables

### Notation Conventions

We use the following notation throughout the paper. The set $\mathcal{V}^T$ denotes all token sequences of length $T$ drawn from a vocabulary $\mathcal{V}$, with the subscript $t$ indexing token position from 1 to $T$. A superscript on $Q$ names the sampling method, and an asterisk ($*$) marks an optimal value. All conditional distributions such as $p_t(\cdot)$ are implicitly conditioned on the preceding tokens $\mathbf{y}_{<t}$ and the prompt $x$.

## Watermarking Parameters

The following table lists the scheme-specific parameters that appear across the watermarking families.

| Symbol | Type/Dim | Description | Sections |
|---|---|---|---|
| $\delta, \delta^\star$ | Scalar | Bias strength (optimal value $\delta^\star$) | §3, §4 |
| $G_t \subset \mathcal{V}$ | Set | Keyed green token set at step $t$ | §3 |
| $g_t = p_t(G_t)$ | $[0,1]$ | Baseline green mass at step $t$ | §3, App. C |
| $\gamma, \gamma^\star$ | $[0,1]$ | Typical/target green mass (often $\gamma^\star = \frac{1}{2}$) | §3, §4 |
| $k$ | Key | Secret cryptographic key | §3 |
| $E, E_t$ | Code | Keyed code or permutation for bias-free schemes | §3 |
| $R_E$ | Function | Reweighting operator with $\mathbb{E}_E[R_E(p_t)] = p_t$ | §3 |
| $\sigma^2(v), \hat{\sigma}^2$ | Scalar | $\sigma^2(v) = \mathrm{Var}_E[R_E(p_t)(v)]$, $\hat{\sigma}^2 = \sum_v p_t(v)\sigma^2(v)$ | §3, App. D |
| $Z_t$ | $\{0,1\}$ | Keyed binary indicator: $Z_t := g_k(F(S_t))$ | §3 |
| $F$ | Function | Semantic feature map for sentence embedding | §3 |
| $g_k$ | Function | Keyed predicate mapping embeddings to $\{0,1\}$ | §3 |
| $\rho$ | $(0, \frac{1}{2}]$ | Semantic bias parameter controlling watermark strength | §3 |
| PRF | Function | Pseudorandom function for RNG replacement | §3 |
| $\mathcal{W}, \mathcal{W}^\star(\varepsilon)$ | Scheme | Watermarking scheme and the optimal hybrid | §4, App. E |
| $\mathcal{H}$ | Rule | Hybrid watermark selection rule | §4 |
| $K, t$ | Scalars | DP verifier: marked positions $K$ and correction radius $t$ | §4, App. E |

## Important Variables and Distributions

The next table collects the generic objects used throughout the paper, including text sequences, sampling rules, and the induced distributions over outputs.

| Symbol | Type/Dim | Description | Sections |
|---|---|---|---|
| $L, T$ | Scalar | Text length (number of tokens) | §3, §4 |
| $T_s$ | Scalar | Number of sentences | §3, §4 |
| $\mathcal{V}, \Sigma$ | Set | Token vocabulary | §3 |
| $\Sigma^\star$ | Set | Set of all finite sequences over $\Sigma$ | §3 |
| $\Omega$ | Set | Space of complete texts | §3 |
| $x$ | Vector | Initial prompt | §3 |
| $\mathbf{y} = (y_1, \ldots, y_T)$ | $\mathcal{V}^T$ | Generated token sequence | §3 |
| $\mathbf{y}_{<t}$ | $\mathcal{V}^{t-1}$ | Tokens before position $t$ | §3 |
| $\tilde{\mathbf{y}}$ | $\mathcal{V}^T$ | Edited/noisy text | §3 |
| $\mathbf{y}^\star$ | $\mathcal{V}^T$ | Deterministic greedy path | §3 |
| $S_t$ | $\Sigma^\star$ | Sentence at generation step $t$ | §3 |
| $\widetilde{S}_t$ | $\Sigma^\star$ | Edited sentence at step $t$ | §3 |
| $p_t(\cdot), p_\theta(\cdot|\cdot)$ | Function | Baseline LLM conditional probabilities | §3 |
| $q_t(\cdot)$ | Function | Watermarked conditional probabilities | §3 |
| $s, \tilde{s}$ | Rule | Baseline and watermarked sampling rules | §3 |
| $P^s$ | Distribution | Baseline sampling distribution over sequences | §3 |
| $Q^{\mathcal{W}}$ | Distribution | Sequence distribution for scheme $\mathcal{W}$ | §3 |
| $Q^{\mathrm{greedy}}$ | Distribution | Greedy sampling distribution | §3 |
| $Q^{\mathrm{bias}_\delta}$ | Distribution | Biased (tilted) sampling with parameter $\delta$ | §3 |
| $Q_E^{\mathrm{bf}}$ | Distribution | Bias-free sampling with key/code $E$ | §3 |
| $Q^{\mathrm{prf}}$ | Distribution | PRF-based distribution-preserving sampling | §3 |
| $Q_k^{\mathrm{sem}}$ | Distribution | Semantic watermarked distribution with key $k$ | §3 |
| $U_t$ | $[0,1]$ | Uniform random variable used for sampling | §3 |
| $U$ | $\Delta(\Sigma)$ | Uniform distribution on $\Sigma$ | App. D |
| $T_\varepsilon(P)$ | Operator | Edit channel: keep $P$ with prob. $1-\varepsilon$, replace by uniform $U$ with prob. $\varepsilon$ | App. D |
| $p_{t,\varepsilon}, q_{t,\varepsilon}$ | Function | Edited conditionals: $T_\varepsilon(p_t), T_\varepsilon(q_t)$ | App. D |
| $\mathrm{Bern}(p)$ | Distribution | Bernoulli distribution with success probability $p$ | §3 |

## Detectability and Robustness

This table gathers the quantities used to formalize detection and to measure robustness to edits, together with the detector-side statistics that the proofs invoke.

| Symbol | Type/Dim | Description | Sections |
|---|---|---|---|
| $\text{Detect}_{\text{IT}}(\tilde{s})$ | $[0,1]$ | Information-theoretic detectability (unbounded detector) | §3 |
| $\text{Detect}_{\text{comp}}(\tilde{s}_\lambda)$ | $[0,1]$ | Computational detectability (polynomial-time detector) | §3 |
| $\text{Adv}_{D_\lambda}(\lambda)$ | $[0,1]$ | Distinguishing advantage for detector $D_\lambda$ | §3 |
| $D_\lambda$ | Detector | Probabilistic polynomial-time detector | §3 |
| $\lambda$ | Scalar | Security parameter | §3 |
| PPT | Set | Class of probabilistic polynomial-time algorithms | §3 |
| $\text{ED}(y, \tilde{y})$ | Scalar | Edit distance between sequences | §3 |
| $\text{TV}(P, Q)$ | $[0,1]$ | Total variation distance | §3, App. C |
| $\text{KL}(Q\|P)$ | $[0,\infty)$ | Kullback-Leibler divergence (base 2 in proofs) | §3 |
| $\text{Detect}(s)$ | $[0,1]$ | Distinguishability for sampling rule $s$ | §3 |
| $\varepsilon, \hat{\varepsilon}$ | $[0,1]$ | Edit rate (true and estimated) | §3, §4 |
| $\varepsilon_s(\varepsilon)$ | $[0,1]$ | Induced semantic flip rate under token edit rate $\varepsilon$ | §3, §4 |
| $\varepsilon_\beta^{\text{tok}}$ | $[0,1]$ | Maximal tolerable edit rate for token-level schemes | §3 |
| $\alpha, \beta$ | $[0,1]$ | Detector level and miss probability (power $= 1 - \beta$) | §4, App. D |
| $z, z_{\text{threshold}}$ | Scalar | Z-score statistic and threshold | App. D |
| $N_{\text{green}}$ | Scalar | Count of green tokens | App. D |
| $\Phi(\cdot), \Phi^{-1}(\cdot)$ | Function | Standard normal CDF and its inverse | §4 |

## Information Budgets and Channel Capacities

This table lists the per-token and per-sentence information budgets, together with the channel capacities that govern the detectability-robustness trade-off in Section 3.

| Symbol | Type/Dim | Description | Sections |
|---|---|---|---|
| $D_0$ | Bits/token | Per-token information at zero edits | §3, App. D |
| $D_0^{(\text{tok})}$ | Bits/token | Per-token information for token-level schemes | §3 |
| $D_0^{(\text{biased})}$ | Bits/token | Per-token information for biased sampling | §3 |
| $D_0^{(\text{bias-free})}$ | Bits/token | Per-token information for bias-free sampling | §3 |
| $D_0^{(\text{sem})}$ | Bits/sentence | Per-sentence information for semantic schemes | §3 |
| $D_\varepsilon$ | Bits/token | Per-token information at edit rate $\varepsilon$ | App. D |
| $C(\varepsilon)$ | Bits | Total usable information $\approx T(1-\varepsilon)^2 D_0$ | §3, App. D |
| $C_{\text{tok}}(\varepsilon)$ | Bits | Token-level channel capacity: $T(1-\varepsilon)^2 D_0^{(\text{tok})}$ | §3 |
| $C_{\text{sem}}(\varepsilon)$ | Bits | Semantic channel capacity: $T_s(1-2\varepsilon_s)^2 D_0^{(\text{sem})}$ | §3 |
| $\varepsilon_\beta(T, D_0)$ | $[0,1]$ | "Knee": $1 - \sqrt{\log_2(1/\beta)/(TD_0)}$ | App. D |
| $H(\cdot), H_2(\cdot)$ | Function | Entropy, binary entropy | §3 |

## Design Rule Parameters

The following table lists the parameters that enter the design rule of Section 4.1, which trades off reliability, stealth, and signal amplitude.

| Symbol | Type/Dim | Description | Sections |
|---|---|---|---|
| $D_{\text{req}}(\varepsilon, T, \beta)$ | Bits/token | Required information: $\log_2(1/\beta)/T(1-\varepsilon)^2$ | §4.1, App. D |
| $D_{\text{req}}^{\text{tok}}(\varepsilon, T, \beta)$ | Bits/token | Required per-token information for token-level schemes | §4 |
| $D_{\text{req}}^{\text{sem}}(\varepsilon, T_s, \beta)$ | Bits/sentence | Required per-sentence information for semantic schemes | §4 |
| $M, \tau$ | Scalar, $[0,1]$ | Outsider pooled tokens $M$ and TV budget $\tau$ | §4.1, App. D |
| $M_s, \tau_s$ | Scalar, $[0,1]$ | Outsider pooled sentences $M_s$ and TV budget $\tau_s$ | §4 |
| $D_{\text{stealth}}(M, \tau)$ | Bits/token | Stealth cap $\frac{2\tau^2}{M \ln 2}$ | §4.1, App. D |
| $D_{\text{stealth}}^{\text{tok}}(M, \tau)$ | Bits/token | Token-level stealth cap | §4 |
| $D_{\text{stealth}}^{\text{sem}}(M_s, \tau_s)$ | Bits/sentence | Semantic stealth cap | §4 |

## Optimization and Operators

The final table collects the optimization variables that appear in the design rule, along with the standard mathematical operators and asymptotic symbols used throughout.

| Symbol | Type/Dim | Description | Sections |
|---|---|---|---|
| $\mathcal{L}(\theta; \hat{\varepsilon}, M, \tau)$ | Scalar | Composite loss | §4.1 |
| $\theta$ | Variable | Scheme parameters | §4 |
| $\lambda_r, \lambda_q, \lambda_a$ | Scalars | Weights for reliability, stealth penalty, amplitude | §4.1 |
| $D^\star$ | Bits/token | Target per-token information after constraints | §4.1, App. E |
| $D_{\mathrm{BF}}^{\max}, D_{\mathrm{B}}^{\max}$ | Bits/token | Available budgets for BF and B families | §4.1, App. E |
| $\mathrm{TV}_{\mathrm{pen}}(D_0; M)$ | Scalar | Monotone detectability penalty used in the loss | §4.1 |
| $\mathrm{Amp}(\theta)$ | Scalar | Amplitude regularizer (e.g., $\sqrt{\hat{\sigma}^2}$ or $|\delta|$) | §4.1 |
| $\mathbb{E}[\cdot], \mathrm{Var}[\cdot]$ | Operator | Expectation, variance | §3 |
| $\mathbf{1}[\cdot]$ | Function | Indicator | §3 |
| $\arg\max, \sup$ | Operator | Maximizer, supremum | §3 |
| $\ln, \log, \log_2$ | Function | Natural log, log, base-2 log | §3 |
| $O(\cdot), o(\cdot), \Theta(\cdot), \omega(\cdot), \Omega(\cdot)$ | Notation | Asymptotic notation | §3 |
| $\approx$ | Operator | Approximately equal | App. D |
| $\infty$ | Symbol | Infinity | App. E |

## C. Proof of Theorem 3.2

This appendix proves the detectability bounds shown in Table 2 of Theorem 3.2. Each subsection treats one sampling family and bounds the total variation distance between the baseline distribution $P^s$ and the watermarked distribution $Q$. The two recurring tools are the chain rule, which expands the KL divergence between two autoregressive distributions as a sum of per-step KL divergences, and Pinsker's inequality, which converts that KL budget into a bound on total variation.

### C.1. Preliminaries

The total variation distance and the KL divergence between distributions $P$ and $Q$ are $\mathrm{TV}(P,Q) = \frac{1}{2} \sum_{\mathbf{y}} |P(\mathbf{y}) - Q(\mathbf{y})|$ and $\mathrm{KL}(Q\|P) = \mathbb{E}_Q[\log(Q(\mathbf{y})/P(\mathbf{y}))]$. Pinsker's inequality states $\mathrm{TV}(P,Q) \leq \sqrt{\frac{1}{2}\mathrm{KL}(Q\|P)}$. For autoregressive distributions factorizing as $Q(\mathbf{y}) = \prod_{t=1}^{T} q_t(y_t \mid y_{<t})$, the KL divergence decomposes step by step via the chain rule:

$$\mathrm{KL}(Q\|P) = \sum_{t=1}^{T} \mathbb{E}_{y_{<t} \sim Q}[\mathrm{KL}(q_t(\cdot \mid y_{<t})\|p_t(\cdot \mid y_{<t}))]. \tag{8}$$

The identity applies to token-level sequences of length $T$ and to sentence-level sequences of length $T_s$. Throughout, $\log$ denotes the natural logarithm, and conversion to bits uses $\mathrm{KL}_2(\cdot\|\cdot) = \mathrm{KL}(\cdot\|\cdot)/\ln 2$. When the main paper reports total variation numerically, the value is the Pinsker upper bound computed from a measured KL divergence rather than a direct estimate over the full text distribution.

### C.2. Greedy Sampling

Let $Q^{\mathrm{greedy}}$ place unit mass on the greedy path $\mathbf{y}^\star$, defined token by token through $y_t^\star = \arg\max_v p_t(v \mid y_{<t}^\star)$. Since $Q^{\mathrm{greedy}}(\mathbf{y}^\star) = 1$ and $Q^{\mathrm{greedy}}(\mathbf{y}) = 0$ otherwise, direct computation yields $\mathrm{TV}(P^s, Q^{\mathrm{greedy}}) = \frac{1}{2}(1 - P^s(\mathbf{y}^\star) + 1 - P^s(\mathbf{y}^\star)) = 1 - P^s(\mathbf{y}^\star) = O(1)$.

### C.3. Biased Sampling

The biased family shifts probability mass toward a keyed green set at every step. At position $t$, let $G_t \subseteq \mathcal{V}$ denote the green set and write $g_t := p_t(G_t)$ for its baseline mass. The biased sampler replaces the conditional distribution with $q_t(v) = p_t(v) \exp\{\delta\mathbf{1}[v \in G_t]\}/Z_t$, where $Z_t = (1 - g_t) + g_t e^\delta$. This is the distribution obtained by adding $\delta$ to the green-token logits before the softmax. The one-step KL divergence has the closed form

$$\mathrm{KL}(q_t\|p_t) = \delta\, q_t(G_t) - \log Z_t = \frac{g_t e^\delta}{(1 - g_t) + g_t e^\delta} \delta - \log\left((1 - g_t) + g_t e^\delta\right). \tag{9}$$

In practice, $|\delta| \ll 1$ since larger tilts degrade text quality. Expanding to second order in $\delta$ gives $Z_t = 1 + g_t(\delta + \delta^2/2) + O(\delta^3)$, $\log Z_t = g_t\delta + g_t(1 - g_t)\delta^2/2 + O(\delta^3)$, and $q_t(G_t) = g_t + g_t(1 - g_t)\delta + O(\delta^2)$. The linear terms cancel and only the quadratic term survives:

$$\mathrm{KL}(q_t\|p_t) = \frac{g_t(1 - g_t)}{2}\delta^2 + O(\delta^3), \tag{10}$$

where $g_t(1 - g_t)$ is the variance of the indicator $\mathbf{1}[v \in G_t]$ under the baseline distribution. Applying the chain rule (8) and Pinsker's inequality bounds the total variation by $\mathrm{TV}(P^s, Q^{\mathrm{bias}_\delta}) \leq |\delta|\sqrt{\frac{1}{4}\sum_{t=1}^{T} \mathbb{E}[g_t(1 - g_t)]} = O(|\delta|\sqrt{T})$.

## C.4. Bias-free Sampling

A bias-free watermark perturbs the per-step token distribution while keeping its average over keys unchanged. The scheme applies a keyed reweighting operator $R_E$ producing $q_{t,E}(\cdot \mid y_{<t}) := R_E(p_t(\cdot \mid y_{<t}))$, with induced sequence distribution $Q_E^{\mathrm{bf}}(y_{1:T}) = \prod_{t=1}^{T} q_{t,E}(y_t \mid y_{<t})$. The unbiasedness requirement $\mathbb{E}_E[R_E(p)] = p$ (expectation over the secret key $E$) ensures that any test that averages over keys and only inspects the marginal token distribution sees the baseline $p$ and so cannot detect the watermark. Expanding the pointwise perturbation as $q_{t,E}(v) = p_t(v) + \epsilon_{t,E}(v)$, the perturbation satisfies $\sum_v \epsilon_{t,E}(v) = 0$ because both $q_{t,E}$ and $p_t$ are valid probability distributions on the vocabulary. In the small-signal regime $|\epsilon_{t,E}(v)| \ll p_t(v)$, expanding $\mathrm{KL}(q_{t,E}\|p_t)$ to second order in $\epsilon_{t,E}$ around the unperturbed point yields

$$\mathrm{KL}(q_{t,E}\|p_t) = \frac{1}{2}\sum_v \frac{\epsilon_{t,E}(v)^2}{p_t(v)} + O(\|\epsilon_{t,E}\|_\infty^3), \tag{11}$$

whose leading term is the chi-square divergence between $q_{t,E}$ and $p_t$. Applying the chain rule and Pinsker's inequality gives the fixed-key bound $\mathrm{TV}(P^s, Q_E^{\mathrm{bf}}) \leq \sqrt{\frac{1}{4}\sum_{t=1}^{T} \mathbb{E}[\sum_v \epsilon_{t,E}(v)^2/p_t(v)]}$.

**Typical-key surrogate.** A keyless adversary does not know $E$ and observes a single watermarked output, so the operationally meaningful bound is the one obtained by averaging the fixed-key result over $E$. Unbiasedness makes the perturbation mean-zero, $\mathbb{E}_E[\epsilon_{t,E}(v)] = 0$, so its second moment is exactly the variance, $\mathbb{E}_E[\epsilon_{t,E}(v)^2] = \mathrm{Var}_E[R_E(p_t)(v)]$. Taking the expectation of equation (11) over $E$ gives $\mathbb{E}_E[\mathrm{KL}(q_{t,E}\|p_t)] \approx \frac{1}{2}\sum_v \mathrm{Var}_E[R_E(p_t)(v)]/p_t(v)$, and hence

$$\mathbb{E}_E[\mathrm{TV}(P^s, Q_E^{\mathrm{bf}})^2] \leq \frac{1}{4}\sum_{t=1}^{T} \mathbb{E}\Big[\sum_v \frac{\mathrm{Var}_E[R_E(p_t)(v)]}{p_t(v)}\Big]. \tag{12}$$

Applying Markov's inequality to the random variable $\mathrm{TV}(P^s, Q_E^{\mathrm{bf}})^2$ converts this bound on the average over $E$ into a bound that holds for most keys: for any $\eta \in (0, 1)$, the inequality $\mathrm{TV}(P^s, Q_E^{\mathrm{bf}}) \leq \eta^{-1/2}\sqrt{\frac{1}{4}\sum_{t,v} \mathrm{Var}_E[R_E(p_t)(v)]/p_t(v)}$ holds for a $(1 - \eta)$ fraction of keys. Both the expected bound and the typical-key bound scale as $O(\sqrt{T})$.

## C.5. Distribution-Preserving Sampling

A distribution-preserving scheme replaces the random source used during generation with a keyed pseudorandom one while keeping every conditional probability the same, so $q_t \equiv p_t$ for every history. The watermarked sequence distribution then factorizes identically to the baseline, $Q^{\mathrm{prf}}(\mathbf{y}) = \prod_{t=1}^{T} p_t(y_t \mid y_{<t}) = P^s(\mathbf{y})$, giving $\mathrm{TV}(P^s, Q^{\mathrm{prf}}) = 0$. Detection relies on the cryptographic key linking specific output positions to a secret seed.

## C.6. Semantic Rejection Sampling

Semantic schemes operate at the sentence level rather than the token level, which lets them survive paraphrases that change individual tokens while preserving meaning. Write $\mathbf{s}_{1:T_s} = (s_1, \ldots, s_{T_s})$ for a sequence of $T_s$ sentences, each $s_t \in \Sigma^\star$. At step $t$, define a keyed accept set $A_{t,k}(\mathbf{s}_{<t}) \subseteq \Sigma^\star$ with acceptance mass $a_t := p_t(A_{t,k}(\mathbf{s}_{<t}) \mid \mathbf{s}_{<t}) \in (0, 1]$. Rejection sampling draws candidates from $p_t$ until one falls in $A_{t,k}$, yielding $q_{t,k}(s \mid \mathbf{s}_{<t}) = p_t(s \mid \mathbf{s}_{<t})\mathbf{1}\{s \in A_{t,k}\}/a_t$.

For $s \in A_{t,k}$ the likelihood ratio equals $1/a_t$, so $\mathrm{KL}(q_{t,k}\|p_t) = \sum_{s \in A_{t,k}}(p_t(s)/a_t)\log(1/a_t) = \log(1/a_t)$, since $\sum_{s \in A_{t,k}} p_t(s) = a_t$. The same $a_t$ controls sampling cost, as rejection sampling needs an expected $1/a_t$ draws, so practical schemes keep $a_t$ bounded away from zero. Applying the chain rule at the sentence level and Pinsker's inequality gives:

$$\mathrm{TV}(P^s, Q_k^{\mathrm{sem}}) \leq \sqrt{\frac{1}{2}\sum_{t=1}^{T_s} \mathbb{E}[\log(1/a_t)]}. \tag{13}$$

Under uniform acceptance $a_t \geq a_{\min} > 0$, this yields $\mathrm{TV}(P^s, Q_k^{\mathrm{sem}}) \leq \sqrt{(T_s/2)\log(1/a_{\min})}$. Converting from sentence count to token count via $T \approx \ell T_s$, where $\ell$ is the typical sentence length, gives the $O(\sqrt{T/\ell})$ scaling.

*Remark* C.1 (Random quantities). The quantities $g_t$, $\text{Var}_E[R_E(p_t)(v)]$, and $a_t$ depend on random histories. They appear inside expectations throughout, so the final bounds are deterministic functions of the prompt, length, and watermark parameters.

### C.7. Information-theoretic versus Computational Detectability

The bounds above control the information-theoretic detectability $\text{Detect}_{\text{IT}}$ corresponding to an unbounded distinguisher. A separate question is whether a computationally bounded adversary can match those bounds in polynomial time. We organize the relationship between the two notions by the three detector access models defined in Section 3, starting from the fact that no efficient distinguisher can do better than an unbounded one.

**Lemma C.2** (Computational detectability is upper bounded by TV). *For every pair of distributions $(P_\lambda, Q_\lambda)$ over $\Omega$,* $\sup_{D_\lambda \in \text{PPT}} |\Pr_{Q_\lambda}[D_\lambda(y) = 1] - \Pr_{P_\lambda}[D_\lambda(y) = 1]| \leq \text{TV}(P_\lambda, Q_\lambda).$

*Proof.* For any randomized PPT detector $D_\lambda$ with internal randomness $R$, let $A_r := \{y : D_{\lambda,r}(y) = 1\}$ denote the acceptance set when $R = r$. For each fixed $r$, $|Q_\lambda(A_r) - P_\lambda(A_r)| \leq \text{TV}(P_\lambda, Q_\lambda)$. Averaging over $R$ and applying Jensen's inequality yields the claim. $\square$

**Keyless versus key-holding distinguishers.** The strength of the optimal unbounded distinguisher depends on whether it knows the secret key. Consider a keyed sampler with key space $\mathcal{K}_\lambda$, where $\lambda$ is the security parameter, and let $Q_{\lambda,k}$ denote the watermarked distribution induced by key $k$. The distinguishing advantage is the largest probability gap between accepting a watermarked output and accepting a baseline output. A *key-holding distinguisher*, such as the verifier, knows $k$ and tests $P_\lambda$ against $Q_{\lambda,k}$, attaining the optimal advantage $\text{TV}(P_\lambda, Q_{\lambda,k})$. A *keyless distinguisher* does not know $k$, so under the watermarked hypothesis its observation is drawn from the key-averaged mixture $\overline{Q}_\lambda := \mathbb{E}_k[Q_{\lambda,k}]$, and its optimal advantage $\text{TV}(P_\lambda, \overline{Q}_\lambda)$ can be much smaller than the fixed-key advantage $\text{TV}(P_\lambda, Q_{\lambda,k})$. The gap is largest for the expectation-preserving (bias-free) family of Section C.4, where the unbiasedness condition $\mathbb{E}_E[R_E(p)] = p$ forces $\overline{Q}_\lambda = P_\lambda$ and so the keyless advantage is exactly zero, even though $Q_{\lambda,k} \neq P_\lambda$ for every fixed $k$.

**Oracle and surrogate detectors: attainability of the Neyman-Pearson bound.** Under oracle access the detector can evaluate the true conditional probabilities $p_t(\cdot \mid x, y_{<t})$ exactly, knows the watermark rule, and (when acting as verifier) also knows the key. In this setting, the Neyman-Pearson lemma identifies the optimal test as a threshold on the likelihood ratio $L_{\lambda,k}(y) := Q_{\lambda,k}(y)/P_\lambda(y)$, namely $D^\star_{\lambda,k}(y) := \mathbf{1}\{L_{\lambda,k}(y) \geq 1\}$. For non-cryptographic watermarks (greedy, biased, bias-free, and semantic), this likelihood ratio is computable in polynomial time, so the optimal test runs efficiently and the computational distinguishing advantage matches the information-theoretic TV bound.

For greedy sampling and for the token-level biased and bias-free samplers, the autoregressive factorization decomposes the log-likelihood ratio across positions, $\log L_{\lambda,k}(y_{1:T}) = \sum_{t=1}^T \log(q_{t,k}(y_t \mid y_{<t})/p_t(y_t \mid y_{<t}))$, and each per-step term is efficient to compute given $p_t$ and the key. Under surrogate access the detector replaces $p_t$ with an independently trained model $\hat{p}_t$, applies the same procedure with surrogate probabilities, and recovers power close to the Neyman-Pearson bound whenever $\hat{p}_t$ approximates $p_t$ accurately enough.

Semantic rejection-sampling schemes are the exception. Their per-step likelihood ratio involves the acceptance mass $a_t = p_t(A_{t,k} \mid \mathbf{s}_{<t})$, and when $A_{t,k}$ is defined through learned sentence embeddings and a complex keyed predicate, $a_t$ may not be efficiently computable. The optimal likelihood-ratio test is then not implementable in polynomial time, and the resulting gap between information-theoretic and PPT-based detection is non-cryptographic in origin and persists even under oracle or surrogate access.

**Sample-only detectors: upper bounds without attainability guarantees.** Under sample-only access, the detector observes only text samples and cannot evaluate token-level probabilities. The Pinsker-based bounds derived above still hold as upper bounds on total variation. However, no general attainability claim can be made, because the Neyman-Pearson test cannot be implemented without probability access, and the achievable distinguishing advantage may be strictly smaller than the information-theoretic bound. This limitation applies to all watermarking families.

**Summary by access model.** The relationship between computational and information-theoretic detectability depends on detector access:

- **Oracle or surrogate access with key (verifier setting):** For token-level biased and bias-free watermarks, the likelihood ratio is computable in polynomial time, so $\text{Detect}_{\text{comp}} = \text{Detect}_{\text{IT}}$.
- **Oracle or surrogate access without key (keyless outsider):** Keyless PPT detectors can exploit key-independent artifacts such as frequency shifts for biased schemes and variance anomalies for bias-free schemes, consistent with empirical black-box detection results, although the attainable advantage may be smaller than the fixed-key TV.
- **Sample-only access:** The detector observes only watermarked text and cannot evaluate token-level probabilities, so the Neyman-Pearson likelihood-ratio test is not implementable. The Pinsker-based bounds still hold as upper bounds, but the achievable advantage may be strictly smaller than $\text{Detect}_{\text{IT}}$, and no general attainability claim holds.

**Cryptographic separations.** Cryptographic pseudorandomness can produce distributions that are far apart in total variation yet computationally indistinguishable for keyless PPT adversaries. For distribution-preserving PRF-seeded watermarks, the ideal target is $Q_{\lambda,k} = P_\lambda$, which gives $\text{TV} = 0$ and hence zero detectability even in the information-theoretic sense.

## D. Proof of Theorem 3.4

This appendix derives Theorem 3.4, which has to cover four watermark families that look different on the surface, namely the biased and bias-free token-level schemes on one side and the semantic or sentence-level schemes PMark and SemStamp on the other. A unified treatment becomes possible once we reduce each family to the same hypothesis-testing question, namely, how much of the watermark signal survives after an attacker edits the text. We follow the same logarithm convention as in Appendix C, namely that bare $\log$ denotes the natural logarithm while $\log_2$ is reserved for the base-2 logarithm.

In this proof, we first set up the post-attack distributions under $H_0$ (unwatermarked) and $H_1$ (watermarked). We then compute the per-unit KL information at zero attack, which gives a clean baseline against which everything else will be compared. We show next how an edit attack contracts this per-unit information, which is the step that connects watermarking to robustness. Finally, we aggregate the per-unit information across the sequence and ask when the accumulated total exceeds the threshold needed for reliable detection at the target miss probability. The same technique applies to every family considered below, which makes unified treatment possible.

### D.1. Edit Channel Model

We model our edit channel via random word substitution. Let $\Sigma$ denote the vocabulary. At each token position $t$, the editor leaves the original token alone with probability $1 - \varepsilon$ and replaces it with probability $\varepsilon$ by an independent draw from a replacement distribution $R$. In symbols, $\tilde{Y}_t = Y_t$ with probability $1 - \varepsilon$ and $\tilde{Y}_t = U_t \sim R(\cdot)$ with probability $\varepsilon$, where $R$ is independent of everything else. For any distribution $P$ on $\Sigma$, the mixture edit channel therefore acts as

$$T_{\varepsilon,R}(P) := (1 - \varepsilon)P + \varepsilon R. \tag{14}$$

We assume $R$ has full support with $\min_v R(v) \geq r_{\min} > 0$, so that every token retains a strictly positive probability under the post-edit distribution. The pre-noise conditionals $p_t$ (baseline) and $q_t$ (watermarked) are then mapped to the post-edit distributions $p_{t,\varepsilon} = T_{\varepsilon,R}(p_t)$ and $q_{t,\varepsilon} = T_{\varepsilon,R}(q_t)$ that the detector observes.

### D.2. Semantic-level Abstraction

Semantic watermarks behave differently from token-level ones because the evidence used for detection is computed at the sentence level even though the attacker edits at the token level. It is helpful to picture each sentence as carrying a one-bit fingerprint, where the attacker can either preserve that fingerprint or flip it. Fix a segmentation into $T_s$ sentences $\mathbf{S}_{1:T_s}$. A semantic watermark defines a keyed evidence function $Z_t := g_k(F(S_t)) \in \{0, 1\}$, where $F$ is a semantic feature map and $g_k$ is a keyed predicate. Balanced schemes satisfy $\mathbb{E}[Z_t \mid H_0] \approx \frac{1}{2}$ under the baseline, while watermarking biases the bit toward one outcome and produces $\mathbb{E}[Z_t \mid H_1] = \frac{1}{2} + \rho$ with $|\rho| \ll 1$.

Let $\widetilde{S}_t$ denote the sentence after token-level edits and $\widetilde{Z}_t := g_k(F(\widetilde{S}_t))$ the corresponding post-edit indicator. The semantic flip probability $\varepsilon_s(\varepsilon) := \Pr[\widetilde{Z}_t \neq Z_t]$ measures how often token edits actually corrupt the sentence-level fingerprint, and it depends both on the attack type and on how stable the semantic feature map $F$ is. We model the flips as a binary symmetric channel, so that $\widetilde{Z}_t = Z_t \oplus N_t$ with $N_t \sim \text{Bern}(\varepsilon_s(\varepsilon))$ and the noise variables $N_t$ are independent across $t$.

### D.3. Preliminaries: KL Expansions and Reliability Bound

Before working with the edit channel, we collect two facts that will do most of the heavy lifting in the rest of the appendix. The first says that for two distributions that are close to each other, the KL divergence reduces to a familiar weighted sum-of-squares form. The second says that a sufficiently large KL budget converts directly into detection power.

**Lemma D.1** (Second-order KL expansion). *Let $p$ be a distribution on a finite set and $q = p + r$ with $\sum_v r(v) = 0$ and $|r(v)| \leq \eta\, p(v)$ for $\eta \ll 1$. Then*

$$D(q\|p) = \frac{1}{2\ln 2} \sum_v \frac{r(v)^2}{p(v)} \cdot (1 + O(\eta)). \tag{15}$$

Read informally, the lemma says that when $q$ is a small perturbation of $p$, the KL divergence behaves like a chi-squared distance and scales with the square of the perturbation. This is the source of every quadratic appearance as follows.

*Proof.* We Taylor expand $\log(1 + x) = x - x^2/2 + O(x^3)$ around $x = 0$ with $x_v = r(v)/p(v)$. Because $\sum_v r(v) = 0$ by assumption, the linear-order terms cancel when we sum over $v$, and only the quadratic terms survive. Multiplying by the base-2 conversion factor produces the stated expression, and the remainder is $O(\eta)$ times the quadratic term. $\square$

**Lemma D.2** (Stein's sufficient condition). *For a binary hypothesis test between product distributions on sequences of length L, if the total KL divergence satisfies $\sum_{t=1}^{L} D(P_t^{(1)} \| P_t^{(0)}) \geq \log_2(1/\beta) + o(L)$, then for sufficiently large L the Neyman-Pearson test at level $\alpha$ achieves miss probability at most $\beta$.*

In other words, as long as the total KL information collected across the sequence exceeds $\log_2(1/\beta)$, the optimal test can be tuned so that its miss probability is at most $\beta$. This lemma is the bridge between information budgets and detection power, and it later allows us to convert the contraction results below into edit-rate thresholds.

*Proof.* The Neyman-Pearson lemma identifies the optimal test as a threshold rule on the per-sequence log-likelihood ratio

$$S_L = \sum_{t=1}^{L} \log_2 \frac{P_t^{(1)}(Y_t)}{P_t^{(0)}(Y_t)},$$

where we reject $H_0$ as soon as $S_L$ exceeds a threshold $\tau_L$ that we still have to choose. The proof has two parts, one for each error type, namely the false-alarm rate under $H_0$, which must be at most $\alpha$, and the miss probability under $H_1$, which we want to bound by $\beta$.

**Controlling the level.** Under $H_0$, the random variable $2^{S_L}$ is the per-sequence likelihood ratio, and a direct computation shows that its expectation is exactly one, because the product structure gives

$$\mathbb{E}_0\big[2^{S_L}\big] = \prod_{t=1}^{L} \sum_y P_t^{(0)}(y) \cdot \frac{P_t^{(1)}(y)}{P_t^{(0)}(y)} = \prod_{t=1}^{L} \sum_y P_t^{(1)}(y) = 1.$$

Applying Markov's inequality to $2^{S_L}$ then yields

$$\Pr_0(S_L \geq \tau_L) = \Pr_0\big(2^{S_L} \geq 2^{\tau_L}\big) \leq \mathbb{E}_0\big[2^{S_L}\big] \cdot 2^{-\tau_L} = 2^{-\tau_L},$$

so the choice $\tau_L = \log_2(1/\alpha)$ pins the false-alarm probability at $\alpha$ and certifies the level of the test.

**Controlling the miss probability.** The miss probability is the chance that the log-likelihood ratio falls short of the threshold under $H_1$, namely $\Pr_1(S_L \leq \tau_L)$. We bound this with a Chernoff argument tailored to product distributions. Define the per-position cumulant-generating function

$$\psi_t(s) := -\log_2 \sum_y P_t^{(1)}(y)^{1-s} P_t^{(0)}(y)^s,$$

which encodes the moments of the per-position log-likelihood ratio. Direct differentiation gives $\psi_t(0) = 0$ and $\psi_t'(0) = D(P_t^{(1)} \| P_t^{(0)})$, so the derivative of $\psi_t$ at zero is exactly the per-position KL divergence. The standard Chernoff bound on a

sum of independent quantities then turns the small-deviation event $\{S_L \leq \tau_L\}$ under $H_1$ into an exponential rate driven by the total KL divergence, yielding

$$\Pr_1(S_L \leq \tau_L) \leq 2^{-\left(\sum_t D(P_t^{(1)} \| P_t^{(0)}) - \log_2(1/\alpha) - o(L)\right)}.$$

Substituting the hypothesis $\sum_t D(P_t^{(1)} \| P_t^{(0)}) \geq \log_2(1/\beta) + o(L)$ pushes the exponent above $\log_2(1/\beta) - \log_2(1/\alpha)$, which forces the miss probability to be at most $\beta$ for sufficiently large $L$ as claimed. $\square$

### D.4. Per-unit Information at $\varepsilon = 0$

We now compute the per-unit KL divergence $D_0$ before any edits are applied. This baseline quantity is what every subsequent subsection will track as the attacker erodes it, and the fact that all watermark families admit the same $D_0$-shaped expression makes the unified theorem possible.

**Token-level.** Consider biased sampling first. At each position, the biased sampler picks a keyed subset of tokens $G \subseteq \Sigma$, sometimes called the green set, and tilts probability mass toward this subset by multiplying the unnormalized probability of every $v \in G$ by an exponential factor $e^\delta$. The tilted conditional is therefore $q_{t,\delta}(v) \propto p_t(v)e^{\delta \mathbf{1}[v \in G]}$, and we let $\gamma = p_t(G)$ denote the baseline mass that the model originally placed on $G$. Expanding the KL divergence $D(q_{t,\delta} \| p_t)$ to second order in the small tilt $\delta$ gives

$$D(q_{t,\delta} \| p_t) = \delta^2 \gamma(1 - \gamma)/(2 \ln 2) + O(\delta^3).$$

The factor $\gamma(1 - \gamma)$ is the variance of the indicator $\mathbf{1}[v \in G]$ under the baseline distribution, so the per-token information is largest when the green set carries roughly half the baseline mass and shrinks toward zero whenever $\gamma$ approaches either extreme.

The bias-free case proceeds along the same lines but with a different perturbation. The watermark applies a key-dependent reweighting to each conditional, writing $q_{t,E}(v) = p_t(v)(1 + \Delta_E(v))$ together with the unbiasedness condition $\mathbb{E}_E[\Delta_E(v)] = 0$. Unbiasedness has a concrete meaning: an outsider who averages over the random key sees exactly the baseline distribution and thus cannot detect the watermark from the marginals alone. When we average the per-step KL over the key, the linear-in-$\Delta_E$ terms drop out for the same reason, and what remains at second order is

$$\mathbb{E}_E[D(q_{t,E} \| p_t)] = \hat{\sigma}^2/(2 \ln 2) + O(\|\Delta_E\|_\infty^3),$$

where $\hat{\sigma}^2 = \sum_v p_t(v)\mathrm{Var}_E[\Delta_E(v)]$ is the variance of the keyed reweighting averaged across the vocabulary under the baseline distribution. The intuition is the same as in the biased case, since the per-token information is again driven by the variance of the watermark perturbation rather than by its mean.

**Semantic.** For semantic watermarks the per-unit object is a Bernoulli rather than a categorical distribution, because the sentence-level evidence is a single bit $Z_t \in \{0, 1\}$ rather than a token drawn from the full vocabulary. Under $H_0$ the bit is unbiased and $Z \sim \mathrm{Bern}(\frac{1}{2})$, while the watermark biases it toward one outcome by a small amount $\rho$ to give $Z \sim \mathrm{Bern}(\frac{1}{2} + \rho)$ under $H_1$. The KL divergence between these two Bernoullis is the standard binary KL expression

$$D_0^{(\mathrm{sem})} = (\tfrac{1}{2} + \rho) \log_2(1 + 2\rho) + (\tfrac{1}{2} - \rho) \log_2(1 - 2\rho) = \frac{2\rho^2}{\ln 2} + O(\rho^4). \tag{16}$$

The quadratic approximation on the right follows by Taylor expanding $\log_2(1 \pm 2\rho)$ to second order in $\rho$, and the odd-order terms cancel because the Bernoulli is symmetric around the unbiased baseline $\frac{1}{2}$. The $\rho^2$ scaling here mirrors the $\delta^2$ scaling for biased sampling and the $\hat{\sigma}^2$ scaling for bias-free sampling, and all three reflect the same underlying fact, namely that a small perturbation contributes detection information only at second order in its amplitude.

### D.5. Edits Contract the Signal Quadratically

This subsection establishes the headline contraction property that drives every subsequent result. The reader should remember the takeaway as follows: when an editor randomizes a fraction $\varepsilon$ of the tokens, the KL information available to the detector does not scale linearly with $\varepsilon$. It is attenuated by a factor of $(1 - \varepsilon)^2$, which is much harsher than the linear loss one might naively expect.

**Lemma D.3** (Mixture-channel KL contraction). *Let $q = p + r$ with $|r(v)| \leq \eta p(v)$ for $\eta \ll 1$, and let $p_\varepsilon = T_{\varepsilon,R}(p)$, $q_\varepsilon = T_{\varepsilon,R}(q)$. Then*

$$D(q_\varepsilon \| p_\varepsilon) = (1 + o(1))(1 - \varepsilon)^2 D(q \| p). \tag{17}$$

*Proof.* The first step is to compute how much of the original perturbation between $q$ and $p$ remains visible after the mixture edit channel has acted on both distributions. Write the perturbation as $r = q - p$, so that $r$ encodes the displacement between the watermarked and baseline distributions and is the quantity whose magnitude controls $D(q\|p)$. The mixture channel $T_{\varepsilon,R}$ is *affine* in its input, meaning that it depends linearly on its argument together with a fixed offset that does not depend on the argument, namely $T_{\varepsilon,R}(P) = (1 - \varepsilon)P + \varepsilon R$. Affinity is useful here because the fixed offset $\varepsilon R$ cancels when we subtract the two post-edit distributions, leaving only the scaled linear part:

$$\widetilde{r} := q_\varepsilon - p_\varepsilon = \big((1 - \varepsilon)q + \varepsilon R\big) - \big((1 - \varepsilon)p + \varepsilon R\big) = (1 - \varepsilon)\,(q - p) = (1 - \varepsilon)\,r. \tag{18}$$

In words, editing shrinks the watermark perturbation by a factor of $(1 - \varepsilon)$, because the unchanged fraction $1 - \varepsilon$ of the tokens still carries the original signal while the replaced fraction $\varepsilon$ contributes the same noise distribution $R$ on both sides and washes out.

We now apply Lemma D.1 to the post-edit distributions $p_\varepsilon$ and $q_\varepsilon$, which gives

$$D(q_\varepsilon \| p_\varepsilon) = \frac{1}{2 \ln 2} \sum_v \frac{\widetilde{r}(v)^2}{p_\varepsilon(v)} \cdot \big(1 + O(\eta)\big). \tag{19}$$

The numerator squares the post-edit perturbation, so substituting $\widetilde{r}(v) = (1 - \varepsilon)\,r(v)$ from (18) pulls a factor of $(1 - \varepsilon)^2$ out of the sum and gives the headline scaling. For the denominator, we expand $p_\varepsilon(v) = (1 - \varepsilon)p(v) + \varepsilon R(v)$ around the baseline $p(v)$ to obtain

$$\frac{1}{p_\varepsilon(v)} = \frac{1}{(1 - \varepsilon)p(v) + \varepsilon R(v)} = \frac{1}{p(v)}\big(1 + O(\varepsilon)\big), \tag{20}$$

where the $O(\varepsilon)$ term collects all the corrections that come from the small mixing weight on the replacement distribution $R$. Plugging (27) back into (19) preserves the leading piece $(1 - \varepsilon)^2 \cdot (1/(2 \ln 2)) \sum_v r(v)^2/p(v)$ untouched, and the perturbative correction $(1 + O(\varepsilon))$ in the denominator combines with the $(1 + O(\eta))$ factor inherited from Lemma D.1 into a single multiplicative $(1 + o(1))$ term. Recognizing the leading sum as a second application of Lemma D.1, this time to the pre-edit pair $(q, p)$, identifies it with $D(q\|p)$ and gives the stated identity. $\square$

The same $(1 - \varepsilon)^2$ contraction holds for any mixture channel that linearly attenuates the perturbation, not just uniform substitution. For correlated paraphrasing attacks, empirical observations suggest a similar quadratic attenuation once $\varepsilon$ is calibrated to the observed token edit rate, which justifies using this simple model in the rest of the appendix even when the underlying attack is more structured.

**Lemma D.4** (Semantic evidence contraction). *Under the BSC model $\widetilde{Z} = Z \oplus N$ with $N \sim \mathrm{Bern}(\varepsilon_s)$:*

(i) *Under $H_0$, $\widetilde{Z} \sim \mathrm{Bern}(\frac{1}{2})$ (unbiased bits are invariant under symmetric flips).*
(ii) *Under $H_1$, $\widetilde{Z} \sim \mathrm{Bern}(\frac{1}{2} + \rho_\varepsilon)$ with $\rho_\varepsilon = (1 - 2\varepsilon_s)\rho$.*
(iii) *The post-attack KL satisfies $D_\varepsilon^{(\mathrm{sem})} = (1 - 2\varepsilon_s)^2 D_0^{(\mathrm{sem})} + O(\rho^4)$.*

The semantic analogue of the previous lemma carries the same moral, namely that random flips attenuate the watermark bias linearly and the KL divergence quadratically. The only difference is that the relevant flip rate is now $\varepsilon_s$ rather than $\varepsilon$ itself.

*Proof.* Part (i) follows by direct computation, since $\Pr(\widetilde{Z} = 1) = \frac{1}{2}(1 - \varepsilon_s) + \frac{1}{2}\varepsilon_s = \frac{1}{2}$, so an unbiased bit remains unbiased after symmetric flipping. For part (ii) we compute $\Pr(\widetilde{Z} = 1) = (\frac{1}{2} + \rho)(1 - \varepsilon_s) + (\frac{1}{2} - \rho)\varepsilon_s$, which simplifies to $\frac{1}{2} + \rho(1 - 2\varepsilon_s)$ and identifies the attenuated bias as $\rho_\varepsilon = (1 - 2\varepsilon_s)\rho$. Part (iii) is then immediate by applying (16) with $\rho$ replaced by $\rho_\varepsilon$. $\square$

## D.6. Sequence-level Aggregation

So far the analysis has been local, treating one token or one sentence at a time. We now stitch the per-unit budgets into a sequence-level budget by appealing to the KL chain rule. The chain rule states that the joint divergence decomposes as

$$D(Q_\varepsilon \| P_\varepsilon) = \sum_{t=1}^{T} \mathbb{E}_{\tilde{Y}_{<t} \sim Q_\varepsilon}\big[D(Q_\varepsilon(\tilde{Y}_t \mid \tilde{Y}_{<t}) \| P_\varepsilon(\tilde{Y}_t \mid \tilde{Y}_{<t}))\big].$$

In the small-signal regime, each conditional inherits the same quadratic attenuation that Lemma D.3 established for a single token, and summing over positions gives the token-level information budget

$$C_{\text{tok}}(\varepsilon) := D(Q_\varepsilon \| P_\varepsilon) \approx T(1-\varepsilon)^2 D_0 \quad \text{(bits)}. \tag{21}$$

The semantic case is even cleaner, because the per-sentence indicators are modeled as i.i.d. and the chain rule reduces to exact additivity:

$$C_{\text{sem}}(\varepsilon) := D(P^{(1)}_{\widetilde{Z}_{1:T_s}} \| P^{(0)}_{\widetilde{Z}_{1:T_s}}) = T_s D^{(\text{sem})}_\varepsilon \approx T_s(1 - 2\varepsilon_s(\varepsilon))^2 D^{(\text{sem})}_0. \tag{22}$$

The two right-hand sides are the post-edit information budgets that will be matched against the detection requirement in the next subsection.

## D.7. Power Condition and Knee Edit Rate

We now convert the aggregated information budget $C(\varepsilon)$ into a concrete edit rate. The conversion is mechanical once Lemma D.2 is in place, because the lemma tells us that reliable detection at miss probability $\beta$ requires $C(\varepsilon) \geq \log_2(1/\beta)$. The largest edit rate that still meets this requirement is what we call the *knee*, because the detection power drops off rapidly once $\varepsilon$ crosses this threshold.

**Token-level.** Setting $T(1-\varepsilon)^2 D_0 \geq \log_2(1/\beta)$ and solving for $\varepsilon$ gives

$$\varepsilon_\beta(T, D_0) = 1 - \sqrt{\frac{\log_2(1/\beta)}{TD_0}}. \tag{23}$$

**Semantic-level.** Setting $T_s(1 - 2\varepsilon_s)^2 D^{(\text{sem})}_0 \geq \log_2(1/\beta)$ and solving for $\varepsilon_s$ gives the analogous bound

$$\varepsilon_s(\varepsilon) \leq \frac{1}{2}\left(1 - \sqrt{\frac{\log_2(1/\beta)}{T_s D^{(\text{sem})}_0}}\right). \tag{24}$$

The factor of $\frac{1}{2}$ in front reflects the binary symmetric structure of the semantic flip channel, which can only be informative when fewer than half of the indicator bits are flipped.

*Proof.* The first step is to compute how much of the original perturbation between $q$ and $p$ remains visible after the mixture edit channel has acted on both distributions. Write the perturbation as $r = q - p$, so that $r$ encodes the displacement between the watermarked and baseline distributions and is the quantity whose magnitude controls $D(q\|p)$. The mixture channel $T_{\varepsilon,R}$ is *affine* in its input, meaning that it depends linearly on its argument together with a fixed offset that does not depend on the argument, namely $T_{\varepsilon,R}(P) = (1-\varepsilon)P + \varepsilon R$. Affinity is useful here because the fixed offset $\varepsilon R$ cancels when we subtract the two post-edit distributions, leaving only the scaled linear part:

$$\widetilde{r} := q_\varepsilon - p_\varepsilon = \big((1-\varepsilon)q + \varepsilon R\big) - \big((1-\varepsilon)p + \varepsilon R\big) = (1-\varepsilon)(q-p) = (1-\varepsilon)r. \tag{25}$$

In words, editing shrinks the watermark perturbation by a factor of $(1-\varepsilon)$, because the unchanged fraction $1-\varepsilon$ of the tokens still carries the original signal while the replaced fraction $\varepsilon$ contributes the same noise distribution $R$ on both sides and washes out.

We now apply Lemma D.1 to the post-edit distributions $p_\varepsilon$ and $q_\varepsilon$, which gives

$$D(q_\varepsilon \| p_\varepsilon) = \frac{1}{2\ln 2} \sum_v \frac{\widetilde{r}(v)^2}{p_\varepsilon(v)} \cdot \big(1 + O(\eta)\big). \tag{26}$$

The numerator squares the post-edit perturbation, so substituting $\widetilde{r}(v) = (1-\varepsilon)r(v)$ from (25) pulls a factor of $(1-\varepsilon)^2$ out of the sum and gives the headline scaling. For the denominator, we expand $p_\varepsilon(v) = (1-\varepsilon)p(v) + \varepsilon R(v)$ around the baseline $p(v)$ to obtain

$$\frac{1}{p_\varepsilon(v)} = \frac{1}{(1-\varepsilon)p(v) + \varepsilon R(v)} = \frac{1}{p(v)}\big(1 + O(\varepsilon)\big), \tag{27}$$

where the $O(\varepsilon)$ term collects all the corrections that come from the small mixing weight on the replacement distribution $R$. Plugging (27) back into (26) preserves the leading piece $(1-\varepsilon)^2 \cdot (1/(2\ln 2)) \sum_v r(v)^2/p(v)$ untouched, and the perturbative correction $(1 + O(\varepsilon))$ in the denominator combines with the $(1 + O(\eta))$ factor inherited from Lemma D.1 into a single multiplicative $(1 + o(1))$ term. Recognizing the leading sum as a second application of Lemma D.1, this time to the pre-edit pair $(q, p)$, identifies it with $D(q\|p)$ and gives the stated identity. $\qquad\square$

### D.8. Scope of Validity

It is worth pausing to record the regime in which each of the above statements is valid. The token-level results assume the small-signal regime, meaning $|\delta| \ll 1$ for biased sampling and $\|\Delta_E\|_\infty \ll 1$ for bias-free sampling, together with the requirement that the baseline probabilities $p_t(v)$ stay bounded away from zero so that the division by $p_t$ in Lemma D.1 remains well controlled. The same lemma quantifies the resulting approximation error as lower-order in the perturbation. The semantic results require $|\rho| \ll 1$ and rely on the binary symmetric channel abstraction, in which the effect of any attack is summarized through a single scalar $\varepsilon_s(\varepsilon)$. The mapping $\varepsilon \mapsto \varepsilon_s(\varepsilon)$ depends on the attack and is estimated empirically, which is exactly what we do in Section 5.

### D.9. Conclusion of Proof

Pulling the four ingredients together, namely the per-unit KL expressions, the quadratic attenuation under edits, the chain-rule aggregation, and Stein's sufficient condition, produces the two headline statements of Theorem 3.4:

$$\text{Token-level:} \quad T(1-\varepsilon)^2 D_0 \geq \log_2(1/\beta) \Rightarrow \varepsilon_\beta = 1 - \sqrt{\log_2(1/\beta)/(TD_0)}, \tag{28}$$

$$\text{Semantic:} \quad T_s(1-2\varepsilon_s)^2 D_0^{(\text{sem})} \geq \log_2(1/\beta) \Rightarrow \varepsilon_s \leq \tfrac{1}{2}(1 - \sqrt{\log_2(1/\beta)/(T_s D_0^{(\text{sem})})}). \tag{29}$$

$\qquad\square$

### Proof of Corollary 3.5

The corollary follows directly from the theorem, and we treat its two assertions in turn.

**Information-theoretic impossibility.** Theorem 3.4 requires the post-edit information budget to satisfy $T(1-\varepsilon)^2 D_0 \geq \log_2(1/\beta)$ before a level-$\alpha$ test can certify power $1 - \beta$. The right-hand side is fixed by the operating point of the detector, while the left-hand side shrinks quadratically in the edit rate. Once $\varepsilon$ crosses the knee $\varepsilon_\beta(T, D_0)$, the inequality is violated, the surviving KL information falls short of the threshold required by the lemma above, and no test can drive the miss probability below $\beta$. This is the impossibility region claimed by the corollary in the absence of a stealth requirement.

**Stealth-aware tightening.** The stealth-aware version of the bound adds the further constraint that the watermark must remain hard to detect for a keyless outsider who can pool many tokens together. Suppose such an outsider observes $M$ tokens and the watermark is held within total-variation distance $\tau$ of the baseline distribution. Pinsker's inequality, applied in the bit convention used throughout this appendix, then bounds the per-token KL information by $D_0 \leq \frac{2}{\ln 2} \cdot \frac{\tau^2}{M}$.

The stealth requirement, therefore, caps the watermark strength the designer can use, which in turn limits the robustness the theorem can certify. Substituting $D_0$ into the condition $T(1-\varepsilon)^2 D_0 \geq \log_2(1/\beta)$ and solving for $\varepsilon$ produces:

$$\varepsilon \leq 1 - \sqrt{\frac{\log_2(1/\beta)}{T} \cdot \frac{M\ln 2}{2\tau^2}}. \tag{30}$$

Any edit rate $\varepsilon$ that exceeds the right-hand side of (30) is infeasible whenever the stealth constraint is active. $\qquad\square$

### D.10. Information-theoretic vs. Computational Hardness

The theorem above is information-theoretic in nature, because it bounds the power of an unrestricted (potentially unbounded-time) detector. A natural follow-up question asks whether a polynomial-time detector can match this bound, or whether there is a statistical-computational gap. The next lemma is the easy direction of this question.

**Lemma D.5** (Computational power bounded by IT power). *For any $\lambda$, $\varepsilon$, and $\alpha$:* $\text{Power}_{\text{comp},\lambda}(\varepsilon, \alpha) \leq \text{Power}_{\text{IT},\lambda}(\varepsilon, \alpha)$.

In plain terms, restricting the detector to polynomial time cannot increase its power, because polynomial-time detectors form a subset of the detectors that the Neyman-Pearson test already optimizes over.

*Proof.* The Neyman-Pearson test maximizes power among all level-$\alpha$ tests. Since polynomial-time (PPT) detectors form a subset of all detectors, the supremum of power over PPT detectors cannot exceed the information-theoretic supremum. $\square$

The interesting question is when this inequality is tight, and the answer depends on the watermark family. We work through the cases one at a time.

**Equality for non-cryptographic families under oracle or surrogate access.** For biased and bias-free watermarks, the post-edit likelihood ratio decomposes additively across positions, so $\log_2(Q_\varepsilon(y_{1:T})/P_\varepsilon(y_{1:T})) = \sum_{t=1}^{T} \log_2(q_{t,\varepsilon}(y_t)/p_{t,\varepsilon}(y_t))$ is computable in $O(T)$ time once the detector has oracle or surrogate access to the model probabilities and the watermark parameters. The Neyman-Pearson test therefore runs in polynomial time and attains the information-theoretic boundary, so $\text{Power}_{\text{comp}} = \text{Power}_{\text{IT}}$ for these families.

**Sample-only access.** The situation changes if the detector only sees samples from the model and cannot evaluate token-level probabilities directly. In that case the Neyman-Pearson test is no longer implementable. The Pinsker-based upper bounds on total variation remain valid, but we make no attainability claim for sample-only detectors.

**Semantic schemes may exhibit a gap.** Semantic watermarks are harder to evaluate, because the likelihood ratio at each position depends on acceptance masses $a_t = p_t(A_{t,k} \mid \mathbf{s}_{<t})$ that require normalization over semantic regions. Such normalization can be computationally intractable, so semantic schemes may exhibit $\text{Power}_{\text{comp}} < \text{Power}_{\text{IT}}$ even when no cryptographic assumption is invoked.

**Cryptographic separations.** A clean separation appears once cryptography enters. Consider a pseudorandom generator $G : \{0,1\}^\lambda \to \{0,1\}^{T(\lambda)}$ together with $P_\lambda = U_{T(\lambda)}$ and $Q_\lambda = \text{Law}(G(U_\lambda))$. The sequence-level KL divergence is $D(Q_\lambda \| P_\lambda) = T(\lambda) - \lambda$, so after editing the budget $C_{\text{IT}}(\lambda, \varepsilon) \approx (1-\varepsilon)^2(T(\lambda) - \lambda)$ grows linearly in $T(\lambda)$, and an unrestricted detector achieves arbitrarily high power as $T(\lambda)$ grows.

However, any polynomial-time detector with non-negligible power would immediately yield a PRG distinguisher by the following reduction. Given a sample $z$ from the PRG game, the distinguisher draws $\tilde{z} \sim T_\varepsilon(\delta_z)$ and returns the detector's guess. When $z$ is uniform, $\tilde{z}$ is distributed as $P_{\lambda,\varepsilon}$, and when $z$ is pseudorandom, $\tilde{z}$ is distributed as $Q_{\lambda,\varepsilon}$. Non-negligible detection power would therefore contradict PRG security, which forces $\text{Power}_{\text{comp}} \leq \text{negl}(\lambda)$ even though $\text{Power}_{\text{IT}} \geq 1 - \beta$.

**Empirical support.** Recent black-box detectors (Gloaguen et al., 2025) achieve near-perfect AUROC whenever $T_{\text{tot}} D_0 \gtrsim \log_2(1/\beta)$, which matches our bounds quantitatively. Their methods exploit frequency shifts for biased schemes and variance anomalies for bias-free schemes without any internal model access, and this provides further evidence that no meaningful statistical-computational gap exists for non-cryptographic families under surrogate access.

### D.11. Relation to Coding-theoretic Bounds

The scaling $C(\varepsilon) \approx T(1-\varepsilon)^2 D_0$ should not be interpreted as a Shannon-style capacity formula for generic edit channels, even though the symbols look similar. Classical insertion-deletion codes can achieve a rate that degrades only linearly as $1 - O(\varepsilon)$, but they enjoy the freedom to choose any codeword in the input space. Our setting is more constrained on three fronts. All outputs must lie in the typical set of a fixed base model $P$, a stealth requirement bounds the per-token KL drift $D_0$, and the underlying task is binary hypothesis testing rather than message recovery. Theorem 3.4 translates the Chernoff-Stein criterion into an explicit robustness bound that is appropriate for the distribution-constrained watermarking setting and should be read as such, not as a coding-theoretic upper limit.

## E. Proof Details for Section 4

This appendix collects the derivations behind the selection rule in Section 4. Two ideas recur throughout. The first is a Chernoff-Stein-style reliability condition that yields a lower bound on the available KL divergence in terms of target detection power. The second is Pinsker's inequality, which converts a per-unit KL budget into a total variation cap that a keyless outsider can pool against. With these pieces in hand, we prove the minimal-information optimality theorem, walk through the family-selection rule that the main text uses, and close with the estimation protocol for $(\varepsilon, \varepsilon_s(\varepsilon))$ and a sensitivity analysis. Throughout, $\log$ denotes the natural logarithm and $\log_2$ denotes the base-2 logarithm, with KL divergences and information budgets stated in bits via $\log_2$. We reuse the TV and KL definitions, Pinsker's inequality, and the chain rule from the Preliminaries of Appendix C. The budgets $D_0^{(\text{tok})}$ and $D_0^{\text{S}}$ are the per-step and per-sentence KL drifts computed there.

### E.1. Reliability via KL Divergence

The selection rule in Section 4 rests on a single reliability principle. To reach a target miss probability $\beta$ at a fixed false-alarm level $\alpha$, the test requires sufficient cumulative evidence that the watermarked distribution differs from the baseline. If the sequence-level KL divergence under $H_1$ relative to $H_0$ exceeds $\log_2(1/\beta)$ up to lower-order terms, then there is a level-$\alpha$ Neyman-Pearson test whose miss probability is at most $\beta$.

**Lemma E.1** (KL sufficiency for miss probability $\beta$)**.** *Consider a binary hypothesis test between distributions* $(P, Q)$ *with false-alarm constraint* $\Pr_{Y \sim P}[D(Y) = 1] \leq \alpha$. *Suppose that the log-likelihood ratio decomposes as a sum of independent contributions with a finite moment-generating function near the origin. Then, for any fixed* $\alpha, \beta \in (0, 1)$ *and sufficiently large blocklengths, there exists a level-$\alpha$ test with miss probability at most $\beta$ whenever*

$$\sum_{t=1}^{L} D(P_t^{(1)} \| P_t^{(0)}) \; \geq \; \log_2 \frac{1}{\beta} + \log_2 \frac{1}{\alpha} + o(L). \tag{31}$$

*Proof.* This is the standard Chernoff-Stein sufficiency rule. Define the cumulative log-likelihood ratio

$$L_{1:L} \; = \; \sum_{t=1}^{L} \log_2 \frac{P_t^{(1)}(Y_t)}{P_t^{(0)}(Y_t)}. \tag{32}$$

By the Neyman-Pearson lemma (Cover & Thomas, 2006, Theorem 11.7.1), the best level-$\alpha$ test compares $L_{1:L}$ to a single threshold $\tau$ and rejects $P^{(0)}$ when $L_{1:L} \geq \tau$. So we only need to find a threshold that meets both error constraints at once.

The threshold $\tau$ acts as a slider between the two error types. Pushing $\tau$ up shrinks the type I error because $L_{1:L}$ rarely exceeds a high threshold under $P^{(0)}$, where its mean is negative. Pushing $\tau$ down shrinks the type II error because $L_{1:L}$ rarely falls below a low threshold under $P^{(1)}$, where its mean equals the cumulative KL divergence $\sum_t D(P_t^{(1)} \| P_t^{(0)})$.

The finite moment-generating function assumption gives us an exponential concentration bound on $L_{1:L}$ in both directions, which is Cramér's theorem in the form used by Csiszár & Körner (2011, Chapter 2). Setting $\tau = \log_2(1/\alpha)$ keeps the type I error at level $\alpha$ up to a correction of order $\sqrt{L}$. The same concentration bound under $P^{(1)}$ keeps the type II error below $\beta$ as long as the mean $\sum_t D(P_t^{(1)} \| P_t^{(0)})$ sits at least $\log_2(1/\beta) + o(L)$ above $\tau$. Adding the two requirements and using $\tau = \log_2(1/\alpha)$ produces the sufficient condition

$$\sum_{t=1}^{L} D(P_t^{(1)} \| P_t^{(0)}) \; \geq \; \log_2 \frac{1}{\beta} + \log_2 \frac{1}{\alpha} + o(L), \tag{33}$$

which is exactly (31). $\qquad \square$

In other words, holding the false-alarm level $\alpha$ fixed turns the condition into $D(Q \| P) \geq \log_2(1/\beta) + O(1)$, which is the leading-order threshold $\log_2(1/\beta)$ that appears throughout the main text. Each watermark family enters this picture only through the KL budget that survives after edits, which is what the next two subsections compute.

### E.2. Token-Level Channel Derivation

We now compute how much of a token-level watermark's KL budget actually survives edits, since this is the quantity that the reliability condition consumes. Let $Y_{1:T}$ denote the generated token sequence. Under the null hypothesis $H_0$, the baseline autoregressive distribution is $P^s(y_{1:T}) = \prod_{t=1}^{T} p_t(y_t \mid y_{<t})$. Under a token-level watermark with a fixed key, the pre-edit distribution is $Q(y_{1:T}) = \prod_{t=1}^{T} q_t(y_t \mid y_{<t})$.

The attacker is modeled as a token substitution channel at rate $\varepsilon$, which independently replaces each token with a draw from a fixed full-support distribution $R$ with probability $\varepsilon$ and leaves it unchanged with probability $1 - \varepsilon$. At the level of conditional distributions, this acts as $T_{\varepsilon, R}(P) := (1 - \varepsilon)P + \varepsilon R$. In the small-signal regime we write $q_t = p_t + r_t$ with $\sum_v r_t(v) = 0$, where $r_t$ is the perturbation that the watermark introduces. Linearity of $T_{\varepsilon, R}$ then gives $q_{t,\varepsilon} - p_{t,\varepsilon} = (1 - \varepsilon)r_t$, so the perturbation that survives editing is just the original perturbation rescaled by the survival probability $1 - \varepsilon$. Since KL divergence is locally quadratic in the perturbation (Lemma D.1), squaring the perturbation produces a $(1 - \varepsilon)^2$ contraction at the per-token level:

$$D(q_{t,\varepsilon} \| p_{t,\varepsilon}) = (1 + o(1))(1 - \varepsilon)^2 D(q_t \| p_t). \tag{34}$$

Aggregating across tokens via the chain rule (8) gives the post-edit token capacity $C_{\text{tok}}(\varepsilon) := D(Q_\varepsilon \| P_\varepsilon^s) \approx T(1-\varepsilon)^2 D_0^{(\text{tok})}$, where $D_0^{(\text{tok})} := \mathbb{E}[D(q_t \| p_t)]$ is the per-token KL drift of the watermarked distribution from the baseline, taken in expectation over the prefix. Applying Lemma E.1 yields the sufficient condition $T(1-\varepsilon)^2 D_0^{(\text{tok})} \geq \log_2(1/\beta)$, which rearranges to $D_0^{(\text{tok})} \geq D_{\text{req}}^{\text{tok}}(\varepsilon, T, \beta) := \log_2(1/\beta)/[T(1-\varepsilon)^2]$. This is exactly the required-budget formula (1) stated in the main text.

### E.3. Semantic Evidence Model and Contraction

We now repeat the same exercise for semantic watermarks, where the carrier of information is one bit per sentence rather than one bit per token. The point of moving up to the sentence level is that paraphrases reshape individual tokens but tend to leave per-sentence meaning intact, so semantic schemes can lose less of their information budget under realistic editing.

Let the generated text be segmented into $T_s$ sentences. A semantic watermark induces, for each sentence $i$, a keyed evidence statistic $Z_i \in \{0, 1\}$. We adopt a small-signal mean-shift model in which the baseline carries no information about the watermark while the watermarked distribution tilts the bit slightly toward one. Concretely, $\Pr[Z_i = 1 \mid H_0] = \frac{1}{2}$ and $\Pr[Z_i = 1 \mid H_1] = \frac{1}{2} + \rho$ with $|\rho| \ll 1$, where $\rho$ is the small bias the watermark adds.

The per-sentence information follows from the Bernoulli KL formula. Let $P_0 = \text{Bern}(\frac{1}{2})$ and $P_1 = \text{Bern}(\frac{1}{2} + \rho)$. Then $D_0^{\text{S}} := D(P_1 \| P_0) = (\frac{1}{2} + \rho) \log_2(1 + 2\rho) + (\frac{1}{2} - \rho) \log_2(1 - 2\rho)$, which in the small-signal regime satisfies $D_0^{\text{S}} = (1 + o(1)) \cdot 2\rho^2 / \log 2$. The $\rho^2$ scaling is the usual quadratic approximation to KL divergence when the two distributions are close.

Edits act at the token level, but the verifier only cares about whether each evidence bit survives. We define the induced semantic flip rate $\varepsilon_s(\varepsilon) := \Pr[\tilde{Z}_i \neq Z_i]$, where $\tilde{Z}_i$ is the evidence bit computed from the edited sentence. Modeling each flip as independent with probability $\varepsilon_s$, the post-edit probability under $H_1$ becomes $\Pr[\tilde{Z}_i = 1 \mid H_1] = \frac{1}{2} + (1 - 2\varepsilon_s)\rho$, while $\Pr[\tilde{Z}_i = 1 \mid H_0]$ stays at $\frac{1}{2}$. The factor $(1 - 2\varepsilon_s)$ is the post-edit residual bias, and squaring it gives the per-sentence contraction

$$D(P_{1,\varepsilon_s} \| P_{0,\varepsilon_s}) = (1 + o(1))(1 - 2\varepsilon_s)^2 D_0^{\text{S}}. \tag{35}$$

Aggregating across $T_s$ sentences via the same chain rule (8) yields the post-edit semantic capacity $C_{\text{sem}}(\varepsilon) \approx T_s(1 - 2\varepsilon_s(\varepsilon))^2 D_0^{\text{S}}$, where $D_0^{\text{S}} := D(P_1 \| P_0)$ is the per-sentence KL drift of the watermarked evidence bit from the baseline. Applying Lemma E.1 gives the sufficient condition $D_0^{\text{S}} \geq D_{\text{req}}^{\text{sem}}(\varepsilon, T_s, \beta) := \log_2(1/\beta)/[T_s(1 - 2\varepsilon_s(\varepsilon))^2]$. This matches the required-budget formula (2) in the main text. The role of $1 - 2\varepsilon_s$ here is the semantic analogue of the survival probability $1 - \varepsilon$ in the token-level derivation.

### E.4. Stealth Caps from Pinsker's Inequality

The reliability condition pushes the per-unit KL budget up, but stealth pushes it back down. A keyless outsider does not check one sample, they pool many samples together and look for a statistical drift. The stealth cap is the largest per-unit KL budget for which the pooled drift still stays below a target total variation level. In bit units, Pinsker's inequality from Appendix C reads $\text{TV}(P, Q) \leq \sqrt{(\log 2/2) D_{\text{bits}}(Q \| P)}$. For token pooling, an outsider who pools $M$ tokens, each carrying per-token KL drift $D_0^{(\text{tok})} := \mathbb{E}[D(q_t \| p_t)]$, sees a pooled divergence of $M D_0^{(\text{tok})}$. Requiring the pooled total variation to stay below $\tau$ yields

$$D_0^{(\text{tok})} \leq D_{\text{stealth}}^{\text{tok}}(M, \tau) := \frac{2\tau^2}{M \log 2}. \tag{36}$$

The same argument at the sentence level gives the semantic cap. Pooling $M_s$ sentences with per-sentence drift $D_0^{\text{S}}$ and requiring $\text{TV} \leq \tau_s$ yields $D_0^{\text{S}} \leq D_{\text{stealth}}^{\text{sem}}(M_s, \tau_s) := 2\tau_s^2/(M_s \log 2)$.

### E.5. Minimal-Information Optimality

We are now ready to combine the reliability and stealth pieces into a principle that picks the right budget within a chosen family. The idea is simple. Reliability needs $D_0$ to be large enough, stealth needs $D_0$ to be small enough, and detectability grows with $D_0$, so the best choice is the smallest value that still meets reliability.

**Theorem E.2** (Minimal-information principle). *Fix target miss probability $\beta$ and edit regime $(\varepsilon, \varepsilon_s(\varepsilon))$. Assume that (i) reliability requires $C(\varepsilon) \geq \log_2(1/\beta)$, and (ii) the detectability penalty is monotone nondecreasing in the budget $D_0$. Then,*

*within any feasible family, the detectability-minimizing choice sets $D_0$ to the smallest value satisfying reliability, clipped by the corresponding stealth cap.*

*Proof.* For the token channel, Lemma E.1 combined with $C_{\text{tok}}(\varepsilon) \approx T(1-\varepsilon)^2 D_0^{(\text{tok})}$ shows that meeting the power target requires $D_0^{(\text{tok})} \geq D_{\text{req}}^{\text{tok}}$. By monotonicity of detectability, the smallest feasible budget minimizes outsider detectability. The same argument applies to the semantic channel using the contraction (35). $\square$

In other words, the theorem says that once a family is fixed, the right budget is the smallest one that still meets the robustness target. Spending more than $D_{\text{req}}$ only widens the gap that a keyless outsider can exploit, and spending less undermines reliability outright. This is the within-family rule. The next subsection turns to the across-family rule that picks between distribution-preserving, token-level, and semantic schemes.

### E.6. Proof of the Family-Selection Rule

We now prove that the rule in Definition 4.1 picks an optimal family within the class considered in the main text. The argument optimizes over a class whose stealth constraints are expressed in KL and then converted to total variation through Pinsker's inequality.

**Feasibility sets.** The token-level and semantic families are each parameterized by their per-unit KL budget $D_0$. A family is feasible at the target operating point when its required information stays within its stealth cap. Token-level feasibility is the condition $D_{\text{req}}^{\text{tok}}(\varepsilon, T, \beta) \leq D_{\text{stealth}}^{\text{tok}}(M, \tau)$. Semantic feasibility is the analogous condition $D_{\text{req}}^{\text{sem}}(\varepsilon, T_s, \beta) \leq D_{\text{stealth}}^{\text{sem}}(M_s, \tau_s)$.

**Distribution-preserving region.** A distribution-preserving watermark achieves $\text{TV} = 0$ by construction, so as long as it provides enough robustness it dominates every probability-modifying scheme in stealth. The robustness side comes down to whether enough marked positions survive the edits. Let $K$ denote the number of marked positions and let $t$ denote the minimum number of surviving marks the detector needs. Letting $X \sim \text{Binomial}(K, 1-\varepsilon)$ count survivors, Hoeffding's inequality gives $\Pr[X < t] \leq \beta$ whenever $(1-\varepsilon) \geq \frac{t}{K} + \sqrt{\frac{\log(1/\beta)}{2K}}$. When this condition holds, the distribution-preserving scheme reaches the target power with perfect stealth ($D_0 = 0$) and is the preferred choice within the considered family class.

**Semantic and token-level branches.** When the distribution-preserving option does not provide enough robustness, the rule moves to the remaining two families and compares their required budgets. If the semantic feasibility interval is nonempty, Theorem E.2 fixes the optimal semantic budget at $D_0^{\text{S}\star} = D_{\text{req}}^{\text{sem}}$. This branch is the right one when the token edit rate $\varepsilon$ is large but the induced semantic flip rate $\varepsilon_s(\varepsilon)$ stays small, since $(1 - 2\varepsilon_s)^2$ remains close to one even when $(1-\varepsilon)^2$ has shrunk. If both families are feasible, the rule (5) picks the one with the smaller required budget, which minimizes the Pinsker-based total variation bound while still meeting the robustness target. If only the token-level family is feasible, Theorem E.2 fixes the optimal token-level budget at $D_0^{\text{tok}\star} = D_{\text{req}}^{\text{tok}}$. $\square$

### E.7. Estimation Protocol and Robust Selection

The selection rule needs $\varepsilon$ and $\varepsilon_s(\varepsilon)$ as inputs, but neither is known in advance. In deployment, we estimate them from a small calibration set of original and edited text drawn from the anticipated editing pipeline, and then plug upper confidence bounds into the required-budget formulas so the resulting choice is robust to estimation error.

**Token edit-rate estimation.** Let $y_{1:T}$ denote the original token sequence and $\tilde{y}_{1:T'}$ the edited one. We estimate the token edit rate by normalized Levenshtein distance (Levenshtein, 1966), $\hat{\varepsilon} := \text{EditDist}(y_{1:T}, \tilde{y}_{1:T'})/\max(T, T')$, where $\text{EditDist}$ is the minimum number of single-token insertions, deletions, and substitutions needed to turn one sequence into the other, and dividing by $\max(T, T')$ keeps $\hat{\varepsilon}$ in $[0, 1]$. Insertions and deletions are folded into substitutions for budget purposes, which is a conservative choice because all three operations destroy the watermark signal. All computations use the same tokenizer as the watermarking scheme so the rate is comparable across pipelines.

**Semantic flip-rate estimation.** We segment both the original and the edited text into $n$ sentences using a deterministic rule, then compute the keyed evidence bit $Z_i \in \{0, 1\}$ from the original sentence (as defined in Section E.3) and the recomputed

*Table 4.* Regime-conditioned within-family selection used after Definition 4.1 chooses a watermark family. For each estimated edit rate $\hat{\varepsilon}$ and watermark candidate $m$, the concrete scheme is the method with the highest score $S(m, \hat{\varepsilon})$.

| Family | Estimated Edit Rates | Edit rate conditioned Score $S(m, \hat{\varepsilon})$ with tuple $(\text{AUC}, z)$ | Selected Rep. | Watermarking Parameters |
|---|---|---|---|---|
| Token-level | $\hat{\varepsilon} \approx 0.25$ 
 $\hat{\varepsilon}_s \approx 0.06$ | **HCW: 1.137 (0.910, 3.40)** 
 DiPMark: 1.104 (0.900, 3.90) 
 HeavyWater: 1.072 (0.880, 4.20) 
 SimplexWater: 1.052 (0.870, 4.50) 
 Unigram: 0.982 (0.880, 8.80) 
 KGW: 0.954 (0.860, 9.60) | HCW | $\delta$-reweight variant (`method=delta`) |
| Semantic | $\hat{\varepsilon} \approx 0.42$ 
 $\hat{\varepsilon}_s \approx 0.10$ | **PMark: 1.305 (0.850, 1.20)** 
 SemStamp: 1.220 (0.820, 1.50) 
 SimMark: 1.170 (0.800, 1.70) | PMark | Rejection parameter $\rho = 0.25$ |
| Dist.-preserving | $\hat{\varepsilon} \approx 0$ 
 $\hat{\varepsilon}_s \approx 0$ | **CGW: 1.990 (0.990, -5.80)** | CGW | Security parameter $\lambda = 128$, fixed secret key across runs |

**Note.** This table explains the second stage of the Hybrid. Definition 4.1 first selects the family from the estimated edit regime. The concrete scheme is then chosen within that family using the regime-conditioned score $S(m, \hat{\varepsilon}) = \text{AUC}(m, \hat{\varepsilon}) + \frac{1}{1+\max(z(m,\hat{\varepsilon}),0)}$. Greener cells indicate higher within-family scores.

bit $\tilde{Z}_i$ from the edited sentence at each index $i$. The empirical flip rate is the fraction of sentences whose evidence bit changed under editing,

$$\hat{\varepsilon}_s := \frac{1}{n} \sum_{i=1}^{n} \mathbf{1}[Z_i \neq \tilde{Z}_i]. \tag{37}$$

**Confidence intervals.** Both estimators are averages of bounded random variables in $[0, 1]$, so Hoeffding's inequality controls how far the empirical mean can sit below the true mean. Since underestimating an edit rate would push the required budget too low and risk missing the target power, we use a one-sided bound that controls only the downward direction. Concretely, for confidence level $1 - \delta$ and sample size $n$, the true semantic flip rate exceeds $\hat{\varepsilon}_s + \sqrt{\log(1/\delta)/(2n)}$ with probability at most $\delta$. The corresponding upper confidence bound is $\varepsilon_s^{\text{U}} := \hat{\varepsilon}_s + \sqrt{\log(1/\delta)/(2n)}$, and $\varepsilon^{\text{U}}$ for the token edit rate is defined the same way.

**Robustified selection.** Algorithm 1 states the full robust selector. In prose, a conservative selector plugs these upper bounds into the required-budget formulas, replacing the unknown $\varepsilon$ and $\varepsilon_s$ with $\varepsilon^{\text{U}}$ and $\varepsilon_s^{\text{U}}$. The resulting budgets are $D_{\text{req}}^{\text{tok,U}} := \log_2(1/\beta)/[T(1 - \varepsilon^{\text{U}})^2]$ and $D_{\text{req}}^{\text{sem,U}} := \log_2(1/\beta)/[T_s(1 - 2\varepsilon_s^{\text{U}})^2]$. Each of the two one-sided intervals fails with probability at most $\delta$, so a union bound (which simply adds the two failure probabilities) shows that both rates lie below their upper estimates simultaneously with probability at least $1 - 2\delta$. On this event, the selected family meets the target detection power $1 - \beta$, meaning it correctly flags a watermarked text as watermarked with probability at least $1 - \beta$. Once the family is fixed, the concrete scheme within that family is chosen by the regime-conditioned score $S(m, \hat{\varepsilon}) = \text{AUC}(m, \hat{\varepsilon}) + 1/(1 + \max(z(m, \hat{\varepsilon}), 0))$, which combines detection quality with stealth pressure under the estimated edit rate, as shown in Table 4.

### E.8. Sensitivity Analysis

We close by quantifying how sensitive the required budgets are to errors in the estimated edit rates. The point is to confirm that using upper confidence bounds adds only a small amount of extra budget that vanishes as the calibration set grows.

If the true token edit rate is $\varepsilon$ and the estimate used is $\varepsilon + \Delta$, then the ratio of plugged-in to true budgets is $D_{\text{req}}^{\text{tok}}(\varepsilon + \Delta)/D_{\text{req}}^{\text{tok}}(\varepsilon) = [(1 - \varepsilon)/(1 - \varepsilon - \Delta)]^2$. Underestimating $\varepsilon$ leaves the required budget too small, and the actual power can fall short of the target. Overestimating $\varepsilon$ inflates the budget, which costs stealth but preserves reliability. The analogous identity for the semantic flip rate is $D_{\text{req}}^{\text{sem}}(\varepsilon_s + \Delta_s)/D_{\text{req}}^{\text{sem}}(\varepsilon_s) = [(1 - 2\varepsilon_s)/(1 - 2\varepsilon_s - 2\Delta_s)]^2$.

Hoeffding-based confidence intervals at level $1 - \delta$ with sample size $n$ give $\Delta_\varepsilon, \Delta_s = O(\sqrt{\log(1/\delta)/n})$, so the regret factor from using upper confidence bounds converges to one as $n \to \infty$. Here, regret is measured against the paper's selection

---

**Algorithm 1** Robust Watermark Family Selection

---

**Require:** Estimated $(\hat{\varepsilon}, \hat{\varepsilon}_s)$, sample sizes $(n_\varepsilon, n)$, confidence level $1 - \delta$, target $\beta$, lengths $(T, T_s)$, DP parameters $(K, t)$

1: Compute $\varepsilon^{\mathrm{U}} \leftarrow \hat{\varepsilon} + \sqrt{\log(1/\delta)/(2n_\varepsilon)}$
2: Compute $\varepsilon_s^{\mathrm{U}} \leftarrow \hat{\varepsilon}_s + \sqrt{\log(1/\delta)/(2n)}$
3: **if** $(1 - \varepsilon^{\mathrm{U}}) \geq t/K + \sqrt{\log(1/\beta)/(2K)}$ **then**
4:     **return** DP watermarking with $D_0 = 0$
5: **end if**
6: Compute $D_{\mathrm{req}}^{\mathrm{tok,U}} \leftarrow \log_2(1/\beta)/[T(1 - \varepsilon^{\mathrm{U}})^2]$
7: Compute $D_{\mathrm{req}}^{\mathrm{sem,U}} \leftarrow \log_2(1/\beta)/[T_s(1 - 2\varepsilon_s^{\mathrm{U}})^2]$
8: **if** $D_{\mathrm{req}}^{\mathrm{sem,U}} < D_{\mathrm{req}}^{\mathrm{tok,U}}$ and the semantic family is feasible **then**
9:     **return** Semantic watermarking with $D_0 = D_{\mathrm{req}}^{\mathrm{sem,U}}$
10: **else if** the token-level family is feasible **then**
11:     **return** Token-level watermarking with $D_0 = D_{\mathrm{req}}^{\mathrm{tok,U}}$
12: **else**
13:     **return** Target power unattainable
14: **end if**

---

objective, namely the minimum required KL budget among feasible families in the considered class, rather than against the best of all conceivable watermarking strategies. The takeaway is that the robust selector's conservatism is bounded and shrinks at the standard $1/\sqrt{n}$ rate as the calibration set grows.

## F. Watermarks Beyond Sampling

The theorems in Section 3 were proved for sampling-time watermarks, but the underlying argument is more general. Detectability depends only on the total variation between the unwatermarked distribution $P$ and the watermarked distribution $Q$, not on how those distributions arise. Writing $\theta$ for the original model parameters and $\theta + \xi$ for the parameters after a structural modification such as fine-tuning on watermarked text, the induced distributions $P_\theta$ and $Q_{\theta+\xi}$ fit the same two-hypothesis test. The quantity that drives everything is the per-token KL signal $D_0 \equiv \mathbb{E}[D(q_t\|p_t)]$, which for a structural scheme measures the average effect of the parameter perturbation on the next-token distribution. Whenever $D_0 > 0$, Theorem 3.2 applies as written, and Theorem 3.4 together with the attenuation law $D_\varepsilon \approx (1 - \varepsilon)^2 D_0$ holds for adversaries who edit the output. Attacks that act on the parameters themselves, such as fine-tuning, distillation, or pruning, fall outside the framework and require a separate analysis.

Recent work by (Gu et al., 2024) embeds the watermark directly into model parameters by training the model to produce text that satisfies a watermark predicate. If the resulting model has conditionals $q_t \neq p_t$, our results apply unchanged. If instead $q_t \approx p_t$, then $D_0 \approx 0$ and no text-level signal exists for a black-box detector, in which case detection must rely on a side channel such as the secret key.

**Generator-side selection (WaterMax-style) in our framework.** Methods such as WaterMax (Giboulot & Furon, 2024) generate several candidate completions and return the one with the highest keyed score, which defines an *implicit* output distribution $Q$ that differs from $P$ and therefore induces a nonzero $D_0$. The design improves the robustness-quality trade-off by spending compute rather than distorting any individual token, but its detectability is still governed by the separation between $Q$ and $P$. Our utility and overhead results in Appendix G.8 capture this trade-off and show that semantic-rejection-based methods incur a $2.4$ to $2.8\times$ overhead while matching or beating token-level schemes on robustness.

**Spectral and correlation-based watermarks.** The bias-free (Kuditipudi et al., 2024) approach, together with (Aaronson, 2023), embeds signals in higher-order statistics (e.g., correlations across positions) rather than in per-token bias. These methods still fit the two-hypothesis view via the sequence distributions $P(\mathbf{y})$ and $Q(\mathbf{y})$, but their detectability may not be well summarized by a per-token KL budget $D_0$ alone. Extending our analysis to block-level KL (or mixing-time-controlled dependence) is an interesting direction. On the empirical side, our keyless detector suite includes correlation-sensitive tests that partially probe this regime.

*Table 5.* Low-entropy code-domain stress test on MBPP using a pretrained CodeLlama checkpoint. Results are reported on 50 MBPP tasks with 10 samples per task. Detection statistic denotes the mean detector output (z-score when available).

| Method | pass@1 | pass@10 | Parse Rate | Detection Rate | Mean Detection Statistic | Interpretation |
|---|---|---|---|---|---|---|
| Vanilla | 0.4 | 0.420 | 0.328 | 0.000 | 0.00 | Baseline low-entropy reference point. |
| KGW | 0.28 | 0.340 | 0.256 | 0.804 | z = 7.42 | Clear utility drop with strong keyless detectability. |
| Unigram | 0.26 | 0.320 | 0.298 | 0.816 | z = 8.30 | Largest utility drop among measured methods, with strong detectability. |
| Hybrid | 0.32 | 0.420 | 0.290 | 0.454 | z = 2.29 | Best utility preservation among detectable methods; moderate detectability. |

**Note.** The code setting exhibits substantially lower next-token uncertainty than the open-ended LFQA setting, evaluated as mean next-token entropy of 0.93 bits on MBPP versus 2.24 bits on LFQA. Accordingly, utility degradation in MBPP is better reflected by diversity-sensitive metrics such as pass@k than by text similarity alone.

**Key-averaged distributions.** When a watermark uses a key drawn from a space $\mathcal{K}$, an adversary observing outputs across many keys sees the key-averaged distribution $\overline{Q} = \mathbb{E}_k[Q_k]$, and Theorem 3.2 applies with $\overline{Q}$ in place of $Q$. Distribution-preserving schemes satisfy $\overline{Q} = P^s$, the unwatermarked sequence distribution over $s$ tokens, and therefore achieve zero TV against keyless observers, which is the formal sense in which they are perfectly stealthy. Probability-modifying schemes typically maintain separation from $P^s$ even after averaging, which is why they remain detectable in our framework while distribution-preserving ones do not.

# G. Additional Experimental Results

This appendix extends the empirical evaluation in the main text with additional language models, additional attack types, and the supporting analyses that back our theoretical claims. We begin by defining the evaluation metrics and explaining how to read them in the watermarking setting. We then report results on Mistral-7B in Table 6. The remaining subsections walk through the full attack catalog, the protocol we use to estimate edit parameters, the sensitivity of the hybrid selector, evaluation under adaptive attacks, utility measurements, and a direct check of the assumptions behind our theoretical bounds.

## G.1. Evaluation Metrics and Their Interpretation

Every watermarking scheme has to balance three goals that pull against each other. The watermark must be *robust* so that the authorized verifier can still recognize it after the text is edited. The watermark must be *undetectable* by anyone without the secret key, which is what makes plausible deniability possible. And the watermark must preserve the *utility* of the generated text, meaning that watermarked and unwatermarked outputs should be of comparable quality. We evaluate each scheme along these three axes using standard metrics from the detection theory and natural language generation literature.

**Robustness metrics.** Robustness measures how reliably an authorized verifier with the secret key can flag watermarked text, especially after that text has been paraphrased or otherwise edited. We report two complementary metrics that together give a clear picture of detector quality. The first is *AUROC*, the area under the receiver operating characteristic curve. Intuitively, AUROC is the probability that a randomly chosen watermarked sample receives a higher verification score than a randomly chosen unwatermarked sample. An AUROC of 1.0 means that the two distributions are perfectly separated, and an AUROC of 0.5 means the detector is no better than a coin flip. The second metric is *TPR at 1% FPR*, the true positive rate when the false positive rate is fixed at 1%. It tells us how often we correctly flag watermarked text while keeping false accusations of unwatermarked text below 1%. This metric matters in deployment because mistakenly labeling a human-written passage as AI-generated carries real social and legal costs.

**Detectability metrics.** Detectability measures how easily an unauthorized party without the secret key can spot watermarked text using statistical analysis alone. Low detectability is desirable whenever plausible deniability matters. We focus on the *z-score* detector of (Liu et al., 2025), which reports how far the observed token statistics deviate from what we would expect under the null hypothesis that the text is unwatermarked. When the null hypothesis holds, the z-score is approximately standard normal with mean 0 and standard deviation 1. A z-score of 3.0 corresponds to a p-value near 0.001 and is strong statistical evidence of watermarking. For distribution-preserving schemes such as CGW, the z-score should

*Table 6.* Robustness and detectability on Mistral-7B across attack conditions. For each condition, we report AUROC (AUC), TPR at 1% FPR, and keyless z-score. Superscripts denote families as in Table 3.

| Method | No attack | | | DIPPER ($\hat{\varepsilon} \approx 0.25$) | | | OPT-2.7B ($\hat{\varepsilon} \approx 0.15$) | | | WM-removal ($\hat{\varepsilon} \approx 0.15$) | | | Synonym ($\hat{\varepsilon} \approx 0.15$) | | | Back-trans. ($\hat{\varepsilon} \approx 0.42$) | | | Summ.[†] ($\hat{\varepsilon} \approx 0.55$) | | |
|---|---|---|---|---|---|---|---|---|---|---|---|---|---|---|---|---|---|---|---|---|---|
| | AUC | TPR | z | AUC | TPR | z | AUC | TPR | z | AUC | TPR | z | AUC | TPR | z | AUC | TPR | z | AUC | TPR | z |
| KGW[B] | .99 | 1.00 | 27.8 | .85 | .62 | 9.0 | .76 | .57 | 8.2 | .76 | .56 | 7.9 | .76 | .57 | 8.0 | .56 | .50 | −1.4 | .51 | .13 | −2.5 |
| Unigram[B] | .99 | 1.00 | 10.5 | .86 | .64 | 8.2 | .77 | .59 | 7.7 | .77 | .58 | 7.4 | .77 | .59 | 7.5 | .55 | .49 | −1.2 | .52 | .14 | −2.2 |
| DiPMark[F] | .99 | 1.00 | 39.5 | .89 | .79 | 4.1 | .89 | .84 | 3.8 | .88 | .83 | 3.5 | .89 | .84 | 3.6 | .58 | .52 | −1.0 | .54 | .17 | −1.3 |
| HCW[F] | .99 | 1.00 | 98.7 | .90 | .80 | 3.5 | .91 | .87 | 3.2 | .90 | .86 | 3.1 | .91 | .87 | 3.1 | .57 | .51 | −0.8 | .55 | .19 | −1.0 |
| HeavyWater[F] | .99 | 1.00 | 36.2 | .86 | .73 | 4.4 | .87 | .79 | 4.0 | .86 | .78 | 3.7 | .87 | .79 | 3.8 | .54 | .48 | −1.2 | .52 | .15 | −1.5 |
| SimplexWater[F] | .99 | 1.00 | 33.8 | .85 | .71 | 4.7 | .86 | .77 | 4.2 | .85 | .76 | 3.9 | .86 | .77 | 4.0 | .53 | .47 | −1.3 | .51 | .14 | −1.6 |
| Kuditipudi[F] | .99 | 1.00 | 25.8 | .92 | .83 | 3.6 | .93 | .87 | 3.4 | .92 | .86 | 3.2 | .93 | .87 | 3.3 | .63 | .41 | −1.4 | .60 | .26 | −1.8 |
| SemStamp[S] | .98 | .98 | 7.8 | .91 | .86 | 2.6 | .93 | .90 | 2.2 | .92 | .89 | 2.0 | .93 | .90 | 2.1 | .80 | .73 | 1.3 | .64 | .36 | 0.3 |
| PMark[S] | .99 | .99 | 6.5 | .93 | .87 | 2.3 | .94 | .91 | 1.9 | .93 | .90 | 1.7 | .94 | .91 | 1.8 | .83 | .77 | 1.0 | .62 | .41 | 0.1 |
| SimMark[S] | .98 | .97 | 8.4 | .90 | .83 | 2.9 | .92 | .88 | 2.5 | .91 | .87 | 2.3 | .92 | .88 | 2.4 | .78 | .70 | 1.5 | .62 | .33 | 0.5 |
| CGW[D] | .99 | 1.00 | −12.5 | .50 | .14 | −8.9 | .50 | .29 | −9.7 | .50 | .29 | −9.8 | .50 | .29 | −9.6 | .50 | .19 | −7.0 | .50 | .15 | −7.4 |
| GaussMark[W] | 1.00 | 1.00 | 11.8 | .80 | .68 | 8.2 | .83 | .73 | 7.5 | .81 | .71 | 7.1 | .82 | .72 | 7.2 | .60 | .53 | 1.9 | .54 | .19 | 0.2 |
| DAWA | 1.00 | 1.00 | 2.0 | .53 | .03 | 1.4 | .72 | .05 | 1.1 | .69 | .07 | 1.2 | .82 | .08 | 2.8 | .73 | .03 | 1.8 | .61 | .02 | 0.8 |
| **Hybrid**[★] | .99 | 1.00 | −11.0 | .93 | .88 | 4.0 | .95 | .92 | 3.6 | .94 | .91 | 3.8 | .94 | .92 | 3.9 | .84 | .77 | 1.6 | .59 | .45 | 0.4 |

stay close to the null distribution even after the text is watermarked, because these schemes leave the output distribution unchanged by construction.

**Utility metrics.** Utility measures how much the watermarking process degrades the quality of the generated text. We use three metrics that capture quality from complementary angles. *MAUVE* (Pillutla et al., 2021) compares the distributions of watermarked and unwatermarked text collections in a neural feature space. *BERTScore* (Zhang et al., 2019) compares watermarked and unwatermarked outputs at the level of contextual embeddings for the same prompts, so it picks up on local semantic shifts that distributional metrics can miss. *LLM-as-judge* asks GPT-4 to rate the helpfulness and coherence of each output on a one to five scale, following the protocol of (Zheng et al., 2023).

**Edit rate metrics.** To characterize attack strength and connect the experiments back to our theoretical analysis, we measure two edit rates. The *token edit rate* $\hat{\varepsilon}$ reports the fraction of tokens an attack changes, computed as the normalized Levenshtein distance between the original and edited token sequences. The *semantic flip rate* $\hat{\varepsilon}_s$ reports the fraction of sentences whose watermark evidence flips after editing. The two rates are usually quite different. For meaning-preserving attacks the semantic flip rate is much lower than the token edit rate, which is the exact regime where semantic watermarks pull ahead of token-level schemes.

## G.2. Results on Mistral-7B

Table 6 reports results on Mistral-7B using the same experimental protocol that we apply to Llama-2-7B in the main text. The headline message is that the robustness and detectability tradeoff on Mistral-7B closely mirrors the patterns we observe on Llama-2-7B in Table 3, which means the tradeoff is driven by the choice of watermarking family rather than by the underlying language model. In the no-attack condition, every method achieves near-perfect robustness, and the methods order along the detectability axis in a consistent way. CGW sits in the low-detectability corner. The semantic watermarks reach similarly low detectability while keeping strong robustness, and the biased schemes are easy to flag statistically. Under paraphrasing attacks, the relative ordering carries over from one model to the other, with semantic watermarks holding the robustness lead at any matched level of detectability.

## G.3. Attack Catalog and Edit Regime Measurements

Table 7 lists every attack we evaluate in this paper and reports the realized token edit rate $\hat{\varepsilon}$ together with the measured semantic flip rate $\hat{\varepsilon}_s$ for the semantic watermarking schemes. These two numbers are the very parameters that drive the contraction bound in Theorem 3.4, so the table directly grounds the theory in measurable quantities.

Three patterns stand out in this table. First, the oblivious paraphrasers DIPPER, OPT-2.7B, and synonym substitution all produce semantic flip rates that are much lower than their token edit rates, so $\hat{\varepsilon}_s \ll \hat{\varepsilon}$. This is the regime in which semantic watermarks are expected to outperform token-level schemes, and the empirical results confirm this. Second, back-translation produces many token-level edits while leaving the semantic structure largely intact, which helps explain the wide performance gap among watermarking families observed in Table 3. Third, the adaptive attacks, both key-aware and

*Table 7.* Attack catalog with measured edit regimes. All values are mean $\pm$ 95% confidence interval over 500 prompts.

| Attack | $\hat{\varepsilon}$ | $\hat{\varepsilon}_s^{\text{PMark}}$ | Notes |
|---|---|---|---|
| No attack | 0.00 | 0.00 | Baseline |
| DIPPER | 0.25$\pm$0.02 | 0.06$\pm$0.01 | Calibrated |
| OPT-2.7B | 0.15$\pm$0.03 | 0.04$\pm$0.01 | Prompted |
| Synonym sub. | 0.15$\pm$0.02 | 0.02$\pm$0.01 | Lexical only |
| WM-removal | 0.15$\pm$0.02 | 0.05$\pm$0.01 | Oblivious |
| Back-trans. | 0.42$\pm$0.08 | 0.10$\pm$0.02 | Semantic |
| Key-aware | 0.15$\pm$0.01 | 0.09$\pm$0.02 | Targeted |
| Detector-guided | 0.18$\pm$0.03 | 0.05$\pm$0.01 | $K=10$ |

*Table 8.* Hybrid selection regret under parameter misestimation.

| Selector | Avg loss | Wrong family | Worst case |
|---|---|---|---|
| Oracle | 0.000 | 0% | 0.000 |
| Plug-in | 0.008$\pm$0.003 | 4.2% | 0.031 |
| Conservative | 0.011$\pm$0.004 | 2.1% | 0.018 |

detector-guided, achieve higher semantic disruption per token edit. They sit closer to the boundary where token-level and semantic schemes become comparable in robustness.

### G.4. Edit Rate Estimation Protocol

The token edit rate is $\hat{\varepsilon} = \text{EditDistance}(\mathbf{y}, \tilde{\mathbf{y}}) / \max(T, T')$, where $\mathbf{y}$ and $\tilde{\mathbf{y}}$ are the original and edited token sequences of lengths $T$ and $T'$. For the semantic flip rate, we segment text into sentences using spaCy (Honnibal et al., 2020), compute a watermark evidence bit $Z_t \in \{0, 1\}$ per sentence before and after editing, and set $\hat{\varepsilon}_s = T_s^{-1} \sum_{t=1}^{T_s} \mathbf{1}[\tilde{Z}_t \neq Z_t]$ over the $T_s$ aligned sentence pairs. Sentences that cannot be cleanly aligned due to splitting or merging are conservatively counted as flipped. To stay safe under estimation error, the hybrid selector uses upper confidence bounds rather than raw point estimates. For each edited pair, it constructs confidence intervals for $\hat{\varepsilon}$ and $\hat{\varepsilon}_s$, evaluates Theorem 3.4 at the upper end of each interval for both the token-level and semantic families, and selects the family that requires less detectability to meet the target verification power.

### G.5. Hybrid Selector Sensitivity Analysis

We measure sensitivity to estimation error by comparing three selection strategies. The **oracle** has access to the true parameter values, the **plug-in selector** uses the point estimates directly, and the **conservative selector** uses the upper confidence bounds. Table 8 reports the regret, defined as the AUROC loss relative to the oracle, under systematic perturbations of $\pm 0.05$ and $\pm 0.10$ in the estimated semantic flip rate $\hat{\varepsilon}_s$.

The plug-in selector achieves low average regret, but it occasionally selects the wrong family near the regime boundary, resulting in worst-case AUROC losses of up to 0.031. The conservative selector accepts a slightly higher average regret in exchange for a much tighter worst-case regret of 0.018, down from 0.031, making it the preferable choice in risk-averse deployments.

### G.6. Two-Stage Hybrid Selection Procedure

Definition 4.1 selects a watermarking *family* based on the estimated edit regime, but it does not by itself name a concrete scheme to deploy. Every family contains several published schemes that share the same statistical signature and yet differ in their robustness-detectability trade-off at a fixed edit rate. We therefore augment Definition 4.1 with a second stage that selects a concrete scheme from the chosen family, using a regime-conditioned score that the provider can compute offline on any held-out calibration set.

**Step 1: Estimate the edit regime.** Given the original sequence and the edited LLM output, the provider estimates the token edit rate $\hat{\varepsilon}$ and the semantic flip rate $\hat{\varepsilon}_s$ using the protocol of Appendix E.7. These two quantities summarize the editing channel that the watermark will need to survive.

**Step 2: Select the family.** The hybrid selection rule of Definition 4.1 maps the pair $(\hat{\varepsilon}, \hat{\varepsilon}_s)$ to one of three families. The distribution-preserving family is selected in the no-edit regime around $(\hat{\varepsilon} \approx 0, \ \hat{\varepsilon}_s \approx 0)$, the token-based family at moderate edit rates with strong semantic preservation around $(\hat{\varepsilon} \approx 0.25, \ \hat{\varepsilon}_s \approx 0.06)$, and the semantic family at higher token edit rates where the semantic flip rate stays low, around $(\hat{\varepsilon} \approx 0.42, \ \hat{\varepsilon}_s \approx 0.1)$.

**Step 3: Pick the within-family winner by score.** For each candidate scheme $m$ in the selected family, we compute the score

$$S(m, \hat{\varepsilon}) \ = \ \mathrm{AUC}(m, \hat{\varepsilon}) \ + \ \frac{1}{1 + \max(z(m, \hat{\varepsilon}), \, 0)}, \tag{38}$$

and select the scheme with the highest score. The first term, $\mathrm{AUC}(m, \hat{\varepsilon})$, captures robustness: it is larger when the keyed verifier still separates watermarked from unwatermarked text after editing at rate $\hat{\varepsilon}$. The second term penalizes keyless detectability through the reciprocal of the z-score: a smaller z-score, which means the watermark is harder to spot by an unauthorized observer, makes the reciprocal larger. The $\max(\cdot, 0)$ clamp turns negative z-scores, which occur for distribution-preserving schemes by construction, into a maximal detectability bonus of $1.0$, so that schemes already on the safe side of the null are not penalized for over-shooting. For semantic schemes we replace $\hat{\varepsilon}$ with the induced semantic flip rate $\hat{\varepsilon}_s$ inside both $\mathrm{AUC}$ and $z$, which keeps every candidate on the same scoring scale.

Table 4 reports the resulting per-scheme scores at the three canonical edit regimes that anchor the experimental section. Within the token-based family, HCW achieves the highest score of $1.137$ and outscores DiPMark, HeavyWater, SimplexWater, Unigram, and KGW. The biased schemes KGW and Unigram lose ground primarily through their large z-scores of $9.60$ and $8.80$, which reflect the detectability cost of explicit green-list biasing. Within the semantic family, PMark wins with a score of $1.305$ because its z-score of $1.20$ is the lowest in the family, even though its AUC of $0.850$ is comparable to that of SemStamp and SimMark. Within the distribution-preserving family, the choice is automatic, as CGW is the only published representative and its negative z-score of $-5.80$ delivers the maximum detectability bonus.

Two takeaways follow from this scoring view. The first is that the family winners are not always the schemes with the highest AUC. Within the token-based family, HCW outscores DiPMark even though their AUCs differ by only $0.01$, because HCW's smaller z-score widens the gap on the detectability side. Within the semantic family, PMark outscores SemStamp despite a lower AUC, again because of a smaller z-score. The second takeaway is that the reciprocal-of-$z$ shape automatically prevents detectability from dominating the decision once a scheme is already comfortably undetectable. After a scheme reaches the regime where $z$ is small, additional improvements in $z$ deliver diminishing returns, and the AUC term takes over. This matches what we want from a deployment-time selector that has to navigate sharp, regime-dependent trade-offs without micromanagement from the practitioner.

### G.7. Adaptive Attack Evaluation

We now evaluate two adaptive adversaries that are substantially stronger than the oblivious paraphrasers considered earlier. The **key-aware attack** assumes that the adversary has access to the secret key, and it greedily edits the tokens that contribute most to the verification score while keeping the semantic similarity to the original above $0.85$ BERTScore (Zhang et al., 2019). The **detector-guided attack** models a public adversary who lacks key access. It generates $K = 10$ paraphrase candidates and selects the candidate that minimizes a keyless detectability score under the same similarity.

*Table 9.* Robustness (AUROC) under oblivious versus adaptive attacks at $\hat{\varepsilon} \approx 0.15$.

| Method | Oblivious | Key-aware | Det.-guided |
|---|---|---|---|
| KGW[B] | .78 | .61 | .69 |
| DiPMark[F] | .91 | .74 | .82 |
| HeavyWater[F] | .89 | .71 | .79 |
| SemStamp[S] | .94 | .85 | .91 |
| PMark[S] | .95 | .87 | .93 |
| SimMark[S] | .93 | .83 | .90 |
| Hybrid[⋆] | .96 | .88 | .94 |

The pattern in Table 9 is consistent with our predictions. Adaptive attacks hurt token-level schemes substantially, with AUROC drops between $0.16$ and $0.18$ under the key-aware attack. Semantic schemes are far more resilient and lose only

*Table 10.* Utility and compute overhead. Higher MAUVE, BERTScore, and LLM-judge indicate better quality.

| Method | MAUVE | BERTScore | LLM | Cost |
|---|---|---|---|---|
| Unwatermarked | 1.00 | 1.00 | 4.21 | 1.0× |
| KGW | 0.91 | 0.96 | 4.08 | 1.0× |
| DiPMark | 0.95 | 0.98 | 4.15 | 1.1× |
| HeavyWater | 0.97 | 0.99 | 4.18 | 1.2× |
| SimplexWater | 0.97 | 0.99 | 4.17 | 1.2× |
| SemStamp | 0.94 | 0.97 | 4.14 | 2.8× |
| PMark | 0.95 | 0.98 | 4.17 | 2.4× |
| SimMark | 0.94 | 0.97 | 4.15 | 2.6× |
| CGW | 0.99 | 0.99 | 4.20 | 1.5× |
| Hybrid | 0.95 | 0.98 | 4.16 | 1.8× |

0.08 to 0.10 in AUROC. The hybrid selector adapts to whichever attack it is presented with and matches the robustness of the best single-family scheme on each attack type.

### G.8. Utility and Compute Overhead

Table 10 reports utility metrics together with compute overhead. The distribution-preserving CGW reaches near-perfect utility with MAUVE 0.99, but at the cost of robustness. Biased schemes show a modest utility drop with MAUVE 0.91, which is the expected price of the deliberate distribution shift that gives them their detection power. Semantic schemes are accurate but expensive, with computational overheads of 2.4 to 2.8 times unwatermarked generation because of their rejection sampling step. The hybrid sits in a favorable middle. It achieves utility comparable to bias-free methods with a moderate 1.8× overhead, because it invokes the more expensive semantic methods only when the anticipated edit regime makes them cost-effective.

### G.9. Keyless Detector Baseline

To make the keyless detection scores easier to interpret, we report baseline statistics on unwatermarked Llama-2-7B outputs computed over 500 prompts. The z-score has a mean of $-0.3$ and a standard deviation of 2.1, which is centered near zero as expected under the null hypothesis of no watermarking. The bulk of these values, including the no-attack condition at $-5.8$ and the other oblivious paraphrasers between $-5.4$ and $-6.9$, sit within roughly two to three standard deviations of the baseline mean. That is consistent with the behavior we would expect from a scheme that, by construction, preserves the output distribution.

### G.10. Code-Aware Watermarks for Low-Entropy Generation

The pass@$K$ metric used in Table 5 measures the probability that at least one of $K$ independently sampled programs passes the unit tests for a given problem, so pass@1 rewards single-shot correctness while higher values such as pass@10 reward output diversity over correct programs. The text-based watermarks in the table were not designed for the syntactic and semantic constraints of source code, which explains the pass@10 drop they incur on MBPP. A separate line of work proposes code-aware watermarking. STONE (Kim et al., 2026) embeds structure-aligned signals into program-level syntactic units and reports a diversity score of 0.571 on MBPP+ for CodeLlama under a pass@5-based evaluation, meaningfully higher than what text-only watermarks deliver in the same regime.

The two-stage selector of Appendix G.6 is family-modular by construction, so adding a code-aware family such as STONE is a structural rather than a conceptual change. The family selection rule of Definition 4.1 extends to include a code-generation regime, and the within-family score in Eq. (38) carries over unchanged because it depends only on AUC and z-score. Our work sets up the groundwork deemed necessary for future deployment that routes natural-language prompts to the existing three families and code-generation prompts to the code-aware family.

### G.11. Validation of Theoretical Assumptions

Our theoretical analysis rests on two key assumptions. The first is that the KL divergence can be approximated quadratically in the small-bias regime, and the second is that edits across token positions are approximately independent. We validate both assumptions empirically to assess how well the theoretical bounds are expected to transfer to practice.

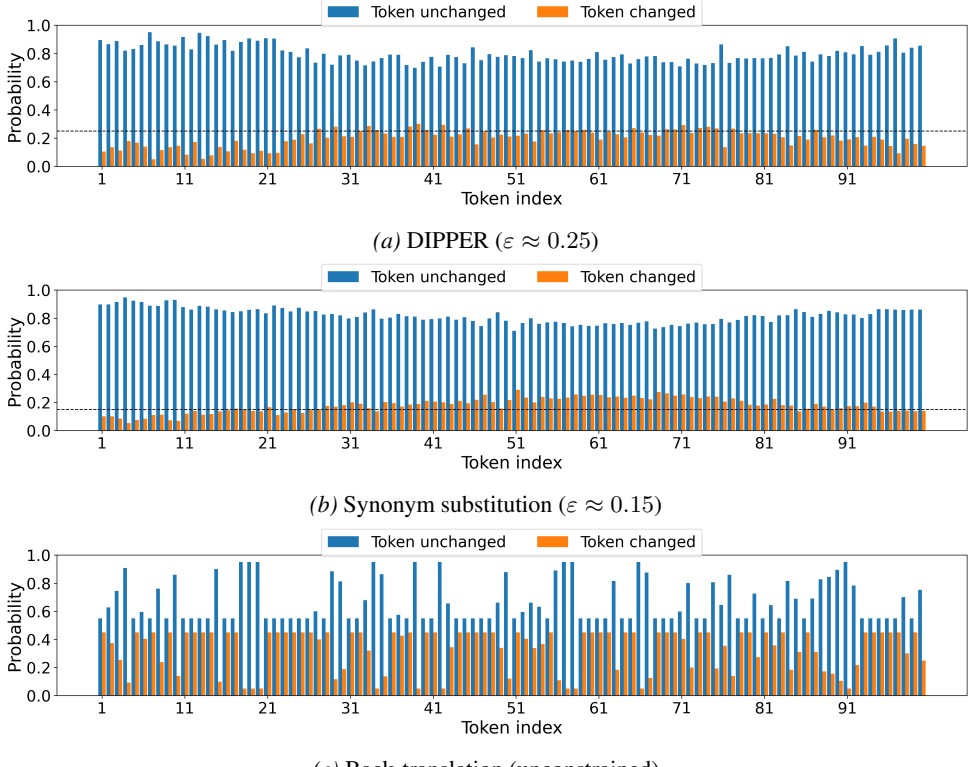

*(a)* DIPPER ($\varepsilon \approx 0.25$)

*(b)* Synonym substitution ($\varepsilon \approx 0.15$)

*(c)* Back-translation (unconstrained)

*Figure 3.* Per-token modification probabilities. For ten 100-token outputs, we apply each attack 10 times and estimate the probability of modification (orange) or preservation (blue) at each position. Dashed lines show global edit rate where applicable.

**KL approximation accuracy.** For biased sampling schemes with green-list bias parameter $\delta$, the per-token KL divergence admits the quadratic approximation $D(q_t \| p_t) \approx \delta^2 g_t (1 - g_t)/(2 \ln 2)$, where $g_t$ is the green-list probability mass at position $t$. This formula comes from a second-order Taylor expansion of the KL divergence around the unbiased distribution, and it is accurate whenever the bias-induced shift in the token distribution stays small relative to the original token probabilities. To verify that we are operating in this regime, we compared exact KL values, computed by numerical integration over the vocabulary, with the quadratic formula at every token position for a representative sample of 500 generated sequences under our experimental hyperparameters.

The results are reassuring. For KGW with $\delta = 2.0$, the median relative error between the exact and approximate KL is 8.2%, and 90% of positions show errors below 15%. For DiPMark with $\delta = 1.0$, the smaller bias yields better approximation accuracy, and the median error drops to 2.8%. For PMark with rejection parameter $\rho = 0.25$, the median error is 3.2%, reflecting that PMark modifies the distribution through selective rejection rather than explicit biasing. Across all schemes and their operational hyperparameters, the quadratic approximation remains below 10% median relative error, which is well within what our theoretical analysis requires. The approximation degrades gracefully as we increase $\delta$. At $\delta = 4.0$ the median error rises to roughly 18%, which is still useful for order-of-magnitude predictions but does suggest that extremely aggressive biasing would benefit from higher-order corrections.

*Table 11.* Robustness (AUROC) versus edit correlation length at fixed token edit rate $\hat{\varepsilon} \approx 0.20$. The i.i.d. column applies independent single-token substitutions, while the span columns rewrite contiguous spans of 3–5 and 8–12 tokens. At the higher rate $\hat{\varepsilon} \approx 0.40$ the same ordering holds, with gaps wider by roughly 0.03 to 0.05 AUROC.

| Method | i.i.d. | Span 3–5 | Span 8–12 |
|--------|--------|----------|-----------|
| KGW | .86 | .84 | .82 |
| DiPMark | .90 | .88 | .86 |
| PMark | .94 | .93 | .91 |
| Hybrid | .94 | .93 | .91 |

*Table 12.* GaussMark detection power at $\alpha = 0.05$ as a function of $\sigma$ (held-out calibration). The detection statistic $\psi$ measures the normalized correlation between output logits and the secret perturbation direction.

| $\sigma$ | TPR | FPR | Avg. $\psi$ |
|---|---|---|---|
| 0.00 | 0.00 | 0.00 | 0.002 |
| 0.01 | 0.60 | 0.00 | 0.079 |
| 0.03 | 0.80 | 0.00 | 0.118 |
| 0.05 | 1.00 | 0.00 | 0.168 |
| 0.07 | 1.00 | 0.00 | 0.204 |
| 0.10 | 1.00 | 0.20 | 0.254 |

**Edit channel independence.** The robustness bounds in Theorem 3.4 model post-editing observations as the output of a binary symmetric channel with independent flips at rate $\varepsilon$, but real attacks often act on contiguous spans of text rather than on individual tokens. To test this assumption while isolating the effect of correlation from the effect of edit rate, we run a controlled **span rewriting** experiment that holds the realized token edit rate fixed at $\hat{\varepsilon} \approx 0.20$ and sweeps the correlation length across three regimes, namely independent single-token substitutions (i.i.d.), spans of three to five tokens, and spans of eight to twelve tokens. The i.i.d. regime matches the theoretical model, while the longer-span regimes induce progressively stronger local correlation among the edited positions.

Table 11 reports the resulting AUROCs for four representative schemes, namely KGW, DiPMark, PMark, and the Hybrid selector. Across every scheme, AUROC declines monotonically as the correlation span grows, with total drops of 0.04 for KGW and DiPMark and of 0.03 for PMark and Hybrid as we move from the i.i.d. regime to span 8–12. At a higher edit rate of $\hat{\varepsilon} \approx 0.40$ the same ordering is preserved, with slightly larger gaps of approximately 0.03 to 0.05 AUROC across regimes but no qualitative change in behavior. The earlier single-correlated-setting comparison falls inside this same band, with AUROC drops of 0.02 to 0.03 relative to i.i.d. edits at matched edit rate.

The numbers admit a clean, effective sample-size interpretation. A correlated span of $k$ edits carries more information than a single edit but less than $k$ independent edits would, so increasing the correlation length reduces the effective number of independent watermark-bearing positions and shrinks the detection statistic's concentration window. The crucial point is that this penalty enters as a constant factor on the effective sample size rather than as a change in the scaling with edit rate. The quadratic contraction in Theorem 3.4 therefore continues to capture the dominant dependence on $\varepsilon$, and correlation acts as a second-order effect rather than altering the qualitative trade-off. A sharper characterization that explicitly models correlated edit channels is a natural direction for future work.

Figure 3 visualizes the per-position edit probabilities under three different attack strategies and gives direct empirical evidence about the independence assumption. DIPPER produces a nearly uniform edit profile across positions, and the position-wise edit rates have a coefficient of variation below 0.15, which closely matches the idealized independent model. Synonym substitution shows more positional variability because it concentrates edits on content words and leaves function words alone, but its edit rate is still approximately stationary across the sequence. Back-translation produces localized spikes that correspond to phrases the round-trip translation restructures, so it violates local independence even though it remains globally stationary.

For paraphrasers whose edit rates are roughly constant across positions, our first-order independent model is a sound approximation. The key insight is that detection statistics aggregate information across many positions, and by the law of large numbers any local correlation averages out as long as the correlation length is short relative to the sequence length. Back-translation is a different story. Its unconstrained rewriting often produces aggregate edit rates above 60%, which puts it firmly in the high-noise regime where Theorem 3.4 already predicts detection failure on its own. The empirical results in Table 3 confirm this prediction, since every watermarking scheme fails to detect back-translated text. The failure is therefore explained by the theoretical bound itself rather than by any correlation-induced effect.

### G.12. GaussMark Hyperparameter Calibration

Following GaussMark (Block et al., 2025), we tune the Gaussian perturbation scale $\sigma$ on a test split by sweeping over $\sigma$ and measuring detection power at a fixed significance level $\alpha = 0.05$. GaussMark is a training-time method that requires gradient access for verification, so it is not directly comparable to the inference-time watermarks in our main hierarchy. At $\sigma = 0.05$, GaussMark achieves perfect detection with TPR equal to 1.00 at the target FPR. Larger values of $\sigma$ in Table 12 push the detection statistic $\psi$ higher but start to incur false positives. This is similar to the trade-off between embedding strength and utility degradation that we observe with inference-time watermarks.

