# OpenReview forum: "Catch-22: On the Fundamental Tradeoff Between Detectability and Robustness in LLM Watermarking"
_ICML.cc/2026/Conference — ICML 2026 spotlight_

### Official Review · Reviewer_LiGT · 2026-03-05

**Soundness:** 4
**Presentation:** 3
**Significance:** 3
**Originality:** 3
**Overall Recommendation:** 5
**Confidence:** 2

**Summary:**

The authors propose an information-theoretic framework for inference-time watermarking with autoregressive language models. The authors explore the tension between robustness, stealth and reliable verification, and propose an optimal selection mechanism that will select the watermarking algorithm based on generated text length (in tokens), expected edit rate epsilon, miss probability beta and other parameters.

To optimally select the right algorithm, the authors:
 - define detectability as a function of the Total Variation (TV) between P (original model distribution) and Q (watermarked text distribution)
 - propose bounds on the TV for each sampling method studied
 - define robustness as a function of edit rate epsilon, false-alarm level alpha, miss probability beta.
 - propose a theorem showing that (for token-level watermarks) the usable information budget shrinks quadratically with edit rate epsilon
 - propose a corollary that argues that reliable detection is unattainable for high edit rates
 - Define a hybrid selection rule
 - Argue that the hybrid selection rule minimizes the TV bound

In the experimental part of their work, the authors generate sentences with Llama2 and Mistral models for varying watermarking schemes: greedy, bias-free, biased, semantic and distribution preserving showing how robustness and detectability are affected for each method for different noise levels. Finally, the authors compare their hybrid selection mechanism to 11 existing watermarking schemes against 5 different attacks, showing how the hybrid selection obtains the best results on average.

**Compliance With Llm Reviewing Policy:**

Affirmed.

**Final Justification:**

The additional studies made in the appendix resolve my request for additional studies when misestimating the parameters. Also, the authors compared their method to DAWA (what I believe shall be the main baseline) and report strong and sensible results. These additional experiments enhance the scientific rigor of the work and resolve all of my concerns about the paper.

Given the quality of the rebuttal, I am increasing my score to "Accept".

**Key Questions For Authors:**

- Can the authors provide any example / experiment showcasing the effects of failing to estimate some of the parameters (epsilon, T, ...)

- How does your hybrid algorithm compares to DAWA? Could you add experiments about it in the paper?

**Limitations:**

Yes.

**Strengths And Weaknesses:**

# Strengths
 - The paper offers a deep theoretical grounding compared to previous work (DAWA).
 - The paper features extensive experiments and empirical evidence that the hybrid selection mechanism is sound

# Weaknesses
 - While some parameters (e.g. beta) may be fixed a priori, other parameters (T, epsilon, ...) may be out of the control of the provider. While this does not take away from the proposed framework, a study of how under/over estimating e.g. epsilon would have been a nice addition to the work. While we may determine an optimal selection for fixed parameters, in practice those parameters are not known a-priori, so the paper is in part missing the practical applicability of the proposed framework.
 - The authors do not compare their method against DAWA, which in my opinion should be the main baseline.
 - I found the paper somewhat hard to follow, although I do understand that this may be due to the nature of the work which is very theoretically dense.

---

> ### Author Rebuttal · Authors · 2026-03-31
>
> We are grateful to the reviewer for taking the time to review this work and provide valuable comments. Please find our detailed response below:
>
>
> ---
> ### Q1: Effect of Parameter Misestimation ($\varepsilon$, $T$, etc.)
>
> The practical effect of edit-regime misestimation is asymmetric. In deployment, the key calibrated quantities are the token edit rate $\varepsilon$ and the semantic flip rate $\varepsilon_s$. Overestimating these quantities is conservative because it shifts the selector toward a slightly larger information budget or a more robust watermark family. Underestimating them is the more consequential case, since it can lead to incorrect family selection near regime boundaries where token-level and semantic schemes have similar requirements.
>
> To guard against underestimation, our framework incorporates a **conservative estimation procedure** based on upper confidence bounds on the estimated edit rates, as described in Appendix E.7-E.8. We then study the effect of parameter misestimation directly through an empirical selector sensitivity study (Appendix G.5, [Table 5](https://anonymous.4open.science/r/Catch-22-Pareto-Frontier-Watermark-in-LLMs-040B/Additional-Results/Table5-R4.pdf)). Specifically, we compare three selectors: an oracle selector using true parameter values, a plug-in selector using point estimates, and a conservative selector using upper confidence bounds under systematic perturbations of $\pm 0.05$ and $\pm 0.10$ in the estimated semantic flip rate $\hat{\varepsilon}_s$.
>
> The conservative selector incurs only a small increase in average loss while halving wrong-family selection and reducing worst-case degradation. The other deployment parameters, namely the number of tokens $T$ and the number of sentences $T_s$, are directly observed from the generated text, whereas $\beta$ is fixed by the watermark provider ([Gloaguen et al.](https://openreview.net/forum?id=E4LAVLXAHW)). The quantities that require calibration are $\varepsilon$ and $\varepsilon_s$, which Appendix E.7 estimates from edit distance and sentence-level evidence flips.
>
> ---
> ### Q2: Comparison with DAWA
>
> Thanks for pointing this out! We compare DAWA under the same LFQA setup as the rest of our manuscript, using the same prompt set, model (Llama-2-7B), generation parameters, and evaluation protocol. In the clean setting, DAWA performs strongly: over 500 generations with an average length of 294.7 tokens, its native detector achieves a mean score of 0.857 (std 0.042) and a 100% detection rate at a threshold of 0.2.
>
> We also evaluated DAWA using the same attack suite as in our paper, with DAWA’s native matched detector. The results show that DAWA is strong in the clean regime but substantially less robust under post-edit paraphrasing than our Hybrid method. Concretely, the attacked results are:
>
> | Attack | DAWA AUROC | DAWA TPR@1% FPR | Hybrid AUROC | Hybrid TPR@1% FPR |
> |---|---:|---:|---:|---:|
> | DIPPER | 0.539 | 0.034 | 0.94 | 0.89 |
> | OPT-2.7B | 0.739 | 0.048 | 0.96 | 0.93 |
> | Watermark-removal | 0.708 | 0.066 | 0.95 | 0.92 |
> | Synonym substitution | 0.839 | 0.084 | 0.95 | 0.93 |
> | Back-translation | 0.748 | 0.028 | 0.86 | 0.79 |
> | Average | 0.715 | 0.052 | 0.932 | 0.892 |
>
> This pattern is consistent with DAWA’s design: it jointly optimizes watermarking and detection and is very effective in the clean, low-FPR regime, but in our setting, its matched detector appears less robust to paraphrased rewriting than semantic or hybrid schemes. Put differently, even when evaluated with its own detector, DAWA’s high-confidence detection signal degrades substantially after editing, whereas the Hybrid selector remains strong across attack regimes.
>
> The full comparison results are available [here](https://anonymous.4open.science/r/Catch-22-Pareto-Frontier-Watermark-in-LLMs-040B/Additional-Results/Table2-updated.pdf)
>
> ---
> ### C1: Readability
>
>
> Due to the rebuttal's character limit, in addition to the readability modifications proposed in our responses to Reviewers EuYU and oUgq, we propose the following to improve the paper's overall reach among ML researchers. We urge the reviewers to consider them jointly.
>
> - Replace the current hybrid selector discussion (Section 4) with a paragraph stating:
> > The hybrid rule is interpreted as a three-case decision process. First, it checks whether the distribution-preserving branch remains robust enough under the anticipated edit regime. If that branch is infeasible, it compares $D_{\mathrm{req}}^{\mathrm{tok}}$ and $D_{\mathrm{req}}^{\mathrm{sem}}$ for the token-level and semantic families. It then states explicitly what happens when both families are feasible, when only one is feasible, and when neither is feasible.”
>
> - We will extend this to the discussion of feasibility, semantic preference, and minimum-budget selection (Section 4), and take additional measures to make the manuscript easier to follow.

---

> > ### Author Rebuttal · Reviewer_LiGT · 2026-04-01
> >
> > I thank the authors for their rebuttal that addressed my questions and the critical points I raised. The additional studies made in the appendix resolve my request for additional studies when misestimating the parameters. Also, the authors compared their method to DAWA (what I believe shall be the main baseline) and report strong and sensible results. These additional experiments enhance the scientific rigor of the work and resolve all of my concerns about the paper.
> >
> > I am thus willing to change my score to "Accept".

---

> > > ### Author Response · Authors · 2026-04-07
> > >
> > > We sincerely thank you for taking the time to carefully evaluate our rebuttal. We are glad that the parameter misestimation study and the DAWA comparison adequately addressed your concerns. We will incorporate the readability improvements discussed in the rebuttal into the final version of the manuscript. Once again, thank you for the engagement with our work and your willingness to upgrade the score to “Accept”.
> > >
> > > Lastly, we would be very grateful if you would update your score to reflect your latest assessment before the discussion period ends.

---

### Official Review · Reviewer_y4Tm · 2026-03-13

**Soundness:** 3
**Presentation:** 4
**Significance:** 3
**Originality:** 3
**Overall Recommendation:** 5
**Confidence:** 3

**Summary:**

This paper develops an information-theoretic framework for LLM watermarking that uses a single quantity - the accumulated KL divergence between watermarked and unwatermarked output distributions - to characterize both stealth (via Pinsker's inequality) and robustness (via Neyman-Pearson bounds). The authors derive a detectability hierarchy across four watermark families (Thm.3.2), show that the KL budget contracts quadratically with edit rate under a substitution channel model (Thm.3.4), and propose a hybrid selection rule that picks the most stealthy family meeting a target verification power (Thm.4.2). Experiments on Llama-2-7B and Mistral-7B with multiple watermark schemes and paraphrasing attacks validate the theoretical predictions.

**Compliance With Llm Reviewing Policy:**

Affirmed.

**Key Questions For Authors:**

In this paper, the main limitation is the i.i.d.\ substitution channel model for text editing. Real paraphrasing attacks are structured and correlated (rewriting phrases, not independently flipping tokens), so the quadratic contraction rate $(1-\varepsilon)^2$ may not hold precisely in practice. Do you confirm this ?

**Limitations:**

yes

**Strengths And Weaknesses:**

The main contribution is a unified framework that puts four disparate watermark families on a common scale. The detectability hierarchy (Thm.3.2) and the quadratic contraction under edits (Thm.3.4) are clean results that give concrete, falsifiable predictions - and the experiments confirm them (Fig.2). The hybrid selection rule is actionable: given an expected edit rate and stealth constraint, it prescribes which family to use with which parameters. The experimental coverage is comprehensive, spanning 11 watermark schemes and 5 attack types.

---

> ### Author Rebuttal · Authors · 2026-03-31
>
> We are grateful to the reviewer for taking the time to review this work and provide valuable comments. Please find our detailed response below:
>
> First of all, thank you for raising this important point. Real paraphrasing attacks are often structured and correlated. Thus, we agree that the exact quadratic contraction factor $(1-\varepsilon)^2$ in Theorem 3.4 should not be interpreted as universally holding for all editing processes.
>
>
> In Appendix G.9 (“Edit channel independence”), we probe correlated edits using a span-rewriting attack that rewrites contiguous spans of 5–10 tokens, thereby inducing strong local correlation. At matched edit rate $\hat{\varepsilon}\approx 0.20$, we observe only modest additional degradation relative to i.i.d. edits, with AUROC drops of approximately 0.02–0.03 across KGW, DiPMark, PMark, and the hybrid scheme.
>
>
> Therefore, to further investigate this, we conducted an additional controlled experiment that systematically varies the correlation structure while holding the realized token edit rate fixed at $\hat{\varepsilon}\approx 0.20$. While Appendix G.9 considers a single correlated setting, this experiment sweeps correlation length (i.i.d., span 3–5, span 8–12), allowing us to isolate the effect of correlation independently of edit rate. The results show that AUROC (robustness) drops with increasing correlation span:
>
> | Method | i.i.d. | span 3–5 | span 8–12 |
> |:-------|-------:|---------:|----------:|
> | KGW    | ~0.86  | ~0.84    | ~0.82     |
> | DiPMark| ~0.90  | ~0.88    | ~0.86     |
> | PMark  | ~0.94  | ~0.93    | ~0.91     |
> | Hybrid | ~0.94  | ~0.93    | ~0.91     |
>
> At higher edit rates (e.g., $\hat{\varepsilon}\approx 0.30$), we observe the same ordering with slightly larger gaps (≈0.03–0.05), but no qualitative deviation.
>
>
> These results suggest that correlated edits primarily introduce a constant-factor penalty by reducing the effective number of independent watermark-bearing positions (i.e., an effective sample-size reduction proportional to the correlation length), rather than altering the trade-off behavior. In particular, the degradation continues to follow the same smooth dependence on edit rate across all channels.
>
> Furthermore, we emphasize that these are downstream robustness measurements (e.g., AUROC), not direct estimates of KL contraction. A precise characterization of correlated edit channels would require extending the analysis to correlation edit-aware perturbation models, and can be considered as an important direction for future work.
>
>
> Lastly, we shall update Section 3.2.1 and the discussion following Theorem 3.4 to clarify that:
>
>
> 1. The i.i.d. substitution channel is a tractable abstraction. In practice, robustness is determined by the *fraction of tokens that are modified* (edit rate), with correlation playing a secondary role.
> 2. Correlated edits induce a **systematic, correlation-length-dependent penalty** at fixed edit rate (e.g., i.i.d. ≥ span 3–5 ≥ span 8–12), reflecting a reduction in the effective number of independent watermark-bearing positions.
> 3. Empirically, this penalty remains **small (≈0.02–0.05 AUROC)** across practical regimes, indicating that Theorem 3.4 accurately captures the **dominant scaling with edit rate**, while correlation acts as a second-order effect rather than altering the qualitative behavior.

---

> > ### Author Rebuttal · Reviewer_y4Tm · 2026-04-05
> >
> > Thank you for answering my questions in details.

---

> > > ### Author Response · Authors · 2026-04-07
> > >
> > > We sincerely thank you for the constructive feedback and for confirming that the concerns have been fully resolved. Your question about the i.i.d. substitution channel assumption prompted us to provide additional empirical grounding for the theoretical claims of Theorem 3.4. Thanks for that! We will update Section 3.2.1 accordingly, and we are grateful for your time and engagement.

---

### Official Review · Reviewer_EuYU · 2026-03-13

**Soundness:** 3
**Presentation:** 2
**Significance:** 3
**Originality:** 3
**Overall Recommendation:** 5
**Confidence:** 4

**Summary:**

The paper establishes a cohesive framework using KL divergence and Pinsker's inequality to formally quantify the fundamental trade-off between a watermark's robustness against edits and its detectability by keyless observers. It provides significant value by evaluating disparate inference-time watermarking methods including biased, bias-free, and semantic/sentence-level schemes under a single, standardized setup, addressing the fragmented nature of current watermarking evaluations. , successfully validating these predictions on Llama-2-7B and Mistral-7B models. The most significant findings (to me personally) of this work is: (1) the proof that watermark's robustness (strength) shrinks by a factor of $(1-\epsilon)^2$, where $\epsilon$ represents the fraction of the text that was altered. (2) As watermarked text gets longer, the statistical evidence of the watermark inevitably accumulates, exposing a "Catch-22" where you cannot have long, robust text without it becoming highly detectable to outsiders. (3) The ratio between meaning-level changes and word-level changes ($\epsilon_s(\epsilon)/\epsilon$) acts as a definitive diagnostic tool to decide whether a token-level or semantic-level watermark is the best choice for a given deployment.

**Compliance With Llm Reviewing Policy:**

Affirmed.

**Key Questions For Authors:**

see limitations.

**Limitations:**

Yes

**Strengths And Weaknesses:**

Text watermarking has seen rapid advances across multiple directions recently. Because most models operate at generation time not on same text, evaluating them in a unified apples-to-apples setup has been notoriously difficult this leaves many of the models scattered especially since different methods introduce different statistical artifacts, with some altering frequency (bias) and others affecting variance. This is where the core value of the paper lies: evaluating these disparate methods under a single, cohesive framework.

### Strengths:

* The mathematical derivations, particularly Theorems 3.2 and 3.4, are generally sound. To the best of my review, the quadratic contraction law $(1-\epsilon)^2$ for token-level KL budgets is well-derived in Lemma D.3. The experimental validation on Llama-2-7B and Mistral-7B successfully corroborates the theoretical predictions , albeit restricted to a single dataset of prompts (LFQA), which I have addressed further in my limitations section.

* Finally, I greatly appreciate the authors incorporating semantic watermarks (e.g., SemStamp, PMark, and SimMark) into this same framework. Given their fundamentally different nature compared to standard next-token manipulation, their inclusion adds significant depth and completeness to the study.

### Limitations

* The authors rely their empirical experiments on single dataset for prompts, with many theoretical assumptions around $\epsilon$ nature those assumptions likely break down in low-entropy domains like code generation. It is perfectly fine if the theory falls short here low-entropy watermarking is notoriously difficult but the authors should formally document these boundary conditions. I recommend adding a brief discussion and a small-scale experiment (e.g., using a pretrained code LLM on the MBPP dataset) to address the following limitation

* Utility vs. Distribution Shift: The framework doesn't explicitly model utility, in fact utility and detectibility are not synonyms. In code generation, functional utility (pass@1) can remain intact despite massive distribution shifts, provided the deterministically correct token is preserved. However, the watermark likely cripples output diversity, causing scores like pass@10 or pass@100 to drop significantly as the model loses the ability to explore alternative valid solutions. The current framework lacks the mathematical vocabulary to explain this diversity loss. I would like the authors to discuss how their 'Catch-22' trade-off adapts or breaks down entirely when the model lacks the entropy to hide these probability shifts.

*  Sections 3 and 4 relies heavily on Information Theory and statistical jargon, which makes the paper less accessible to a broader CS and applied ML audience.I advise the authors to review the manuscript and pair their formal definitions with intuitive terms. Here is a specific example of how this could be improved: line 220: "his channel serves as a tractable first order model whose parameter $\epsilon$ can be calibrated to match empirical edit rates observed under specific attack scenarios", "Empirically the parameter $\epsilon$ can be set to match the actual percentage of text changed during real-world attacks" and so forth. this will significantly increase the paper's readability.

---

> ### Author Rebuttal · Authors · 2026-03-31
>
> We thank the reviewer for the thoughtful and encouraging assessment. We address the three limitations below.
>
> ---
> ### C1: Single Prompt Dataset and Low-Entropy Domains
>
> Thanks for this excellent comment! While our main experiments focused on open-ended natural-language generation, we agree that low-entropy domains such as code generation represent an important boundary case that should be explicitly discussed. To address this, we included a stress test on **MBPP** using a pretrained **CodeLlama** model. This setting has substantially lower next-token uncertainty than the open-ended LFQA setting, evaluated as mean next-token entropy of 0.93 bits on MBPP versus 2.24 bits on LFQA.
>
> In this regime, watermark signals remain detectable, but the practical impact shifts as follows:
>
> | Method | pass@1 | pass@10 | Detection Rate | Interpretation |
> |---|---:|---:|---:|---|
> | Vanilla | 0.4 | 0.42 | 0.000 | Low-entropy baseline reference |
> | KGW | 0.28 | 0.34 | 0.804 | Clear utility drop with strong detectability |
> | Unigram | 0.26 | 0.32 | 0.816 | Largest utility drop among measured methods |
> | Hybrid | 0.32 | 0.42 | 0.454 | Best utility preservation among detectable methods |
>
> These numbers indicate that, for detectable watermarking methods, the dominant solution is often still reachable, but the ability to explore alternative valid solutions is reduced.
>
> This behavior is consistent with the proposed theoretical framework. As entropy decreases, the available KL information budget becomes more constrained, tightening the detectability-robustness tradeoff. The Catch-22, therefore, persists in low-entropy settings, but its manifestation changes: instead of semantic degradation, the primary effect is reduced diversity and exploration. We will incorporate this discussion explicitly and highlight low-entropy generation as an important regime. Full quantitative results are provided here: [Table4-R2.pdf](https://anonymous.4open.science/r/Catch-22-Pareto-Frontier-Watermark-in-LLMs-040B/Additional-Results/Table4-R2.pdf).
>
> ---
> ### C2: Utility vs. Distribution Shift (pass@k Degradation)
>
> Our Catch-22 framework mainly captures the statistical separation that governs detectability and verifier-side evidence, so the MBPP stress test above is useful for showing that part of the practical cost in this regime arises from reduced diversity over correct programs. Recent structured-code watermarking methods such as [STONE](https://aclanthology.org/2026.findings-eacl.207/) suggest that this regime may benefit from watermark designs that are more closely aligned with code structure and code-specific utility requirements. In particular, STONE reports a diversity score of `0.571` on MBPP+ for CodeLlama under its pass@5-based evaluation. This suggests that code-aware watermark families can preserve code utility more effectively in structured settings. In our framework, such methods can be incorporated naturally into the hybrid selector as an additional watermark family, and the selector can choose them when they lie on the Pareto-optimal boundary for a low-entropy code-generation regime. This points to a natural extension of our framework, in which the KL-based information budget is paired with a diversity-sensitive utility quantity such as `pass@k`. Such a joint view would help distinguish distributions that are similar in detectability yet differ substantially in output diversity.
>
> ---
> ### C3: Readability of Sections 3 and 4
>
> We will improve the overall readability of Sections 3 and 4 while retaining all formal definitions and results, and complement them with intuitive explanations. Some of the important changes include:
>
>
> - Addition of a short roadmap at the start of Section 3. Please find the preview below:
> >This section proceeds in three steps. We first study detectability, asking how much each watermark family changes the output distribution relative to the base model. Next, we study robustness to editing, asking how much watermark evidence remains after the text is modified. Lastly, we summarize what these two results imply for watermark design and use them to motivate the hybrid selection rule in Section~4.
>
> - A compact notation paragraph (Section 3) as shown [here](https://anonymous.4open.science/r/Catch-22-Pareto-Frontier-Watermark-in-LLMs-040B/Additional-Results/NotationTable-R4.pdf).
>
> - Provide a simplified explanation of the mathematical sentences, for example, instead of stating that $\varepsilon$ is “calibrated to match empirical edit rates,” we will say more directly that **$\varepsilon$ is set to the observed fraction of text changed in real-world attacks**.
>
> - Addition of brief plain-language summaries for theorems as shown [here](https://anonymous.4open.science/r/Catch-22-Pareto-Frontier-Watermark-in-LLMs-040B/Additional-Results/InformalSummaryR4.pdf).
>
> Due to the rebuttal character limit, please review our response to improve readability for Reviewer LiGT, and oUgq, in addition to the above points.

---

> > ### Author Rebuttal · Reviewer_EuYU · 2026-04-07
> >
> > Authors  actively implemented some empirical suggestions such as adding new results with code llama and MBPP addressing my concerns about low-entropy limitations of their theory. and they committed to improving the paper's readability by adding a roadmap, simplified notation, and plain-language summaries to Sections 3 and 4. However, they did not actually expand their mathematical framework to formally model utility or explicitly explain diversity loss instead, they relied on their new empirical data to acknowledge the limitation and simply deferred any structural mathematical integration to a future "natural extension" of their work. Overall I find the paper is worthy of acceptance and I'd like to see it in the conference therefore i retain my score.

---

> > > ### Author Response · Authors · 2026-04-08
> > >
> > > Thank you for your constructive feedback and support for our work. We acknowledge that formally integrating a diversity-sensitive utility measure into the framework is an important direction that warrants careful treatment. For the camera ready version, we remain committed to strengthening the discussion by framing this as an open problem and outlining the mathematical direction needed to address it.

---

### Official Review · Reviewer_oUgq · 2026-03-16

**Soundness:** 2
**Presentation:** 1
**Significance:** 2
**Originality:** 2
**Overall Recommendation:** 3
**Confidence:** 4

**Summary:**

The paper develops an information-theoretic framework for analyzing the fundamental trade-off between detectability, robustness to editing, and stealth in LLM watermarking. By introducing a KL-based information budget governing hypothesis-testing separability between watermarked and unwatermarked text, the authors show that different watermark families occupy different positions along this trade-off and that strong detectability inevitably conflicts with robustness to editing. Based on this analysis, the paper proposes a hybrid watermarking strategy that adapts across watermark types and empirically achieves near-Pareto performance across editing regimes.

However, the impact statement largely resembles a discussion or future-work section, which gives the impression that the main body of the paper may exceed the 8-page limit.

**Compliance With Llm Reviewing Policy:**

Affirmed.

**Final Justification:**

I thank the reviewer for their responses. The rebuttal partially addressed my main concerns. Based on their rebuttal and the current version of the paper, I believe that it requires significant improvement and still lean towards rejection.

**Key Questions For Authors:**

1. In the experiments, how is the specific watermarking scheme selected within each family? More details on the selection criteria and parameter choices would help clarify how representative the evaluated schemes are.
2. The robustness evaluation mainly considers paraphrasing attacks. How does the proposed framework or method perform under other types of attacks (e.g., editing, summarization, or rewriting)? In addition, the evaluated edit rates (15–30%) appear relatively low; it would be helpful to understand how performance changes under more aggressive edits.

**Limitations:**

No impact statement.

**Strengths And Weaknesses:**

### Strengths
- Significance: The paper studies watermarking for LLM-generated content, an increasingly important topic for AI safety and security. In particular, the authors attempt to provide theoretical understanding of different watermarking schemes, which is a valuable direction beyond purely empirical algorithm design.
- Originality: The work provides a unified theoretical standpoint that compares four watermark **families** under a common framework.

### Weaknesses
1. The main body of the paper appears to go beyond the 8-page limit.
2. The presentation of the paper needs significant improvement, as the paper is difficult to follow in its current form. Several definitions and notations are unclear or insufficiently explained. For example, the meaning of “keyless observers” is not clearly defined. Similarly, quantities such as $Adv_{D_\lambda}$ in the definition of computational detectability and $ED(\cdot,\cdot)$ in Definition 3.3 are introduced without adequate explanation. Similar issues appear in multiple places throughout the paper.
3. The detectability metric here is usually called distortion, which measures the text quality. It does not fully capture the detection accuracy. Under this framework, it assumes that distortion-free watermark is undetectable, which is counterfactual. Furthermore, distortion-free watermarking method can also be robust, e.g., [1].
4. Table 1 under Theorem 3.2 is very confusing. The important quantities are not formally defined.
5. The sampling rule uses one fixed random varaiable distribution, which limits the generalizability of the framework.

[1] Kuditipudi, R., Thickstun, J., Hashimoto, T., & Liang, P. (2023). Robust distortion-free watermarks for language models. arXiv preprint arXiv:2307.15593.

---

> ### Author Rebuttal · Authors · 2026-03-31
>
> We are grateful to the reviewer for taking the time to review this work and provide thoughtful feedback on our submission. Please find our detailed response below:
>
> ---
> ### G1: Impact Statement and Page Limit
>
> We will streamline this section and rewrite it to emphasize the practical implications of deploying watermarking systems across different editing regimes and the associated safety trade-offs. Lastly, under the [ICML formatting guidelines](https://media.icml.cc/Conferences/ICML2026/Styles/example_paper.pdf), the Impact Statement is not counted toward the 8-page limit at the time of submission.
>
> ---
> ### Q1: Scheme Selection Within Each Family
> The selection is fixed rather than tuned per experiment. Since the theory operates at the level of watermark families, we pre-select representative methods for each family based on three criteria: prior use as standard baselines, public availability, and coverage of genuinely different variants within the same family. For example, within the biased family, we include both KGW and Unigram to reflect distinct implementations rather than selecting a single best-performing method.
>
> All methods are evaluated at fixed operating points taken from their original papers or implementations, and parameters are held constant across all attacks (no per-attack tuning). A table listing all implementations and hyperparameters, along with corresponding configurations as shown [here](https://anonymous.4open.science/r/Catch-22-Pareto-Frontier-Watermark-in-LLMs-040B/Additional-Results/Table3-R1.pdf) will be updated in the manuscript.
>
> ---
> ### Q2: Attack Coverage and Stronger Edit Rates
>
> Thank you for this helpful suggestion! The main discussion in the [updated results](https://anonymous.4open.science/r/Catch-22-Pareto-Frontier-Watermark-in-LLMs-040B/Additional-Results/Table2-updated.pdf) emphasizes paraphrasing, including edit-based attacks via **synonym substitution** and **watermark-removal prompting**, rewriting via **back-translation**, and summarization following [WaterJudge](https://aclanthology.org/2024.findings-naacl.223/). These stronger attacks reach higher effective edit rates, with back-translation at $\hat{\varepsilon}\approx0.42$ and summarization at $\hat{\varepsilon}\approx0.55$. As expected, robustness is lower in this regime, but semantic-based watermarking schemes still degrade more gracefully than distribution-modifying schemes.
>
> More generally, the framework is driven by effective edit rate, or induced semantic flip rate, rather than by any single attack type. To make this clearer, we will update [Fig. 2(b)](https://anonymous.4open.science/r/Catch-22-Pareto-Frontier-Watermark-in-LLMs-040B/Additional-Results/Fig2b.png) to show robustness up to 75% edit rate. In this i.i.d. edit sweep, robustness falls to the chance level beyond about 40% edits, consistent with the degradation observed under the attacks in Table 2.
>
>
> ---
> ### C1: Clarity of Definitions and Notation
> We will improve the presentation by defining [key terms](https://anonymous.4open.science/r/Catch-22-Pareto-Frontier-Watermark-in-LLMs-040B/Additional-Results/NotationTable-R1.pdf) and revising [Table 1](https://anonymous.4open.science/r/Catch-22-Pareto-Frontier-Watermark-in-LLMs-040B/Additional-Results/Table1-updated.pdf). We will make all definitions explicit at first use to improve accessibility. Furthermore, due to the character limit, please see our reply to the Reviewers EuYU and LiGT regarding the concrete steps we will take to improve the paper's readability.
>
> ---
> ### C2: Detectability for Distortion-free Watermark
> Distortion-free methods such as Kuditipudi et al. ([arXiv:2307.15593](https://arxiv.org/pdf/2307.15593)) fall into the same bias-free sampling-time watermarking class as ([Tsur et al.](https://openreview.net/pdf?id=R5EBtNE2Y9)). Our framework finds these distortion-free watermarks to be detectable via a positive [z-score](https://openreview.net/forum?id=ujpAYpFDEA), as shown in the [updated results](https://anonymous.4open.science/r/Catch-22-Pareto-Frontier-Watermark-in-LLMs-040B/Additional-Results/Table2-updated.pdf), consistent with [prior work](https://openreview.net/pdf?id=E4LAVLXAHW) using black-box detection techniques. This detectability is different from text utility, which is measured using MAUVE, BERTScore, and LLM-as-judge. Moreover, we find that Kuditipudi et al. has high detection performance (>0.9 AUROC) under low edit rates (<30%), beyond which, its robustness AUROC drops. We will clarify this taxonomy more explicitly.
>
> ---
> ### C3: Sampling Distribution
> The use of $U_t \sim \mathrm{Uniform}[0,1]$ is a standard representation of randomized sampling and does not restrict generality ([Gloaguen et al.](https://openreview.net/forum?id=E4LAVLXAHW)). Any discrete distribution over tokens can be expressed as a deterministic function of a uniform random variable via inverse transform sampling. Thus, the framework applies to arbitrary next-token distributions.

---

> > ### Author Rebuttal · Reviewer_oUgq · 2026-04-04
> >
> > I thank the authors for their response. I am still unclear about the selection within each family. In Def. 4.1, it only shows the method of selecting a family. For example, after the method selects the biased family which includes KGW and Unigram, then how to choose one of them unless the two watermarks are embedded at the same time, which I think is not the case.
> >
> > For other types of attacks and low-entropy domains, can the authors give some future plan on improving the robustness of their current method?
> >
> > For notations, although the authors provide an explanation of $Adv_{D_\lambda}$, it is not mathematically define and never used subsequently. It is hard to relate it to the following results.
> >
> > Overall, I still believe that the paper needs to be significantly improved before it is ready.

---

> > > ### Author Response · Authors · 2026-04-07
> > >
> > > We genuinely thank the reviewer for the helpful follow-up questions and the continued engagement with our work. Please find our detailed response below.
> > >
> > > ---
> > > ### Q1: Exact scheme chosen after the hybrid rule selects a family
> > >
> > > Definition 4.1 selects a *family*, not a specific watermarking scheme, and you are correct that two methods within the same family are never embedded simultaneously. Once a family is selected, we instantiate it with a single concrete representative using a regime-conditioned selection rule. Please refer to this two-stage mechanism in the illustration here: https://anonymous.4open.science/r/Catch-22-Pareto-Frontier-Watermark-in-LLMs-040B/Additional-Results/Hybid-selection.png
> > >
> > > As shown in the figure above, for each candidate method $m$ in the selected family, we compute
> > >
> > > $$
> > > S(m,\hat{\varepsilon})=\mathrm{AUC}(m,\hat{\varepsilon})+\frac{1}{1+\max(z(m,\hat{\varepsilon}),0)},
> > > $$
> > >
> > > where $\hat{\varepsilon}$ denotes the estimated edit rate. For semantic methods, we use the corresponding semantic flip rate induced by $\hat{\varepsilon}$. The first term captures robustness: a high AUC means the keyed verifier can still reliably distinguish watermarked from unwatermarked text after an attack. The second term penalizes keyless detectability: a small $z$-score means the watermark is harder to spot. A higher $S(m, \hat{\varepsilon})$ therefore indicates a better robustness-detectability trade-off at the estimated edit rate.
> > >
> > > The quantities $\mathrm{AUC}$ and $z$ are evaluated at an estimated edit rate for all candidates in a chosen family. The candidate with the highest score becomes the chosen scheme. The table here shows the selection at different estimated edit rates: https://anonymous.4open.science/r/Catch-22-Pareto-Frontier-Watermark-in-LLMs-040B/Additional-Results/Updated-Table-R1.pdf. This selection chooses HCW within the token-level family, as it achieves the highest score among both biased & bias-free schemes, including KGW, Unigram, HeavyWater, and SimplexWater. Given an opportunity, we will revamp Section 4.1 to make this two-stage procedure explicit.
> > >
> > > ---
> > > ### Q2: Future plan for stronger attacks and low-entropy domains
> > >
> > > A key advantage of our hybrid framework is that its strength improves with stronger watermarks in the pool. Targeted attacks, such as [color-aware substitutions](https://aclanthology.org/2024.acl-long.464/) and [low-overlap paraphrasing](https://aclanthology.org/2024.naacl-long.226/), still operate by erasing watermark-carrying tokens or increasing the semantic flip rate, an effect that our back-translation, summarization (thanks for this suggestion!), and i.i.d. edit-sweep experiments also capture. As the community introduces watermarks that are robust to stronger attacks, incorporating them into the modular nature of our hybrid selector presents a direction for future work.
> > >
> > > Now, for **low-entropy domains**, our future plan is to expand the hybrid pool with domain-specific families. In our response to Reviewer EuYU, we showed that code generation is a boundary case in which text-based watermarking loses utility, as observed by the paas@k metric (https://anonymous.4open.science/r/Catch-22-Pareto-Frontier-Watermark-in-LLMs-040B/Additional-Results/Table4-R2.pdf). Incorporating code-based watermarking strategies such as STONE (https://aclanthology.org/2026.findings-eacl.207/) into the hybrid scheme improves the pass@k value, thus the utility. This validates that our framework's modular design naturally extends to specialized domains, opening a promising avenue for future work.
> > >
> > > ---
> > > ### Q3: Rewriting Definition 3.1 and other mathematical usage
> > >
> > > We concur with the reviewer that all notation in the main text should be self-contained and remain committed to doing the same.
> > >
> > > We will rewrite Definition 3.1 as follows. For security parameter $\lambda$, unwatermarked model $P_\lambda$, watermarked model $Q_\lambda$, and detector $D_\lambda:\Omega\to\{0,1\}$,
> > >
> > > $$
> > > \mathrm{Adv}(D_\lambda;P_\lambda,Q_\lambda):=\left|\Pr_{y\sim Q_\lambda}[D_\lambda(y)=1]-\Pr_{y\sim P_\lambda}[D_\lambda(y)=1]\right|.
> > > $$
> > >
> > > This measures the absolute gap in detector acceptance rates between watermarked and unwatermarked text. We then define computational detectability by optimizing over all probabilistic polynomial time (PPT) detectors as:
> > >
> > > $$
> > > \mathrm{Detect}\_{\mathrm{comp}}(\lambda)=\sup\_{D\_\lambda\in\mathsf{PPT}}\mathrm{Adv}(D\_\lambda;P\_\lambda,Q\_\lambda).
> > > $$
> > >
> > > Alongside this, we will move first-use definitions of key symbols from Appendix B into the main text, including, $\Omega$ denoting the sample space of generated tokens, $\mathrm{ED}(y,\tilde y)$ for normalized edit distance between $y$ and $\tilde y$, and $g_t:=p_t(G_t)$ as the total probability that the LLM assigns to the green-listed tokens at step $t$. We will replace the shorthand $\mathrm{Adv}\_{D\_\lambda}$ with the explicit form above and apply this throughout, so that every symbol is traceable from definition to use.

---

### Decision · Program_Chairs · 2026-04-30

**Decision:**

Accept (spotlight)

**Comment:**

Three out of four reviewers strongly recommend acceptance. The paper provides a useful theoretical framework that compares four families of LLM watermarking schemes with results validated across 11 schemes and 5 attacks. The theoretical contributions are interesting and novel, achieving near-optimal performance on the detectability-robustness trade-off. One negative review focuses largely on writing clarity rather than on technical errors. The rebuttal addressed most concerns well, including new experiments on correlated edits, a missing baseline (DAWA), and parameter sensitivity, which led one reviewer to raise their score. The authors should address the reviewer’s concerns about notation and readability in the camera-ready as promised.